# Myotonic dystrophy RNA toxicity alters morphology, adhesion and migration of mouse and human astrocytes

Diana M. Dincã[1,8], Louison Lallemant [1,8], Anchel González-Barriga [1,8], Noémie Cresto [2], Sandra O. Braz [1,3], Géraldine Sicot [1], Laure-Elise Pillet[2,4], Hélène Polvèche[5], Paul Magneron [1], Aline Huguet-Lachon[1], Hélène Benyamine[1], Cuauhtli N. Azotla-Vilchis [6], Luis E. Agonizantes-Juárez[6], Julie Tahraoui-Bories [5], Cécile Martinat[5], Oscar Hernández-Hernández [6], Didier Auboeuf [7], Nathalie Rouach [2], Cyril F. Bourgeois [7], Geneviève Gourdon [1✉] & Mário Gomes-Pereira [1✉]

Brain dysfunction in myotonic dystrophy type 1 (DM1), the prototype of toxic RNA disorders, has been mainly attributed to neuronal RNA misprocessing, while little attention has been given to non-neuronal brain cells. Here, using a transgenic mouse model of DM1 that expresses mutant RNA in various brain cell types (neurons, astroglia, and oligodendroglia), we demonstrate that astrocytes exhibit impaired ramification and polarization in vivo and defects in adhesion, spreading, and migration. RNA-dependent toxicity and phenotypes are also found in human transfected glial cells. In line with the cell phenotypes, molecular analyses reveal extensive expression and accumulation of toxic RNA in astrocytes, which result in RNA spliceopathy that is more severe than in neurons. Astrocyte missplicing affects primarily transcripts that regulate cell adhesion, cytoskeleton, and morphogenesis, and it is confirmed in human brain tissue. Our findings demonstrate that DM1 impacts astrocyte cell biology, possibly compromising their support and regulation of synaptic function.

[1] Sorbonne Université, Inserm, Centre de Recherche en Myologie, 75013 Paris, France. [2] Neuroglial Interactions in Cerebral Physiology and Pathologies, Center for Interdisciplinary Research in Biology, Collège de France, CNRS, Inserm, Labex Memolife, 75005 Paris, France. [3] Inserm UMR1163, Institut Imagine, Université Paris Cite, 75015 Paris, France. [4] Doctoral School N°562, Paris Descartes University, Paris 75006, France. [5] Inserm/UEVE UMR861, Université Paris Saclay I-STEM, 91110 Corbeil-Essonnes, France. [6] Laboratory of Genomic Medicine, Department of Genetics, National Rehabilitation Institute (INR-LGII), Mexico City, Mexico. [7] Laboratoire de Biologie et Modelisation de la Cellule, Ecole Normale Superieure de Lyon, CNRS, UMR 5239, Inserm, U1293, Universite Claude Bernard Lyon 1, 46 allée d'Italie, 69364 Lyon, France. [8]These authors contributed equally: Diana M. Dincã, Louison Lallemant, Anchel González-Barriga. ✉email: genevieve.gourdon@inserm.fr; mario.pereira@inserm.fr

Brain dysfunction in neurological diseases is frequently mediated by the impairment of neuronal and non-neuronal cells[1,2]. However, the contribution of non-neuronal cells to many of these disorders has been poorly investigated. Myotonic dystrophy type 1 (DM1) is a complex multisystemic disease that affects patients of all ages[3,4]. In addition to the typical involvement of the skeletal and cardiac muscle, the neurological manifestations are a prominent feature[5]. The DM1 cognitive profile is characterized by multiple deficits, such as global cognition, intelligence, social cognition, memory, language, executive, and visuospatial functioning. Personality and emotional traits include avoidant behavior, apathy anxiety-related disorders, and excessive daytime sleepiness[6]. The broad and heterogeneous cognitive neuropsychological profile of DM1 anticipates the involvement of different brain areas and neuronal circuits, a hypothesis that is corroborated by imaging studies[7,8]. While the involvement of frontal lobe is suggested by the prevalent executive dysfunction, hippocampus neuropathology may contribute to deficits in visuospatial memory and learning[6,9].

DM1 is caused by the expansion of a CTG trinucleotide in the 3′ untranslated region (UTR) of the *DMPK* gene. While unaffected individuals carry 5-37 CTG repeats, pathogenic expansions are longer than 50 CTG repeats, reaching >1000 CTG in the congenital form of the condition: larger repeats are associated with more severe symptoms and earlier onset[10]. RNAs containing expanded CUG triplets accumulate in the nucleus of DM1 cells, forming RNA foci that perturb the localization and function of RNA-binding proteins[11]. Among the mediators of disease, MBNL proteins are sequestered by toxic RNA foci, while CELF (CUG-BP and ETR-3-like factors) family members are upregulated. The dysregulation of these RNA-binding proteins perturbs several gene expression steps, such as transcription, alternative splicing and polyadenylation, translation, mRNA stability and mRNA intracellular localization[12].

In the central nervous system (CNS), *DMPK* is expressed in neuronal and non-neuronal cell types, which results in the accumulation of RNA foci in cortical neurons, astrocytes, and oligodendrocytes in DM1 brains[13,14]. However, we do not know the vulnerability of different cell types to disease, nor do we know their role in brain dysfunction. Interestingly, *DMPK* gene expression is higher in cortical astrocytes than in neurons isolated from adult human and mouse brains[15,16]. Mouse primary cell cultures confirmed the higher levels of *DMPK* transcripts and protein in cortical astrocytes, relative to hippocampal neurons[17]. Together, these results predict mechanisms of RNA toxicity in astrocytes and anticipate the involvement of this cell type in brain disease pathogenesis.

To investigate astrocyte contribution to DM1 brain disease, we used the DMSXL mouse model. Transgenic DMSXL mice carry more than 1,000 CTG repeats in the human *DMPK* locus[18], display toxic RNA foci and missplicing, in association with multiple phenotypes, such as myotonia, muscle weakness, cardiac and respiratory abnormalities[19–21], as well as behavioral and electrophysiological defects in the CNS[14,22]. Foci distribution in DMSXL brains revealed higher frequency in cortical astrocytes than in neurons[14], while co-culturing systems showed the reduced capacity of DMSXL astrocytes to protect neurons against glutamate neurotoxicity, as a result of the defective glutamate uptake[22]. Nevertheless, compelling evidence of astrocyte dysfunction and contribution to DM1 was lacking. In this study, we first report defective astrocyte morphology and orientation in DMSXL brains, as well as reduced astrocyte adhesion and spreading in mouse and human cell culture model systems. Mechanistically, we found robust MBNL-dependent splicing defects in astrocytes, affecting relevant transcripts that regulate cell adhesion, spreading, and membrane dynamics, in line with the cellular phenotypes of these cells.

## Results

**Astrocytes display abnormal morphology and polarity in the brain of DMSXL mice.** Given the higher frequency of nuclear RNA foci in cortical astrocytes relative to neighboring neurons that we previously observed in DMSXL mouse brains[14], we sought to investigate the impact of the CTG repeat expansion on astrocyte phenotypes in vivo. We first estimated astrocyte cell density in the mouse brain by immunodetection of two astrocyte-specific markers: SOX9, SRY-Box Transcription Factor 9; and S100B, S100 calcium-binding protein B (traditionally known as S100ß)[23]. The percentage of co-labeled cells (SOX9+/S100B+) in the frontal cortex was not significantly different between DMSXL and WT mice at 1 month of age, suggesting unaltered astrocyte density (Fig. 1a). Western blot revealed lower levels of astrocyte-specific GFAP (glial fibrillary acidic protein) in DMSXL frontal cortex and hippocampus (Fig. 1b). To investigate the whole-cell morphology of astrocytes, we expressed cytoplasmic GFP under the control of the *Gfap* promoter using AAVs[24] in the mouse frontal cortex to observe the thin and complex ramified morphology of cortical astrocytes (Fig. 1c). Morphological Sholl analysis of GFP-positive astrocyte processes revealed that DMSXL astrocytes were significantly less ramified than WT astrocytes, with a more severe phenotype in processes distanced 25 µm from the nucleus. To explore the progression of astrocyte hypotrophy, we investigated additional mouse ages. While DMSXL astrocyte ramification was unaltered at 10 days of age (Supplementary Fig. 1a), it was significantly reduced at 4 months (Supplementary Fig. 1b). Interestingly, the reduction in astrocyte processes was more severe at 4 months in processes distanced 10–15 µm from the nucleus (Supplementary Fig. 1c). GFAP downregulation appears to precede astrocyte shrinkage, because it was already detected at 10 days, persisting until the age of 4 months (Supplementary Fig. 1d). The parallel between GFAP downregulation (Fig. 1b) and the shrinkage of GFP-expressing processes in DMSXL mice (Fig. 1c) indicates hypotrophy of astrocyte cytoskeleton in DMSXL brains. In contrast, control mice carrying shorter 130 CTG repeats displayed normal astrocyte ramification (Supplementary Fig. 1e) and GFAP levels in frontal cortex (Supplementary Fig. 1f). Similarly, transgenic DM20 mice, which express higher levels of shorter *DMPK* transcripts with 20 CTG repeats[25] did not show altered GFAP levels (Supplementary Fig. 1g), indicating that astrocyte changes could not be accounted for by *DMPK* overexpression alone.

Starting at the third week of postnatal development, i.e., at the time of hippocampal synaptogenesis, the CA1 astrocytes of the *stratum radiatum* change the orientation of GFAP-rich stem processes to a fusiform orientation, perpendicular to the pyramidal cell layer[26], in a process that depends on the dynamic changes of the cytoskeleton and cell morphology. We tested whether such astrocyte polarity was perturbed by the expression of toxic CUG RNA, by performing GFAP immunohistochemistry in the hippocampus of DMSXL mice at 1 month (Fig. 1d). The extent of *stratum radiatum* astrocyte orientation was estimated by the measurement of a polarity index, corresponding to the ratio of crossing points between GFAP-positive processes and parallel or perpendicular axes to the pyramidal layer, as previously described[24]. Hippocampal astrocytes of DMSXL mice showed a significant reduction in their preferential orientation regarding the pyramidal layer, when compared to WT animals, indicative of defective polarization.

In conclusion, the analysis of DMSXL brains revealed postnatal reduction in astrocyte arborization and defective polarity in vivo, associated with alterations in cytoskeleton.

**DMSXL astrocytes show abnormal growth dynamics, reduced cell adhesion, and defective cell spreading.** Following the characterization of astrocyte phenotypes in vivo, we used primary cell

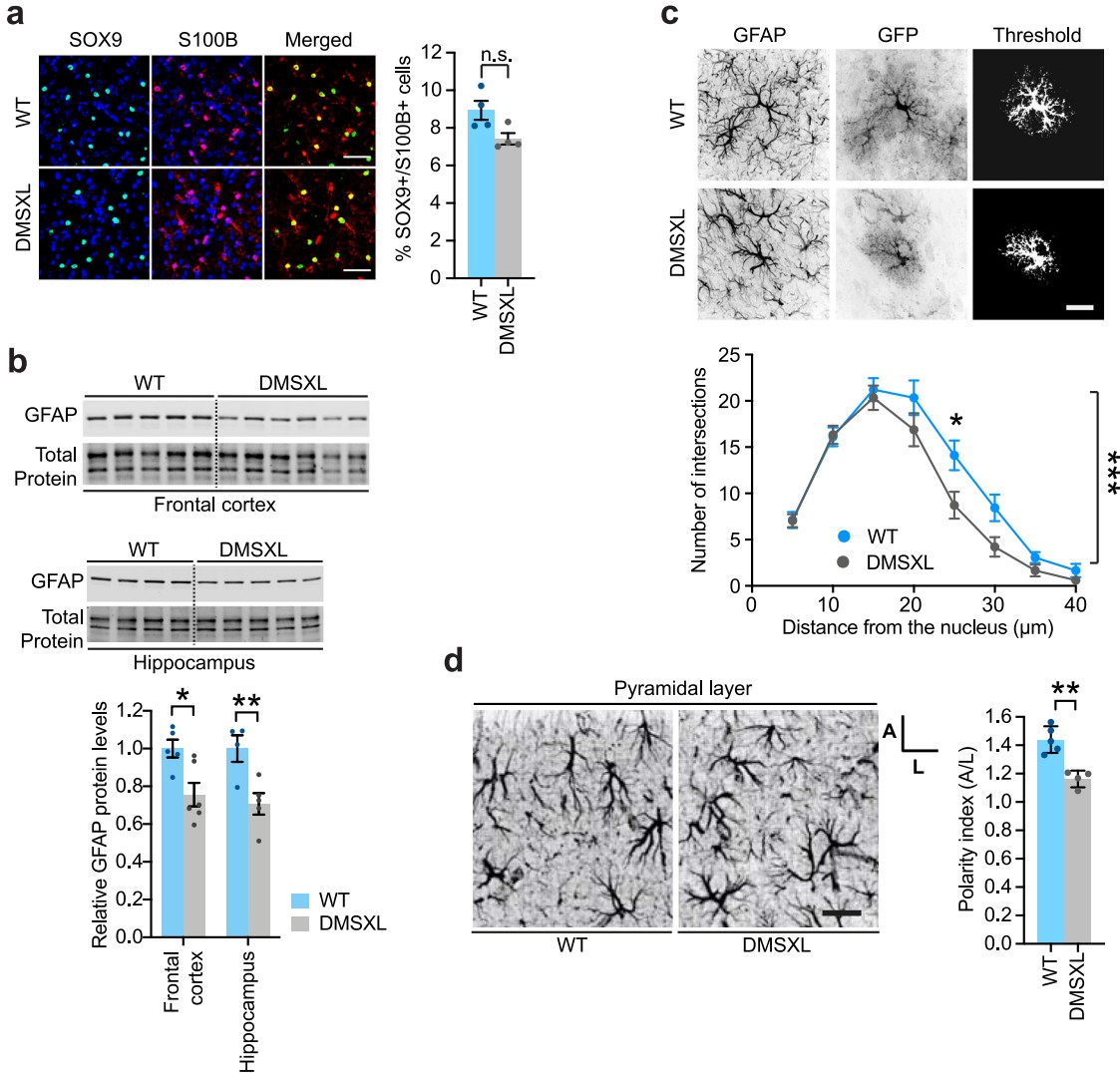

**Fig. 1 DMSXL astrocytes show abnormal morphology and polarity in vivo. a** SOX9 (green) and S100B (red) immunofluorescence in DMSXL frontal cortex at 1 month. Nuclei are stained with DAPI (blue). Scale bar, 50 μm. Quantification of double labeled astrocytes. Data are means ± SEM. $N = 4$ mice, $n = 6867$ cells, WT; $N = 4$ mice, $n = 6933$ cells, DMSXL ($p = 0.1143$, Two-tailed Mann–Whitney $U$ test). **b** Western blot quantification of GFAP in whole frontal cortex and hippocampus tissue lysates from DMSXL mice at 1 month. Representative stain-free protein bands are shown to illustrate total protein loading. Data are means ± SEM. Frontal cortex: $n = 5$, WT; $n = 6$, DMSXL. Hippocampus: $n = 4$, WT; $n = 5$, DMSXL ($p = 0.0165$, frontal cortex; $p = 0.0097$, hippocampus; Two-way ANOVA, Sidak post hoc test for multiple comparisons). **c** Sholl analysis of the branching of DMSXL cortical astrocytes at 1 month. Representative pictures illustrating the specificity of the GFP labeling in astrocytes expressing the GFAP marker. GFP signal was analyzed by applying the same threshold to all images. Scale bar, 20 μm. Data are means ± SEM. $N = 5$ mice, $n = 20$ cells, WT; $N = 4$ mice, $n = 20$ cells, DMSXL ($p = 0.0007$, Two-way ANOVA; $p = 0.0119$, 25 μm, Sidak post hoc test for multiple comparisons). **d** Analysis of astrocyte polarity with respect to the pyramidal cell layer in 1-month-old mice. Schematic representation of grid-baseline analysis for orientation quantification of GFAP-labeled CA1 *stratum radiatum* astrocytes. Polarity index ratio (A/L) larger than one indicates preferential perpendicular orientation towards the pyramidal layer. Scale bar, 20 μm. Data are means ± SEM. $N = 4$ mice, $n = 59$ cells, WT; $N = 5$ mice, $n = 64$ cells, DMSXL ($p = 0.0013$, Two-tailed Student's $t$ test). n.s. not significant; *$p < 0.05$; **$p < 0.01$; ***$p < 0.001$. Source data are provided as a Source Data file.

cultures highly enriched for individual cell types (Supplementary Fig. 2), to determine to what extend CUG RNA expression affects other aspects of astrocyte cell biology and investigate the underlying mechanisms. We first assessed the global impact of the DM1 repeat expansion using xCELLigence Real-Time Cell Analysis technology, which measures the electrical impedance of cell cultures, and provides real-time, non-invasive, quantitative readouts of cell culture growth[27]. DMSXL astrocytes exhibited a significantly lower cell index relative to control WT astrocytes (Fig. 2a). In contrast, impedance readings did not reveal significant changes in the population growth dynamics of primary DMSXL neurons, which remained unaltered over 72 h.

The lower cell index of DMSXL astrocytes relative to WT controls can be accounted for by multiple factors: altered proliferation and cell cycle dynamics, increased cell death, abnormal adhesion, and/or spreading of cultured cells. We investigated the contribution of each one of these factors to the abnormal growth of DMSXL astrocyte cultures. To analyze cell proliferation, we synchronized primary astrocytes by serum deprivation prior to bromodeoxyuridine (BrdU) incorporation and detection assay, combined with propidium iodide (PI) counterstaining to evaluate DNA content. Florescence-Activated Cell Sorting (FACS) did not reveal significant changes in the distribution between different phases of the cell cycle, hence

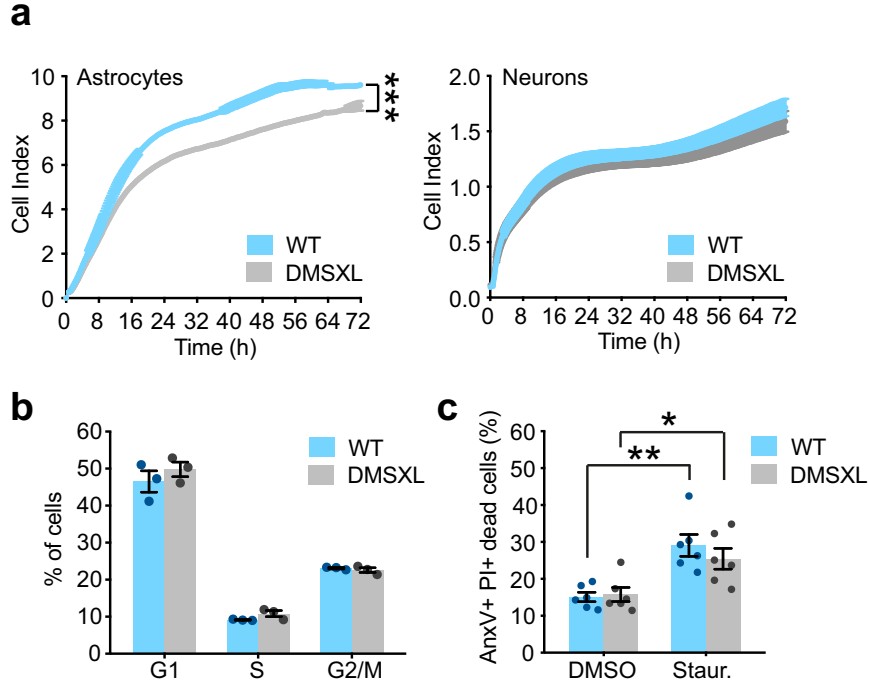

**Fig. 2 Primary DMSXL astrocytes display abnormal population dynamics. a** xCELLigence cell index of primary astrocytes and primary neurons over 72 h. Data are mean ± SEM, $n = 6$ independent cultures per group, each one established from a different animal ($p < 0.0001$, Two-way repeated measures ANOVA). **b** Percentage of DMSXL and WT astrocytes in G1, S, and G2/M phases of the cell cycle by FACS analysis of DNA content (PI levels) and DNA synthesis (BrdU incorporation). Data are mean ± SEM, $n = 3$ independent cultures per genotype. **c** FACS analysis of DMSXL and WT astrocyte cell death under control conditions (DMSO) and in the presence of Staurosporine (Staur). Percentage of apoptotic cells labeled with annexin V (AnxV+), and necrotic cells labeled with propidium iodide (PI+). Data are mean ± SEM, $n = 6$ independent cultures per genotype ($p = 0.0043$, WT; $p = 0.0240$, DMSXL; Two-way ANOVA and Sidak post hoc test for multiple comparisons). Source data are provided as a Source Data file. $*p < 0.05$; $**p < 0.01$; $***p < 0.001$.

excluding abnormal cell cycle progression of primary DMSXL astrocyte cultures (Fig. 2b and Supplementary Fig. 3a). We next tested the contribution of cell death by AnnexinV binding and PI incorporation, indicators of early apoptotic and late cell death, respectively. Similarly, FACS showed no difference in the number of AnnexinV- and PI-positive cells between DMSXL and WT astrocytes under basal culture conditions (Fig. 2c and Supplementary Fig. 3b). To test if DMSXL astrocytes were more vulnerable to stress, we treated cultures with staurosporine to induce apoptosis. Staurosporine enhanced the levels of cell death in both genotypes to a similar extent (Fig. 2c and Supplementary Fig. 3b). In conclusion, primary DMSXL astrocyte cultures displayed reduced growth, illustrated by abnormal impedance readings, which did not result from abnormal cell cycle progression or increased cell death.

We next asked if the lower cell index of DMSXL astrocytes could be a consequence of modified cell-substrate interactions. We plated primary DMSXL or WT astrocytes and monitored their adhesion and spreading by live cell videomicroscopy[28]. To exclude the confounding effect of different starting cell numbers, we first confirmed that the cell density of DMSXL and WT cultures was identical 45 min after plating (Fig. 3a). In spite of the same number of cells initially plated and attached to the substrate, the total surface occupied by DMSXL astrocytes was lower 45 min after plating, relative to control WT cultures (Fig. 3b). The lower confluence of DMSXL astrocytes persisted up to 48 h (Fig. 3c). In contrast, control DM20 and DM130 astrocyte cultures expressing shorter *DMPK* transcripts with a lower number CTG repeats[25] exhibited normal growth profiles over time (Fig. 3d, e), indicating that the reduced confluence of DMSXL astrocytes could not be accounted for by *DMPK* overexpression or the transgene integration site. Finally, we confirmed the defective spreading and abnormal morphogenesis of

DMSXL astrocytes, by measuring cell size over time and the growth rate of individual living cells, by semi-automated videomicroscopy (Fig. 3f). We found significantly reduced cell sizes, as well as a lower rate of cytoplasm spreading of primary DMSXL astrocytes, relative to WT controls (Fig. 3g). Pairwise comparisons revealed that the surface of DMSXL astrocytes was ~30–40% smaller throughout the 12 h period.

We used an immortalized glial cell line to validate the impact of toxic RNA on the adhesion and spreading of a human cell model of DM1. We used stably transfected MIO-M1 cells, which express a large CTG repeat expansion and accumulated abundant nuclear RNA foci, following the induction of a doxycycline-responsive promoter[29] (Supplementary Fig. 4a). We induced transgene expression for 24 h prior to replating. Like DMSXL astrocytes, for the same number of cells attached (Supplementary Fig. 4b), dox-induced MIO-M1 cells displayed reduced confluence 45 min after plating relative to non-induced controls (Supplementary Fig. 4c). We monitored cell spreading and morphology by live cell videomicroscopy (Supplementary Fig. 4d) and found that MIO-M1 cells expressing toxic CUG transcripts showed decreased cell spreading over a period of 12 h (Supplementary Fig. 4e). Pairwise comparisons revealed that the average cell sizes were reduced by ~35% in CUG-expressing cells at 9 and 12 h after plating.

In summary, DM1 repeat expansion affects the adhesion, spreading, and morphogenesis of primary mouse astrocytes and immortalized human glial cells.

**Defective adhesion and spreading of DMSXL astrocytes are associated with abnormalities in focal adhesion organization, cytoskeleton reorientation, and cell migration.** Cell-substrate adhesion relies on focal aggregates of specialized proteins that

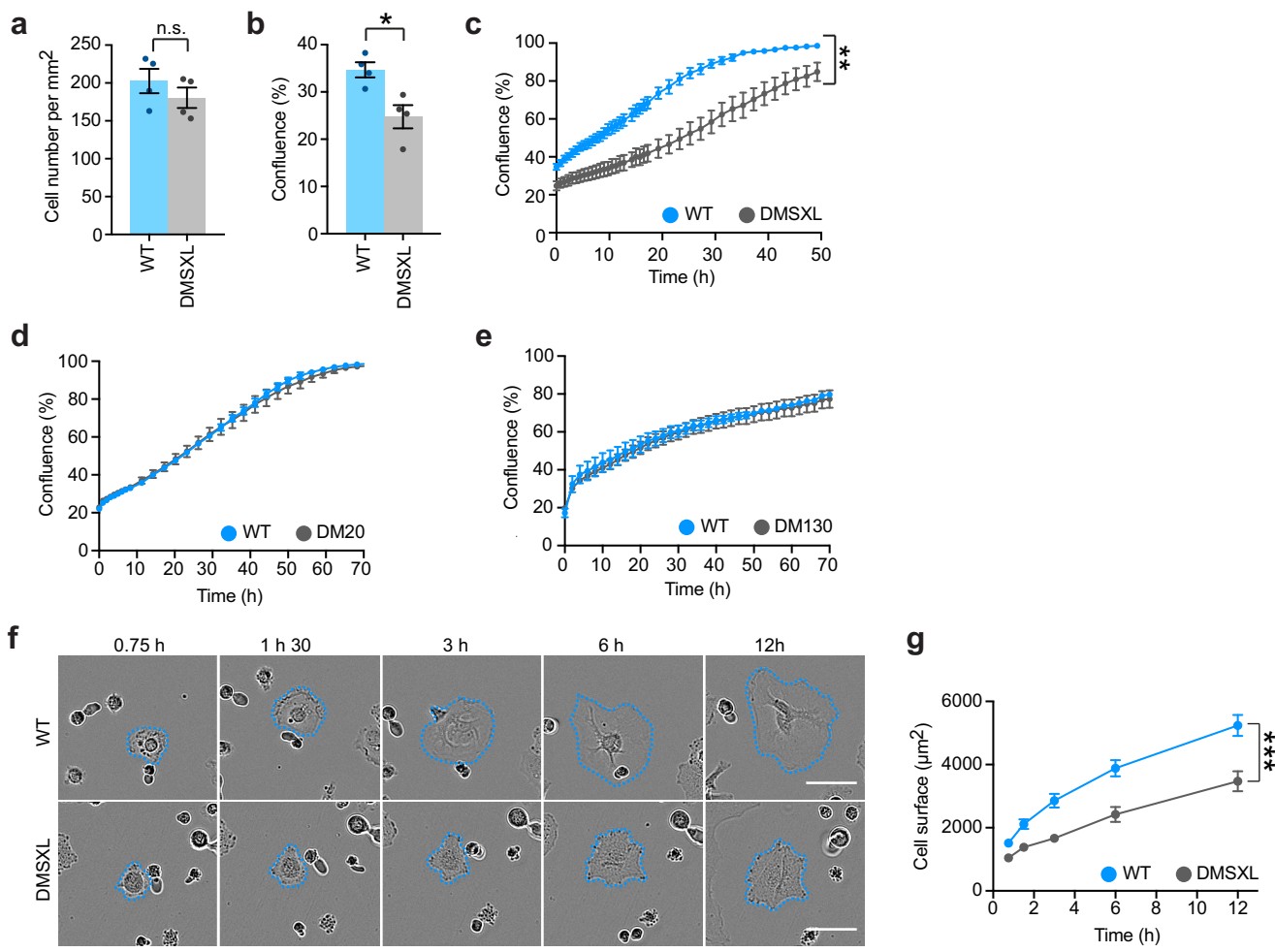

**Fig. 3 Primary DMSXL astrocytes exhibit abnormal adhesion and spreading. a** Quantification of the number of DMSXL and WT astrocytes attached 45 min after plating. Data are means ± SEM, $n = 4$ independent cultures per genotype ($p = 0.3429$, Mann–Whitney $U$ test). **b** Semi-automated quantification of cell culture confluence 45 min after plating. Data are means ± SEM, $n = 4$ independent cultures per genotype ($p = 0.0286$, Two-tailed Mann–Whitney $U$ test). **c** Videomicroscopy semi-automated monitoring of DMSXL and WT astrocyte confluence, from 45 min after plating up to 48 h in culture. Data are means ± SEM, $n = 4$ independent cultures per genotype ($p = 0.0048$, Two-way repeated measures ANOVA). **d** Quantification of the confluence of DM20 and WT astrocyte cultures over time. Data are means ± SEM, $n = 7$ independent cultures per genotype. **e** Quantification of the confluence of DM130 and WT astrocyte cultures over time. Data are means ± SEM; $n = 4$ WT independent cultures; $n = 3$ DM130 independent cultures. **f** Representative time-lapse bright field images of primary DMSXL and WT astrocytes, over 12 h after plating. Scale bar, 50 μm. The experiment was performed on three independent cultures of each genotype. **g** Quantification of the surface of individual primary astrocytes over time ($N = 3$ independent cultures per genotype; $n = 72$ cells, WT; $n = 64$ cells, DMSXL). Data are means ± SEM ($p < 0.001$, Two-way repeated measures ANOVA). Source data are provided as a Source Data file. n.s., not significant; *$p < 0.05$; **$p < 0.01$; ***$p < 0.001$.

serve as structural links between the cytoskeleton and the extracellular matrix. To gain insight into the subcellular structures contributing to the adhesion and spreading phenotypes of DMSXL astrocytes, we quantified focal adhesions in cultured cells. We stained clusters of vinculin, one of the first proteins to be recruited to assembling focal adhesion complexes, and that has an important role in maintaining focal contacts[30]. We used phalloidin to stain the actin cytoskeleton (Fig. 4a). The quantification of vinculin clusters revealed a significantly lower number of focal adhesions per cell in DMSXL astrocytes, relative to WT controls (Fig. 4b), in spite of similar expression levels of total vinculin (Fig. 4c), suggesting disrupted assembly and maintenance of focal adhesion structures. Using the actin fluorescent signal, we confirmed the reduced size of individual DMSXL astrocytes 3 h after plating (Fig. 4d). To investigate whether the adhesion defects in newborn primary DMSXL astrocytes likely reflect in vivo phenotypes, we plated mature cortical astrocytes acutely isolated from 1-month-old mice and investigated cell

spreading and focal adhesion organization. The analysis confirmed a significant reduction in cell size and number of focal adhesions per cell in DMSXL astrocytes (Fig. 4e). Together with reduced ramification of astrocytes in DMSXL brains, these data provide further evidence of the impact of expanded CUG RNA on cell adhesion and morphology.

Focal adhesions play an integral role in cell polarity and migration. Therefore, we asked if the reduced density of focal adhesions in DMSXL astrocytes affected their polarity and migration. One of the most frequently used culture systems to explore cell polarity and migration in culture is the scratch injury assay, during which cells reorient and extend towards the wound[31]. Prior to migration, the cell reorganizes the microtubule-organizing center (MTOC) and Golgi apparatus in front of the nucleus to face the free space, hence directing the membrane protrusion at the leading-edge perpendicular to the wound. We performed a wound healing assay to assess DMSXL astrocyte polarization and migration in culture. Following the

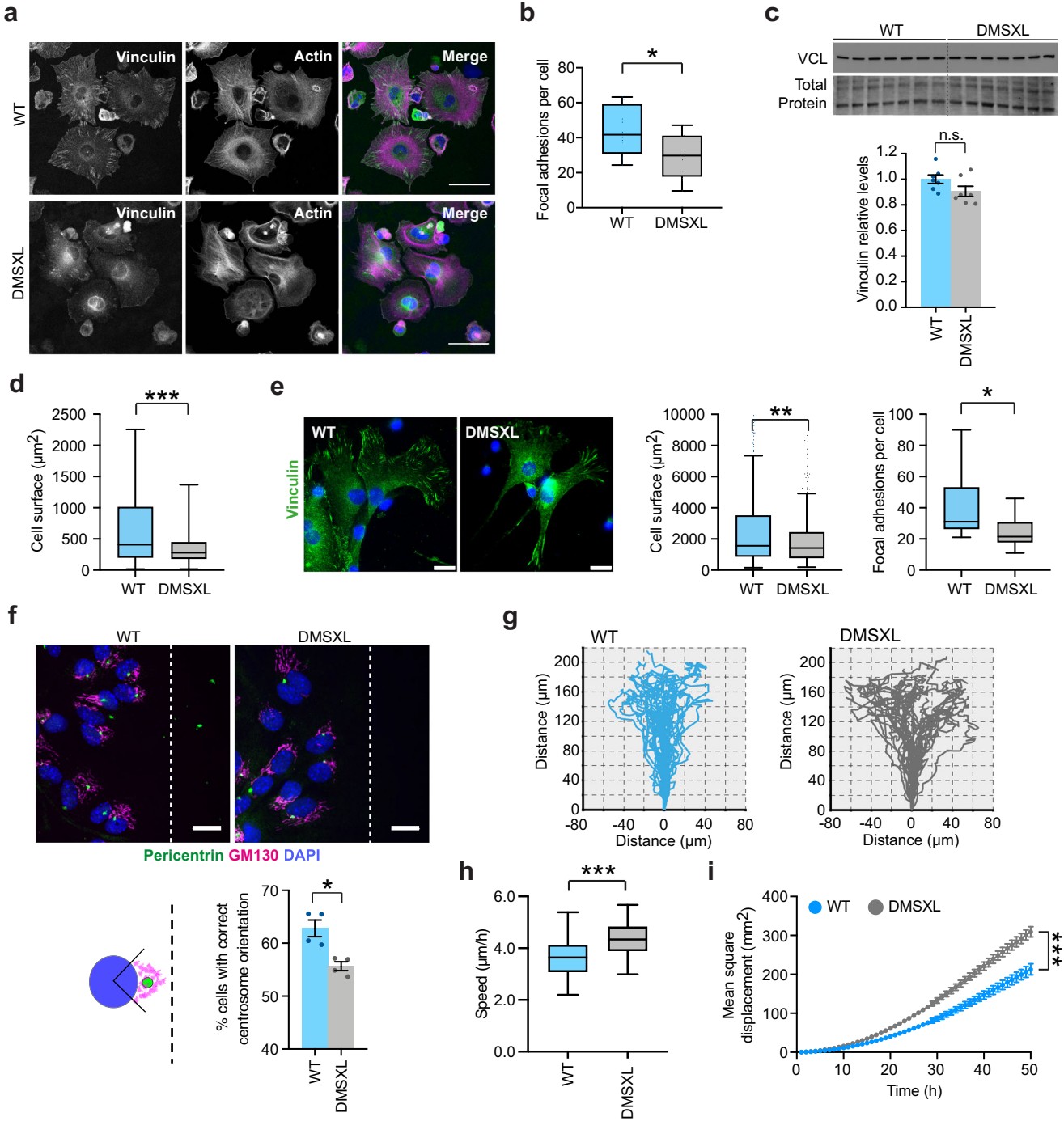

scratch, we monitored DMSXL astrocyte reorientation by immunofluorescence staining of the Golgi apparatus and centrosome and found a mild but significant defect in astrocyte polarization (Fig. 4f). Overall while 62.8 ± 1.6% of WT astrocytes orientated their Golgi apparatus and centrosome perpendicularly to the wound, only 55.7 ± 0.8% of DMSXL astrocytes were correctly orientated. Next, we examined the impact of defective assembly of focal adhesions and cell reorientation on cell migration, by time-lapse videomicroscopy and tracking of the trajectories of individual astrocytes (Fig. 4g). DMSXL astrocytes migrated at a higher speed over 50 h of wound closure (Fig. 4h) and displayed more random movement characterized by a higher mean square displacement (Fig. 4i), indicating that they covered a larger surface area over time until they filled the wound

completely, when compared to WT control astrocytes. The labeling of cell-substrate adhesions and the monitoring of cell movement revealed that the DM1 expansion mutation affects focal adhesion assembly, the re-orientation of the cytoskeleton, and the directionality of cell migration in DMSXL astrocytes.

Finally, we investigated whether gap junctions were affected by the expression of toxic CUG RNA. Gap junctions are specialized structures of direct intercellular communication that crosstalk with the cytoskeleton and focal adhesions, hence affecting cell morphology, adhesion, polarity, and migration[32]. Immunofluorescent detection of GJA1 (or connexin 43), the major connexin in cultured astrocytes[33], did not reveal overt changes in cell-to-cell contacts between primary DMSXL astrocytes (Supplementary Fig. 5a), and the total GJA1 protein levels were not altered (Supplementary Fig. 5b).

**Fig. 4 DMSXL astrocytes show defects in focal adhesion organization, cytoskeleton reorientation, and cell migration. a** Immunofluorescence of vinculin-rich focal adhesions and staining of actin cytoskeleton with phalloidin in DMSXL and WT astrocytes, following 3 h in culture. Scale bar, 50 μm. The experiment was performed on two independent cultures of each genotype. **b** Quantification of focal adhesions, assessed by the number of vinculin-rich clusters. Tukey whisker plots represent the number of focal adhesions per cell. $N = 2$ independent cultures per genotype; $n = 538$ cells, WT; $n = 609$ cells, DMSXL ($p = 0.0144$, Two-tailed Student's $t$ test). **c** Western blot quantification of vinculin expression in primary DMSXL and WT astrocytes. Representative stain-free protein bands are shown to illustrate total protein loading. Data are means ± SEM; $n = 7$ independent cultures per genotype ($p = 0.0934$, Two-tailed Mann–Whitney $U$ test). **d** Measurement of actin cytoskeleton spreading. Data are shown as Tukey whisker plots. $N = 2$ independent cultures per genotype; $n = 440$ cells, WT; $n = 487$ cells, DMSXL ($p < 0.0001$, Two-tailed Mann–Whitney $U$ test). **e** Immunofluorescence of vinculin and detection of focal adhesions in mouse cortical astrocytes acutely isolated at 1 month of age, 3 days after plating. Scale bar, 20 μm. Tukey whisker plots represent cell size and number of focal adhesions per cell. $N = 3$ independent cultures per genotype; $n = 450$ cells, WT; $n = 543$ cells, DMSXL ($p = 0.0043$, cell surface; $p = 0.0186$, focal adhesions per cell; Two-tailed Mann–Whitney $U$ test). **f** Analysis of MTOC and Golgi apparatus orientation through the immunofluorescent labeling of pericentrin and GM130, respectively, 8 h after wound-induced migration. Scale bar, 20 μm. Quantification of the percentage of cells with centrosome and Golgi orientated perpendicularly to wound. Data are means ± SEM. $N = 4$ independent cultures per genotype; $n = 2179$ cells, WT; $n = 2068$ cells, DMSXL ($p = 0.0286$, Two-tailed Student's $t$ test). **g** Representative migration tracking plots of individual DMSXL and WT astrocytes over 50 h, until complete wound closure. **h** Tukey whisker plots show cell speed of primary astrocytes during migration. $N = 2$ independent cultures per genotype; $n = 39$ cells, WT; $n = 39$ cells, DMSXL ($p < 0.0001$, Two-tailed Student's $t$ test). **i** Mean square displacement of primary astrocytes in culture over 50 h of cell migration, until complete wound closure. Data are means ± SEM. $N = 2$ independent cultures per genotype; $n = 39$ cells, WT; $n = 39$ cells, DMSXL ($p < 0.0001$, Two-way ANOVA). Tukey whisker plots display the median and extend from the 25th percentile up to the 75th percentiles. The whiskers are drawn down to the 10th percentile, and up to the 90th percentile. Source data are provided as a Source Data file. n.s. not significant; *$p < 0.05$; **$p < 0.01$; ***$p < 0.001$.

**DMSXL astrocytes enhance neuritogenesis defects of primary neurons**. Astrocytes can promote neurite growth and synapse formation[34], therefore we tested if DMSXL astrocytes affected neuritogenesis. We first monitored primary neuron neuritogenesis by time-lapse video-microscopy. Bright field tracking of neurite growth over longer periods of time (7 days in vitro, DIV) revealed a mild reduction in the growth rate of DMSXL neurites, which translated into significantly shorter neurites at later points, from 4.5 DIV onwards (Fig. 5a). The defect in neuritogenesis was confirmed by the analysis of transfected primary neurons, expressing mKate2 fluorescent protein under the Synapsin 1 promoter (Fig. 5b).

To investigate the influence of DMSXL astrocytes on neuritogenesis, we monitored neurite outgrowth of DMSXL and WT neurons co-cultured with astrocytes of either genotype. Fluorescent time-lapse videomicroscopy revealed a significant reduction of neurite outgrowth in DMSXL/DMSXL co-cultures, relative to WT/WT controls (Fig. 5c). More importantly, we found a negative impact of DMSXL astrocytes on the neurite development of both DMSXL and WT neurons, which resulted in significant lower neurite growth rates. Interestingly, DMSXL and WT neurons exhibited similar neurite growth rates in the presence of DMSXL astrocytes, corroborating the dominant effect of DMSXL astrocytes on neurite projection development.

Since astrocytes tightly interact and functionally regulate synapses[34], we investigated synaptic density in DMSXL mouse brains, as a first indication of potential disrupted synapse formation or maintenance. Immunolabeling of synapses in mouse frontal cortex revealed that the number of excitatory and inhibitory synapses remained unchanged in DMSXL, indicating similar total synaptic contacts (Fig. 5d and Supplementary Fig. 6a, b).

To conclude, the interplay of DMSXL astrocytes with neurons affects neuronal development and maturation in culture.

**DMSXL astrocytes show high expression of toxic RNA and abundant nuclear foci**. To assess the toxicity of CUG RNA repeats and investigate the molecular bases underlying the pronounced phenotypes of DMSXL astrocytes, we quantified the expression of expanded *DMPK* transcripts in primary cultures derived from homozygous DMSXL mice, after 6, 12, and 30 DIV. Transgene expression was significantly higher in DMSXL astrocytes, as *DMPK* expression levels were 4- to 12-fold higher in

astrocytes than in neurons (Fig. 6a, top panel). As expected, *DMPK* transgene expression profiles followed the endogenous *Dmpk* gene in both types of brain cells. Murine *Dmpk* gene expression was also significantly higher in astrocytes, relative to neurons (Fig. 6a, bottom panel). In agreement with higher transgene expression, FISH confirmed a higher percentage of DMSXL astrocytes showing RNA foci, and a higher number of foci per cell, when compared to DMSXL neurons (Fig. 6b and Supplementary Fig. 2c).

We investigated if CUG foci accumulation in primary DMSXL brain cells affected the canonical splicing regulators associated with DM1 RNA toxicity. We found marked cytoplasmic localization of MBNL1 and MBNL2 in primary brain cells using fluorescent immunodetection (Fig. 6c). While both MBNL1 and MBNL2 co-localized with RNA foci in DMSXL astrocytes, only MBNL2 displayed co-localization with RNA foci in neurons. We also found an intriguing reduction in the steady-state levels of MBNL1 and MNBL2 proteins in DMSXL astrocytes by western blot (Fig. 6d), which was confirmed with two independent primary antibodies (Supplementary Fig. 7) and equally detected in DMSXL frontal cortex and hippocampus (Supplementary Fig. 8). In contrast, MBNL protein levels in primary neurons were undistinguishable between genotypes and were significantly lower when compared to DMSXL astrocytes (Supplementary Fig. 7a, b). No obvious changes in CELF1 and CELF2 protein levels were found in DMSXL cells (Fig. 6d).

MBNL protein downregulation could not be attributed to lower transcript levels, since both *Mbnl1* and *Mbnl2* transcripts were surprisingly higher in DMSXL astrocytes (Fig. 6e), suggesting complex mechanisms of gene expression regulation. We studied the splicing of regulatory alternative exons of *Mbnl1* and *Mbnl2*[35] and found noticeably different splicing profiles between WT astrocytes and neurons. Furthermore, the alternative exons studied were abnormally spliced only in primary DMSXL astrocytes, remaining unaffected in DMSXL neurons, relative to WT control cells (Fig. 6f). Since these alternative exons regulate the nuclear localization of MBNL proteins, we studied the impact of missplicing on the intracellular distribution of MBNL1 and MBNL2 between nucleus and cytoplasm in DMSXL astrocytes. We found a significant increase in the nuclear to cytoplasmic ratio of MBNL2 immunofluorescence in DMSXL astrocytes relative to WT controls. MBNL1 distribution remained, however, unchanged between genotypes (Supplementary Fig. 7d).

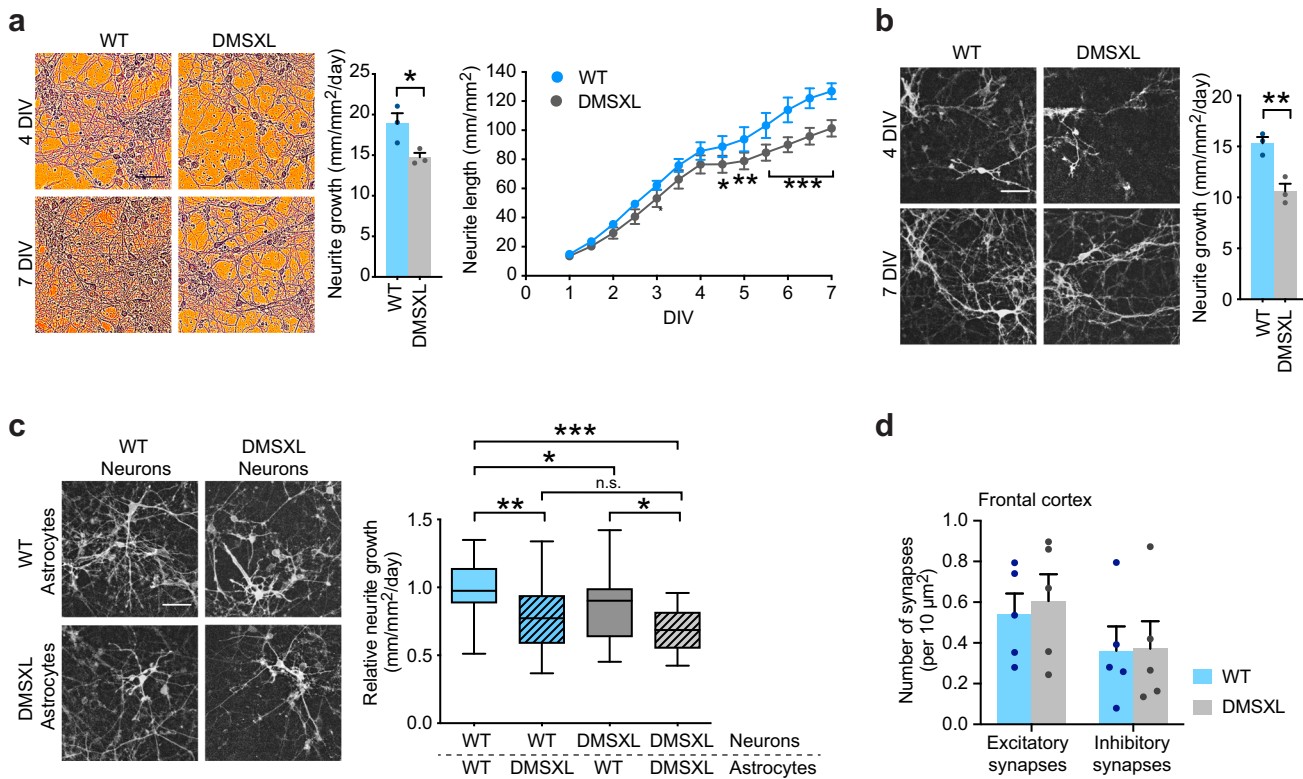

**Fig. 5 DMSXL astrocytes alter neuronal neuritogenesis. a** Representative brightfield images of unlabeled DMSXL and WT neurons at 4 and 7 DIV, showing neurite arborization in culture. Fire pseudocolor was applied on phase contrast images for clearer visualization of neurites. Scale bar, 50 μm. Semi-automated, label-free quantification of neurite growth rate and neurite length over 7 DIV. Data are means ± SEM. $n = 3$ independent cultures per genotype ($p = 0.0436$, Two-tailed Student's $t$ test, neurite growth; $p = 0.0103$, 4.5 DIV, $p = 0.0012$, 5.0 DIV, $p < 0.0001$, 5.5–7.0 DIV, Two-way ANOVA, Sidak post hoc test for multiple comparisons, neurite length). **b** Representative images of mKate2-expressing neurons in the same DMSXL and WT cultures at 4 and 7 DIV. Scale bar, 50 μm. Fluorescent signal was first detected at 4 DIV, and quantification of neurite growth by semi-automated time-lapse videomicroscopy was performed from 4 to 7 DIV. Data are means ± SEM. $n = 3$ independent cultures per genotype ($p = 0.0083$, Two-tailed Student's $t$ test). **c** Representative images of mKate2-expressing red fluorescent DMSXL and WT neurons co-cultured with unlabeled DMSXL and WT astrocytes. Scale bar, 50 μm. Quantification of the impact of astrocyte genotype on DMSXL and WT neurite growth. Tukey whisker plots represent neurite growth relative to normalized WT/WT co-cultures. $N = 4$ independent co-cultures per experiment, $n = 22$ neurons per condition ($p = 0.0032$, WT$_{neurons}$/WT$_{astrocytes}$ vs WT$_{neurons}$/DMSXL$_{astrocytes}$; $p = 0.0451$, DMSXL$_{neurons}$/WT$_{astrocytes}$ vs DMSXL$_{neurons}$/DMSXL$_{astrocytes}$; $p < 0.0001$, WT$_{neurons}$/WT$_{astrocytes}$ vs DMSXL$_{neurons}$/DMSXL$_{astrocytes}$; $p = 0.0124$, WT$_{neurons}$/WT$_{astrocytes}$ vs DMSXL$_{neurons}$/WT$_{astrocytes}$; $p = 0.1244$, WT$_{neurons}$/DMSXL$_{astrocytes}$ vs DMSXL$_{neurons}$/DMSXL$_{astrocytes}$; One-way ANOVA, Sidak post hoc test for multiple comparisons). The plots display the median and extend from the 25th percentile up to the 75th percentiles. The whiskers are drawn down to the 10th percentile, and up to the 90th percentile. **d** Number of excitatory synapses assessed by co-localized clusters of VGLUT1 and HOMER1, and inhibitory synapses assessed by co-localized clusters of GAD1/GAD2 (GAD65/GAD67) and GPHN (gephyrin) in the frontal cortex of DMSXL and WT mice, at 1 month of age. Data are means ± SEM. $N = 5$ mice per genotype, $n = 3$ fields per mouse. Source data are provided as a Source Data file. n.s. not significant; *$p < 0.05$; **$p < 0.01$; ***$p < 0.001$.

In summary, together with their pronounced phenotypes, primary DMSXL astrocytes expressed higher levels of expanded and toxic CUG-containing *DMPK* transcripts, exhibited more pronounced accumulation of toxic RNA foci, as well as MBNL protein sequestration and downregulation.

**DMSXL astrocytes exhibit significant splicing variations of genes involved in cell adhesion, cytoskeleton, and cell membrane dynamics**. To decipher the mechanisms behind abnormal adhesion and spreading of DMSXL astrocytes, we explored the RNA sequencing of primary cells that we have recently collected[36]. Transcripts from a total of 16,878 genes were detected in primary astrocytes derived from newborn mouse cortex (Supplementary Table 1). Stringent selection criteria revealed that the most severe expression changes affected only five transcripts in primary DMSXL astrocytes (Supplementary Table 2). Splicing changes were, however, much more frequent. We found evidence of missplicing in 128 astrocyte transcripts, including

abnormalities in exon skipping, splicing of mutually exclusive exons, splicing of multiple exons, as well as changes in the selection of acceptor and donor sites (Supplementary Table 1). The same thresholds revealed expression abnormalities in two genes, and splicing defects in 12 transcripts among the 17,089 transcripts identified in primary cortical DMSXL neurons (Supplementary Table 1).

To investigate the biological functions associated with the 128 transcripts misspliced in primary DMSXL astrocytes, we performed gene ontology (GO) enrichment analyses and found 69 biological processes, 68 cellular components, and 9 molecular functions significantly overrepresented. Post-processing methods identified cell adhesion, cytoskeleton, and plasma membrane as the low redundancy terms most frequently associated with enriched biological processes, cellular components, and molecular functions in primary DMSXL astrocytes (Fig. 7a). We then selected transcripts associated with representative GO terms for validation of abnormal exon skipping (Supplementary Table 3) and confirmed the significant missplicing of all the exons tested in

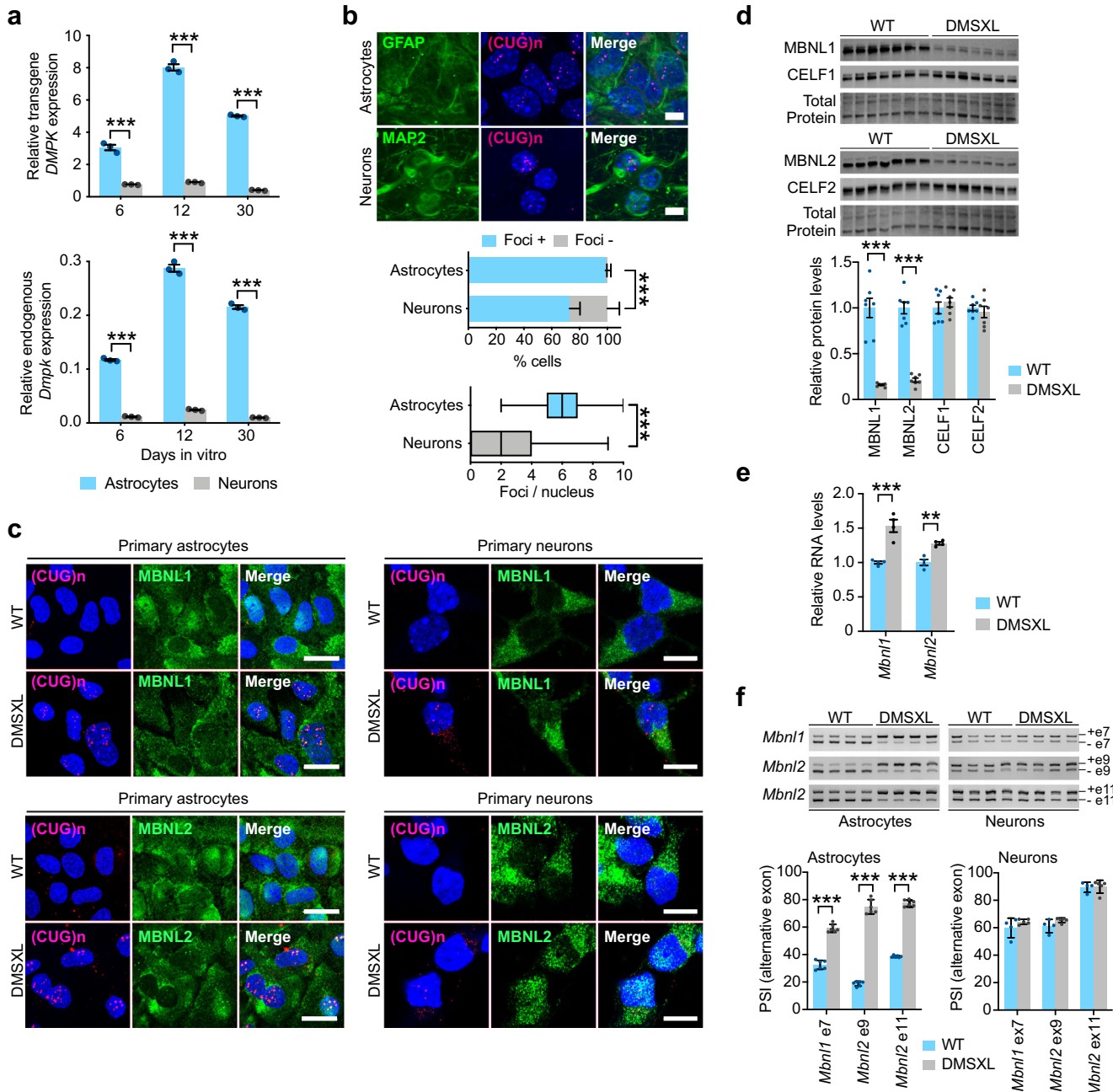

**Fig. 6 DM1 RNA toxicity is more pronounced in mouse astrocytes than in neurons. a** Quantitative RT-PCR expression analysis of human *DMPK* transgene and mouse endogenous *Dmpk* relative to *18S* rRNA internal control in primary DMSXL astrocytes and neurons at 6, 12, and 30 days in vitro. Data are means ± SEM, n = 3 independent cultures per group (*p* < 0.0001, Two-way ANOVA, Sidak post hoc pair-wise comparisons). **b** FISH analysis of nuclear foci accumulation in primary DMSXL astrocytes and neurons. Percentage of GFAP-expressing astrocytes and MAP2-expressing neurons containing foci (*p* < 0.0001, χ² test). Data are means ± SEM. Tukey whisker plots represent the number of foci per nucleus in primary DMSXL astrocytes and neurons (*p* < 0.0001, Two-tailed Mann–Whitney *U* test). The plots display the median and extend from the 25th percentile up to the 75th percentiles. The whiskers are drawn down to the 10th percentile, and up to the 90th percentile. *N* = 3 independent cultures per cell type; *n* = 217 astrocytes; *n* = 125 neurons. Scale bar, 10 μm. **c** FISH of RNA foci (magenta) combined with IF detection of MBNL1 and MBNL2 (green) in primary astrocytes and neurons of DMSXL and WT mice. Scale bar, 10 μm. Nuclei are stained with DAPI (blue). The experiment was performed on three independent cultures of each genotype. **d** Western blot quantification of MBNL and CELF proteins in primary astrocytes and neurons from DMSXL and WT mice. MBNL1 and MBNL2 were detected with antibodies raised against recombinant full length human proteins. Representative stain-free protein bands are shown to illustrate total protein loading. Data are means ± SEM, n = 7 independent cultures per genotype (*p* < 0.0001, MBNL1, MBNL2; *p* = 0.9199, CELF1, *p* = 0.9728; CELF2; Two-way ANOVA, Sidak post hoc test for multiple comparisons). **e** Quantification of *Mbnl1* and *Mbnl2* transcripts in primary DMSXL astrocytes and neurons, relative to WT controls. Data are means ± SEM, n = 4 independent cultures per genotype (*p* < 0.0001, *Mbnl1*; *p* = 0.0054, *Mbnl2*; Two-way ANOVA, Sidak post hoc pair-wise comparisons). **f** Representative RT-PCR splicing analysis of *Mbnl1* and *Mbnl2* transcripts in primary astrocytes and neurons. The graphs show the PSI of alternative exons. Data are means ± SEM, n = 6 independent cultures per genotype (*p* < 0.0001; Two-way ANOVA, Sidak post hoc test for multiple comparisons). Source data are provided as a Source Data file. **p* < 0.01; ***p* < 0.001.

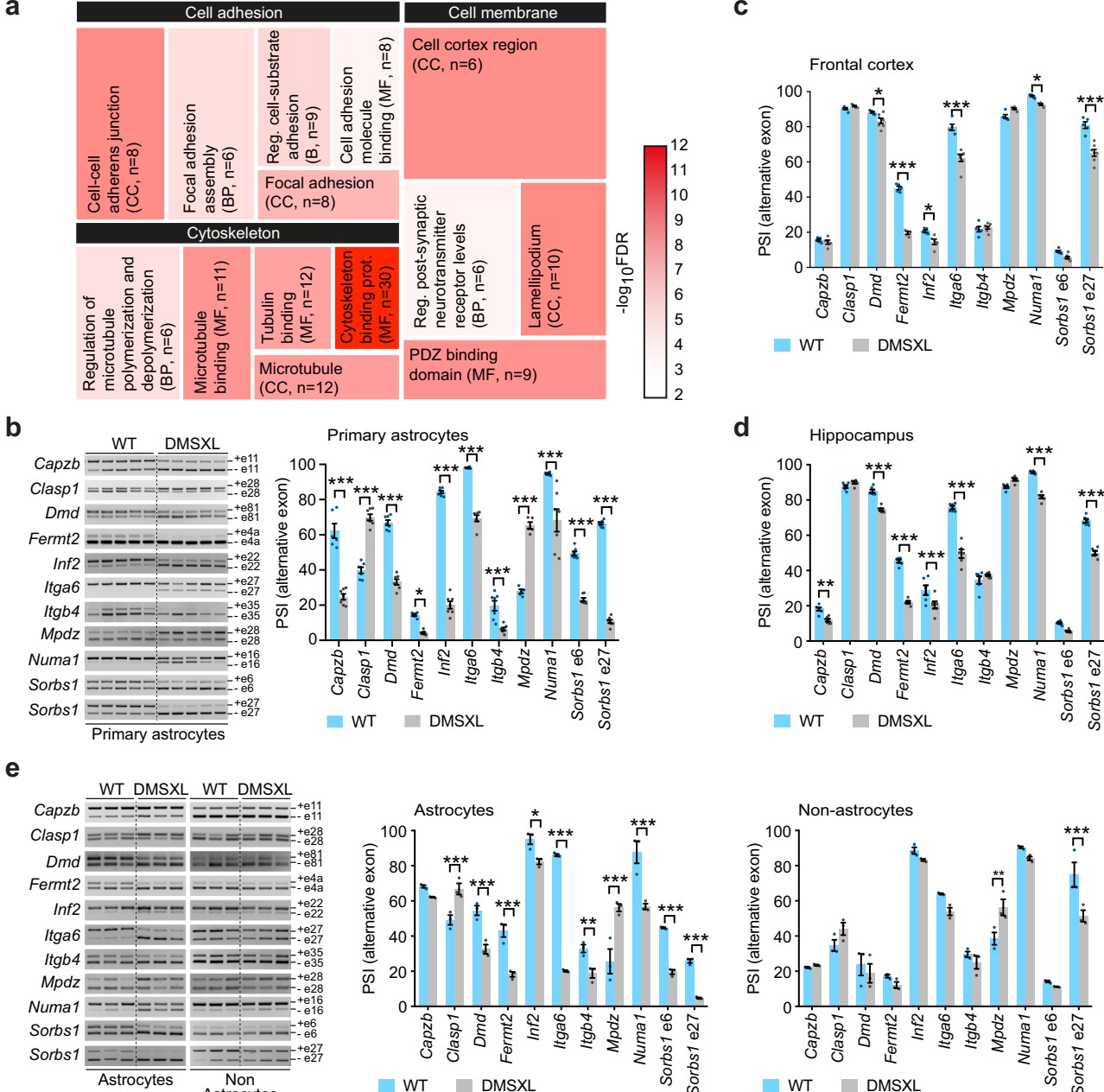

**Fig. 7 RNA sequencing of primary DMSXL astrocytes indicates defects in cell adhesion, cytoskeleton and cell membrane. a** Tree map representation of enriched non-redundant GO terms associated with the abnormally spliced genes in primary DMSXL astrocytes, and distributed by three main categories. Cell sizes are proportional to the enrichment ratio of each GO term. The heatmap scale represents the $-\log_{10}$(FDR) of enrichment. BP, biological process. CC, cellular component. MF, molecular function. The number of misspliced genes (n) in each category is shown. **b** Representative RT-PCR splicing analysis of selected transcripts that regulate cell adhesion and cytoskeleton in primary DMSXL and WT astrocytes. Alternative exons are indicated on the right. The graph represents the PSI of alternative exons. Data are means ± SEM, $n = 5$–7 independent cultures per genotype ($p < 0.0001$, *Capzb, Clasp1, Dmd, Inf2, Itga6, Mpdz, Numa1, Sorbs1* e6, *Sorbs1* e27; $p = 0.0122$, *Fermt2*; $p = 0.0001$, *Itgb4*; Two-way ANOVA, Sidak post hoc test for multiple comparisons). **c** Splicing dysregulation of adhesion- and cytoskeleton-associated transcripts in DMSXL frontal cortex. Data are means ± SEM, $n = 5$–7 mice per genotype ($p = 0.7944$, *Capzb, Itgb4*; $p = 0.7258$, *Clasp1*; $p = 0.0338$, *Dmd*; $p < 0.0001$, *Fermt2, Itga6, Sorbs1* e 27; $p = 0.0093$, *Inf2*; $p = 0.0859$, *Mpdz*; $p = 0.0405$, *Numa1*; $p = 0.1429$, *Sorbs1* e6; Two-way ANOVA, Sidak post hoc test for multiple comparisons). **d** Splicing dysregulation of adhesion- and cytoskeleton-associated transcripts in DMSXL hippocampus. Data are means ± SEM, $n = 6$ mice per genotype ($p = 0.0051$, *Cpazb*; $p = 0.8999$, *Clasp1*; $p < 0.0001$, *Dmd, Fermt2, Inf2, Itga6, Numa1, Sorbs1* e27; $p = 0.7844$, *Itgb4*; $p = 0.1630$, *Mpdz*; $p = 0.0916$, *Sorbs1* e6; Two-way ANOVA, Sidak post hoc test for multiple comparisons). **e** Representative RT-PCR splicing analysis in forebrain astrocytes and non-astrocyte cells isolated from 1-month-old mice. PSI of alternative exons in astrocytes and non-astrocyte cell fractions. Data are means ± SEM, $n = 3$ mice per genotype (Astrocytes: $p = 0.7686$, *Capzb*; $p = 0.0005$, *Clasp1*; $p < 0.0001$, *Dmd, Fermt2, Itga6, Mpdz, Numa1, Sorbs1* e6, *Sorbs1* e27; $p = 0.0244$, *Inf2*; $p = 0.0091$, *Itgb4*; Non-astrocytes: $p > 0.9999$, *Capzb*; $p = 0.3451$, *Clasp1*; $p = 0.9669$, *Dmd*; $p = 9623$, *Fermt2*; $p = 0.9472$, *Inf2*; $p = 0.2362$, *Itga6*; $p = 0.9826$, *Itgb4*; $p = 0.0024$, *Mpdz*; $p = 0.8625$, *Numa1*; $p = 0.9996$, *Sorbs1* e6; $p < 0.0001$, *Sorbs1* e27; Two-way ANOVA, Sidak post hoc test for multiple comparisons). Source data are provided as a Source Data file. *$p < 0.05$; **$p < 0.01$; ***$p < 0.001$.

primary DMSXL astrocytes (Fig. 7b). In contrast, the majority of these transcripts remained unaffected in primary DMSXL neurons: among the genes studied, modest splicing dysregulation was only detected in *Capzb* and *Itga6* transcripts by RT-PCR (Supplementary Fig. 9a).

We confirmed the spliceopathy of relevant transcripts associated with cell adhesion and cytoskeleton in vivo, through the analysis of DMSXL frontal cortex (Fig. 7c and Supplementary Fig. 9b) and hippocampus at 1 month of age (Fig. 7d and Supplementary Fig. 9c). Among all the eleven alternative exons studied, five were significantly dysregulated in DMSXL frontal cortex, while seven were abnormally spliced in DMSXL hippocampus. In contrast to DMSXL, control DM20 and DM130 mice did not show splicing defects in transcripts that were markedly affected in DMSXL mouse brain tissue (Supplementary Fig. 9d).

To gain insight into the onset and progression of splicing dysregulation throughout brain development and ageing, we studied some selected exons at postnatal day 10 and at 4 months. While the absence of defective astrocyte ramification at P10 (Supplementary Fig. 1a), was associated with the nearly complete absence of splicing abnormalities (only *Itga6* was significantly misspliced at this early age) (Supplementary Fig. 8e), we found persistent spliceopathy at 4 months of age (Supplementary Fig. 9f).

The heterogeneity and intermixing of diverse cell types in complex tissues, such as the brain, may dilute severe missplicing events confined to specific cell populations, and it complicates the attribution of the splicing abnormalities to individual cell types. We specifically investigated splicing dysregulation in forebrain astrocytes in vivo, isolated from 1-month-old DMSXL mice with antibody-coupled magnetic beads. The enrichment of the astrocyte cell fraction was confirmed by the semi-quantitative expression analysis of *Gfap* transcripts mainly expressed in astrocytes, while the non-astrocyte fraction showed high expression of neuron-specific *Syp* (synaptophysin) and oligodendrocyte-specific *Gpr17* (G-coupled protein receptor 17) RNA (Supplementary Fig. 9g). We then compared the splicing of adhesion- and cytoskeleton-associated transcripts between the astrocyte-enriched fraction and the non-astrocyte flow-through (consisting of neurons, oligodendroglia, and endothelial cells). The severe spliceopathy in DMSXL forebrain astrocytes in vivo was demonstrated by the significant missplicing of ten out of eleven alternative exons studied, while only two were misspliced in non-astrocyte cells (Fig. 7e).

The splicing analysis in forebrain astrocytes isolated from DMSXL brains revealed pronounced splicing defects that were undetected in whole brain tissue samples, indicating localized RNA toxicity in astroglia. However, astrocytes are a diverse population of cells displaying brain area-specific properties and functions[34], driven by intrinsic molecular programs and context cues provided by neighboring cells[37,38]. We investigated whether CUG RNA toxicity impacted the transcriptomic regulation of astrocytes in a region-specific manner, through the analysis of alternative splicing in mouse astrocytes acutely isolated from frontal cortex, hippocampus, and cerebellum (Supplementary Fig. 10). While *Dmd*, *Itga6*, *Mpdz*, and *Numa1* exhibited widespread missplicing, other transcripts showed region-specific defects, suggesting regional susceptibility of astrocyte subpopulations to toxic CUG RNA. *Sorbs1* exon 6 was only significantly affected in frontal cortex astrocytes, *Fermt2* abnormalities were detected in both frontal cortex and hippocampus astrocytes, and *Capzb* was only significantly misspliced in DMSXL cerebellum.

In summary, we found compelling evidence of pronounced splicing dysregulation in DMSXL astrocytes of genes involved in cell adhesion, cytoskeleton organization, and membrane dynamics, in agreement with the cellular phenotypes.

**MBNL1 and MBNL2 proteins mediate the cell phenotypes and spliceopathy of DMSXL astrocytes.** Given the pronounced reduction in the steady-state levels of MBNL1 and MBNL2 proteins, together with their co-localization with nuclear RNA foci in primary DMSXL astrocytes, we asked if the inactivation of MBNL proteins was sufficient to perturb astrocyte cell growth and adhesion. To answer this question, we used siRNA strategies to knockdown MBNL1 and MBNL2 in primary WT astrocytes (Fig. 8a), as previously described[22], and found significantly lower cell confluence (Fig. 8b) and reduced number of focal adhesions per cell (Fig. 8c), when compared to scrambled siRNA controls. To gain insight into the role of MBNL proteins in the splicing regulation of transcripts affected in primary DMSXL astrocytes, we studied the splicing profiles in MBNL double knockdown astrocytes. Overall, reduced MBNL1 and MBNL2 protein levels recreated the splicing abnormalities of primary DMSXL astrocytes, except for *Clasp1* exon 28, which showed a modest change following siRNA treatment (Fig. 8d).

In conclusion, MBNL proteins regulate the splicing of cell adhesion- and cytoskeleton-related transcripts in primary astrocytes, and their inactivation impacts cell growth and adhesion in culture.

**Human DM1 brains show pronounced foci accumulation in astrocytes and splicing dysregulation of adhesion and cytoskeleton-related transcripts.** To determine if the insight gained from DMSXL astrocytes is relevant for human disease, we examined post-mortem human frontal cortex and hippocampus. We first investigated if the higher abundance of nuclear RNA foci in cortical astrocytes relative to neighboring neurons observed in DMSXL mouse brains[14] was also detected in human disease brains. FISH detection of expanded *DMPK* transcripts in human DM1 frontal cortex confirmed a higher frequency of foci and a higher number of foci per nucleus in astrocytes, relative to neurons (Fig. 9a, b; Supplementary Fig. 11). In spite of inter-individual variability, DM1 patients showed reduced expression of GFAP in frontal cortex (Fig. 9c). We investigated if the target exons perturbed in DMSXL astrocytes were similarly dysregulated in DM1 brains. Overall, most exons studied were significantly misspliced in adult DM1 brain samples, compared to non-DM controls: eight out of eleven exons were affected in frontal cortex (*ITGB4* exon 35 and *MPDZ* exon 28 showed a clear trend but did not reach statistical significance) (Fig. 9d), whereas all exons studied were misspliced in the hippocampus of DM1 patients (Fig. 9e).

Like in DMSXL mice, the brains of DM1 patients exhibited marked RNA foci accumulation in astrocytes and missplicing of transcripts relevant for cytoskeleton, cell adhesion, and morphology in two relevant brain regions: the frontal cortex and hippocampus.

## Discussion

The research on DM1 brain disease was grounded on the idea that repeat-induced missplicing in neurons was the dominant event at the origin of neuropsychiatric symptoms[13,39,40]. Previous attempts to unveil glial involvement in human DM1 brains yielded inconclusive results[7]. Using a mouse model of DM1, we previously reported Bergmann glia abnormalities in the cerebellum, which affected the activity of Purkinje cells[22]. Here, we uncovered the wider vulnerability of astrocytes to CUG RNA toxicity, beyond the involvement of Bergmann glia. We showed that the DM1 repeat expansion impacts critical structural and functional features of astrocytes, such as cell morphology and orientation in the mouse brain, as well as adhesion in cell culture models of the disease.

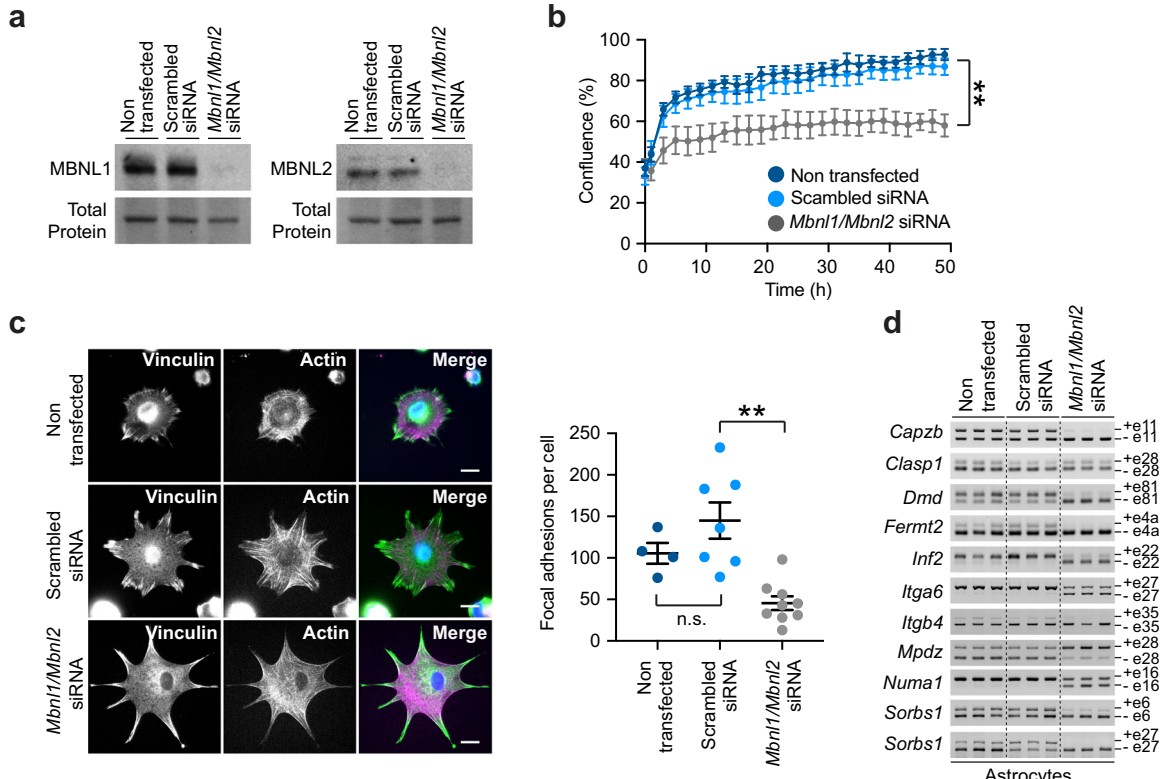

**Fig. 8 MBNL1 and MBNL2 inactivation affects astrocyte cell growth, adhesion, and alternative splicing. a** Western blot detection of MBNL1 and MBNL2 in double knockdown primary astrocytes, relative to scramble siRNA and non-treated controls. Representative stain-free protein bands are shown to illustrate total protein loading. The experiment was performed on two independent cultures per treatment. **b** Monitoring of the confluence of primary astrocytes by live cell video-miscroscopy, from 45 min after plating up to 48 h. Primary MBNL1/MBNL2 knockdown astrocytes are compared with non-transfected and scrambled-transfected controls. Data are means ± SEM, n = 4 independent cultures per genotype (p = 0.0089, Two-way repeated measures ANOVA). **c** Immunofluorescent quantification of vinculin-rich focal adhesions per individual cell in siRNA-treated primary astrocytes, following 3 h in culture. Scale bar, 20μm. Data are means ± SEM. N = 2 independent cultures per cell group; n = 4 cells, non-transfected; n = 7 cells, scrambled siRNA; n = 9 cells, Mbnl1/Mbnl2 siRNA (p > 0.9999, non-transfected vs scrambled siRNA; p = 0.0023, scrambled siRNA vs Mbnl1/Mbnl2 siRNA; Kruskal–Wallis test, Dunn's post hoc test for multiple comparisons). **d** RT-PCR splicing analysis of adhesion- and cytoskeleton-associated transcripts in Mbnl1/Mbnl2 double knockdown astrocytes, compared with non-transfected and scrambled siRNA controls. Alternative exons are shown on the right. The experiment was performed on three independent cultures per treatment. Source data are provided as a Source Data file. n.s. not significant; **p < 0.01.

Pronounced RNA toxicity in astrocytes is accompanied by defects in cell adhesion and spreading. Transgene expression in astrocytes follows the profile of endogenous *Dmpk* mouse gene, decreasing during cell differentiation like the human *DMPK* gene in brain[15] and skeletal muscle[41], and demonstrating that the DMSXL transgene contains important elements for the regulation of its own expression.

Primary DMSXL astrocytes express higher levels of *DMPK* transgene and exhibit greater RNA foci abundance than DMSXL neurons, likely contributing to the more severe spliceopathy and overt phenotypes of astroglia. Previous comparisons between DM1 mouse models revealed that higher expression of toxic CUG transcripts is associated with greater foci content and more severe muscle spliceopathy and pathology[42,43]. In addition to higher *DMPK* expression, DMSXL astrocytes display the highest *DMPK/Mbnl1* and *DMPK/Mbnl2* ratios (relative to other brain cell types), which provide conditions conducive to MBNL protein sequestration as previously suggested[36,40]. In agreement with defective cell spreading and reduced size growth of primary DMSXL astrocytes, RNA sequencing uncovered a significant number of misspliced transcripts associated with cell adhesion, cytoskeleton, and cell membrane, which we confirmed in vivo in astrocytes acutely isolated from adult DMSXL mice, relative to the remaining brain cells.

Defective DMSXL astrocyte branching in vivo emerges between postnatal day 10 and 1 month, a time window corresponding to the developmental switch from embryonic and adult splicing isoforms. We recently reported that DMSXL astrocytes show increased expression of splicing isoforms typical of immature astroglia[36]. These results corroborate the view that DM1 perturbs primarily the splicing of developmentally regulated transcripts[44], and suggest that astrocyte phenotypes result mainly from the postnatal impairment to express adult splicing isoforms.

MBNL protein inactivation in astrocytes is a determinant event behind the coordinated dysregulation of gene sets associated with cytoskeleton and cell adhesion. The lower activity of MBNL in DMSXL astrocytes results from the combined co-localization with RNA foci and the surprising reduction in total protein. MBNL protein downregulation was previously reported in proliferating DM1 myoblasts in the presence of unaltered transcript levels[45], through mechanisms that have not been fully elucidated. The regulation of *Mbnl1* and *Mbnl2* gene expression is complex and multifaceted, involving alternative transcription and translation initiation sites, splicing events, miRNA, and circRNA species[35]. It is tempting to speculate that changes in *Mbnl1/Mbnl2* transcription initiation and/or exon 1 skipping may shift translation initiation to produce unstable protein isoforms, as previously reported in DM1 skeletal muscle[46]. Alternatively, miRNA-

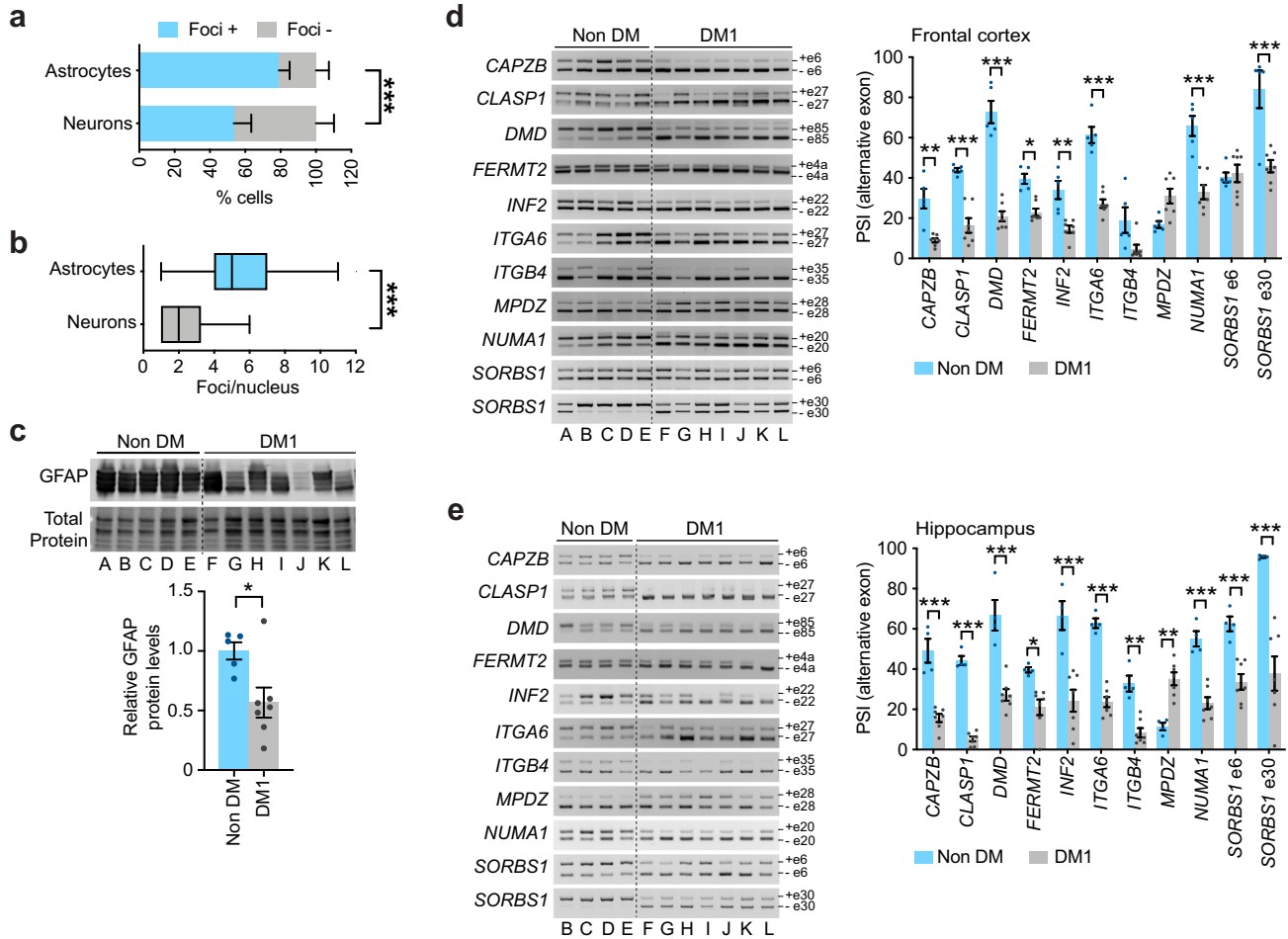

**Fig. 9 Pronounced foci accumulation in astrocytes and splicing dysregulation of adhesion and cytoskeleton-related transcripts in human DM1 brains.**
**a** Percentage of GFAP-positive astrocytes and MAP-positive neurons exhibiting nuclear RNA foci in the frontal cortex of DM1 patients ($p < 0.0001$, $\chi^2$ test). Data are means ± SEM, $N = 3$ individuals, $n = 152$ astrocytes, $n = 101$ astrocytes. **b** Tukey whisker plots representing the number of foci per nucleus in human cortical astrocytes and neurons ($p < 0.0001$, Two-tailed Mann–Whitney $U$ test). The plots display the median and extend from the 25th percentile up to the 75th percentiles. The whiskers are drawn down to the 10th percentile, and up to the 90th percentile. $N = 3$ DM1 patients; $n = 284$, astrocytes; $n = 202$, neurons. **c** Western blot quantification of GFAP in tissue lysates prepared from post-mortem human frontal cortex. Representative stain-free protein bands are shown to illustrate total protein loading. Data are means ± SEM. $n = 5$, non-DM; $n = 7$, DM1 ($p = 0.0480$, Two-tailed Mann–Whitnet $U$ test). **d** RT-PCR splicing analysis of adhesion- and cytoskeleton-associated transcripts in the frontal cortex of adult DM1 patients and non-DM subjects. Alternative exons are indicated on the right. Quantification of PSI of alternative exons. Data are means ± SEM. $n = 5$, non-DM; $n = 7$, DM1 ($p = 0.0017$, *CAPZB*; $p < 0.0001$, *CLASP1*, *DMD*, *ITGA6*, NUMA1, SORBS1 e30; $p = 0.0233$, FERMT2; $p = 0.0040$, INF2; $p = 0.0788$, ITGB4; $p = 0.0919$, MPDZ; $p > 0.9999$, SORBS1 e6; Two-way ANOVA, Sidak post hoc test for multiple comparisons). **e** RT-PCR splicing analysis in the hippocampus of adult DM1 patients and non-DM subjects. Alternative exons are indicated on the right. Quantification of PSI of alternative exons. Data are means ± SEM. $n = 4$, non-DM; $n = 7$, DM1 ($p < 0.0001$, *CAPZB*, *CLASP1*, *DMD*, *INF2*, *ITGA6*, NUMA1, SORBS1 e30; $p = 0.0403$, FERMT2; $p = 0.0019$, ITGB4; $p = 0.0030$, MPDZ; $p = 0.0001$, SORBS1 e6; Two-way ANOVA, Sidak post hoc test for multiple comparisons). Source data are provided as a Source Data file. *$p < 0.05$; **$p < 0.01$; ***$p < 0.001$.

mediated translation suppression in DMSXL astrocytes may result in MBNL1 and MBNL2 protein downregulation[47]. In both cases, lower protein levels would not be accompanied by a corresponding reduction in *Mbnl1* and *Mbnl2* transcripts.

The abnormal splicing of *Itga6* and *Itgb4* in DMSXL astrocytes may perturb focal adhesion assembly and subsequent cell spreading. Exon 27 of *ITGA6* indeed encodes a cytoplasmic protein domain involved in the transmission of biochemical and mechanical signals from the extracellular matrix to the cytoskeleton[48]. A role of MBNL proteins in the regulation of integrin-mediated adhesion was previously described in *Drosophila*[49] and in human cells[50]. In addition, MBNL1 targets have also proposed roles in cytoskeletal rearrangements and post-mitotic cell growth[51,52]. Therefore, we suggest that MBNL

sequestration and partial inactivation by RNA foci affects the adhesion, spreading, and growth of multiple cell types in various tissues, representing an integral feature of disease pathogenesis.

Integrins and other cell adhesion molecules at neuroglial contacts are instrumental for the synaptic coverage by the astrocyte fine processes, and the control of synaptic transmission and plasticity. Defects in integrin expression and localization have been linked to abnormal social and stress behavior, reduced spatial and working memory, and to autism spectrum disorder[53], neuropsychological features that have been reported in DM1 patients to different extents[6]. Importantly, abnormal focal adhesion signaling in the CNS causes neuronal hyperexcitability and seizure activity in *Drosophila*[54], and it may contribute to the increased sensitivity to induced seizures of DMSXL mice[55].

The cytoskeleton plays an instrumental role in the establishment of intrinsic cell polarity. The perturbation of microtubule dynamics, through changes in the activity of microtubule-binding proteins, impairs proper cell polarization[56]. The missplicing of a significant number of transcripts coding for tubulin-binding proteins and regulators of microtubule dynamics in DMSXL astrocytes may therefore contribute to the misorientation of the microtubule-organizing center (MTOC) and the erratic migration during the wound healing assay. Changes in the activity of CLASP1, for example, were shown to cause random cell movement in culture, similar to those detected in DMSXL astrocytes[57]; SORBS1 regulates cell adhesion, spreading, and migration[58]; and NUMA1 controls cell polarity, and the dynamics of microtubules[59]. The uncontrolled cell movement and increased speed of DMSXL astrocytes, may also result from the low number of stable focal adhesions, which under normal conditions tend to inhibit cell migration[60]. Defects in brain cell migration have been associated with multiple neurological disorders, characterized by abnormal cortical function, cognitive and motor impairment, as well as epilepsy[61].

The complex morphology of astrocytes is a defining feature of their function, and it is controlled by extensive cytoskeleton remodeling. Mature astrocytes grow numerous thin processes that reach out to occupy individual, non-overlapping spatial domains in the brain. Through their sophisticated ramification, astrocytes provide extensive coverage of neighboring synapses to guarantee efficient neurotransmitter clearance, ion homeostasis, metabolic support, and gliotransmitter availability to neurons[34]. At birth, astrocyte processes contain primarily microtubules, which are later replaced by GFAP-containing intermediate filaments[62]. The missplicing of microtubule-associated genes, together with the lower levels of GFAP may contribute to the reduced arborization of cortical astrocytes in DMSXL brains. Since GFAP is constitutively expressed in the main processes of white matter astrocytes[63], GFAP downregulation is indicative of localized astrocyte hypotrophy in white matter.

As the most abundant glial cell type in the mammalian brain, astrocytes are integral to the regulation of brain wiring and function. In light of their role in the formation, maintenance, and control of synapses, it is not surprising that their dysfunction is an important contributor to many neurological diseases, including Alzheimer, Parkinson and Huntington disease, amyotrophic lateral sclerosis, spinal muscular atrophy, epilepsy, and fragile X syndrome[64]. In most conditions, astrocyte dysfunction is accompanied by changes in the expression of ion channels, neurotransmitter transporters, membrane receptors, or intracellular proteins[65]. Astrocyte gap junctions also control neuronal transmission and behavior[66]. However, we did not find evidence of defective gap junction assembly or abnormal expression of GJA1 in DMSXL astrocytes.

Alternatively, since astrocyte-neuron communication is intimately dependent on the specialized morphology of both cell types, aberrant astrocyte ramifications are predicted to be detrimental to neurons. In this context, we showed that DMSXL astrocytes delay neurite growth in culture, but the overall synaptic density is not altered in vivo. Appropriate astrocyte reorientation is critical for synaptic maturation[26]. In particular, postnatal structural rearrangements increase the proximity between parallel astrocyte processes and synaptic terminals, to enhance synaptic coverage and facilitate ion buffering, the release of neurotrophic factors, and the uptake of neurotransmitters by astrocytes[67]. We propose that impaired astrocyte arborization and polarity in DMSXL mice (comparable to that found in other mouse models of astrocyte dysfunction[24,68]), may affect the ultrastructure and maturation of the synapse in vivo, in terms of astroglial coverage, synapse-to-astrocyte distance, and dendritic arborization. In this context, it will be important to investigate defects in the ultrastructural and functional interplay between astrocytes and neurons, which may contribute to the aberrant tonic currents and synaptic plasticity of DMSXL mice[69].

Neuroimaging studies have suggested early white matter involvement in DM1, whereas age-dependent cortical gray matter loss may occur later in the disease process in association with neurodegeneration[7]. Early astrocyte atrophy with decreased GFAP staining has been described in some mouse models of Alzheimer Disease[70] and in the prefrontal cortex of young patients suffering from major depressive disorder[65]. It has been proposed that astrocyte atrophy in the frontal cortex impacts the capacity of astroglia to maintain brain homeostasis, contributing to the expression of depression-like phenotypes, such as anhedonia[71], a behavioral feature of DMSXL mice[14] and DM1 patients[72]. Recent studies have also found evidence of demyelination in DM1 white matter lesions, particularly in the anterior temporal lobe[73]. The relative role of myelinating oligodendrocytes in DM1 brain pathology in the brain warrants further investigation.

In conclusion, our results demonstrate the pronounced impact of DM1 toxic RNA on astrocyte cell biology. Altered adhesion of astrocytes in DM1 could affect their migration, orientation, and integration into the complex microenvironment of the brain, hence impairing neural wiring and synaptogenesis. Future therapeutic strategies targeting the CNS must consider the restoring of astrocyte-mediated homeostasis, in conjunction with neuronal rescue and repair strategies.

## Methods

**Transgenic mice**. All experimental procedures and the handling of mice were conducted in accordance with the European Union Directive 86/609/EEC on the protection of animals, and in compliance with the ARRIVE guidelines (Animal Research: Reporting In Vivo Experiments), in order to reduce animal suffering and minimize the number of animals. Experimental protocols were approved by Paris Descartes Ethics Committee for Animal Experimentation (CEEA 34) and Charles Darwin Ethics Committee for Animal Experimentation (CEEA 5), under the agreements number 201511101421606 and 20190701014259655, issued by the French Ministry of Higher Education and Research. Animals were maintained a 12 h light/12 h dark cycle with food and water provided ad libitum. Temperature and humidity were within the recommended range (20–24° and 40–60%, respectively).

DMSXL and DM130 transgenic mice (>99% C57BL/6 background) carry a 45 kb fragment of human genomic DNA from a DM1 patient[18]. The DMSXL mice used in this study carry more than 1500 CTG repeats within the *DMPK* transgene. DM130 mice result from a spontaneous trinucleotide repeat contraction, and carry a short ~130-CTG tract in the same integration site. Control DM20 mice carry a similar transgene with a non-expanded 20 CTG repeat sequence, derived from a healthy individual[25]. Transgenic DM20 status was assessed as previously reported[14]. DMSXL status was assessed by multiplex PCR using the oligonucleotide primers, which hybridize the *DMPK* transgene and the transgene integration site in the mouse *Fbxl7* gene: FBF (forward, TCCTCAGAAGCACTCA TCCG), FBFBR (reverse, AACCCTGTATTTGACCCCAG) and FBWDR (reverse, ACCTCCATCCTTTCAGCACC). FBF and FBFBR primers hybridize the mouse *Fbxl7* gene, amplifying a sequence of 167 bp in WT mice. FBWDR primer hybridizes specifically the human transgene, generating a 236 bp PCR product.

**Human tissue samples**. Human frontal cortex samples were collected from different laboratories: Dr. Yasuhiro Suzuki (Asahikawa Medical Center, Japan) and Dr. Tohru Matsuura (Okayama University, Japan). All experiments using human samples were approved by the Ethics Committees of the host institutions. Written informed consent of specimen use for research was obtained from all patients. No payments of any form have been given to the subjects or their families. Information relative to patients was previously described[14,22].

**Analysis of astrocyte morphology in vivo**. Complex astrocyte morphology in vivo was studied as previously described[68]. Briefly, adult mice were deeply anaesthetized by a mixture of ketamine and xylazine and placed in a stereotaxic frame. AAV expressing a cytoplasmic GFP protein under the GFAP promoter (AAV2/9-GFAP-GFP) was delivered into the mouse frontal cortex with a 34-gauge blunt-tipped needle linked to a 2 μL Hamilton syringe by a polyethylene catheter at the following stereotaxic coordinates: AP: −0.9 mm; L: −1 mm to the bregma at a depth of −2.75 mm to the skull. The needle was left in place for 5 min and was

slowly withdrawn. Animals were sacrificed 15 days after injection. Newborn mice were cryoanesthetized and subsequently placed on a cold metal plate. The injection site was marked with a non-toxic laboratory pen. A 30-gauge needle was used to pierce the skull just anterior to bregma and 1 mm lateral to the midline line, and 0.5 μl of AAV2/9-GFAP-GFP was injected. Neonatal mice were kept with their mother until the sacrifice at P10. Z-stack images were acquired at a confocal microscope (Leica SP5 inverted confocal), using a 40×/1.0NA objective, and Leica Application Suite X software. Isolated astrocytes were selected based on their GFP staining. The ImageJ plugin "Sholl analysis"[74] was used to measure the number of intersections between GFP stained processes and concentric circles spaced by 5 μm and centered on astrocyte nucleus, visualized by DAPI staining.

**SDS PAGE and western blot detection.** Proteins from primary cells, whole mouse frontal cortex and hippocampus, and human frontal cortex collected at post-mortem were extracted using RIPA buffer (Thermo Scientific; 89901) supplemented with 0.05% CHAPS (Sigma; C3023), 1× complete protease inhibitor (Sigma-Aldrich; 04693124001) and 1× PhosSTOP phosphatase inhibitor (Sigma; 04906845001). Protein concentrations were determined using the Pierce BCA Protein Assay Kit (Thermo Scientific; 23227). Between 10 and 40 μg proteins were mixed with 2X Laemmli Sample Buffer (Sigma; S3401), denatured for 5 min at 95 °C, and resolved in 10% TGX Stain-Free polyacrylamide gels (Bio-Rad; 1610183). After electrophoresis, gels were activated for 2 min under UV light, proteins were transferred onto Nitrocellulose membranes using Trans-Blot® Transfer System (Bio-Rad) and total protein on the membrane was imaged using the ChemiDoc Imaging System (Bio-Rad) as used for protein normalization[75]. Membranes were then blocked in 2.5–5% Blotto non-fat dry milk (Santa Cruz Biotech; sc2325) in 1× TBS-T (10 mM Tris-HCl, 0.15 M NaCl, 0.05% Tween 20) during 1 h at room temperature (RT) and incubated with the primary antibody over night at 4 °C. After three washes in TBS-T membranes were incubated with IRDye® 800CW donkey anti-rabbit (LI- COR Biosciences; P/N 926-32213) or 680RD donkey anti-mouse (LI-COR Biosciences; P/N 926-68072) for 1 h at RT, washed three times and imaged using LI-COR Odyssey ® CLx Imaging System. Band intensity was quantified using Image Studio Lite. Primary antibodies are indicated in Supplementary Table 4. Normalized protein expression levels are expressed relative to the average of control samples.

**Capillary electrophoresis immunoassay (Simple Western).** Protein levels of GFAP, GJA1, MBNL1, and MBNL2 were analyzed using a Simple Western JESS™ capillary-based electrophoresis instrument (ProteinSimple). Total protein 12–230 kDa Separation Module 8 × 25 capillary cartridges compatible with protein normalization assay (ProteinSimple, AM-PN01) were used following the manufacturer's instructions. Optimal antibody dilutions and protein concentrations were determined in pilot experiments for each assay. Primary antibodies were diluted in Antibody Diluent 2 (ProteinSimple, 042-203), while protein samples were diluted in 0.1× sample buffer (ProteinSimple, 042-195) and denatured in the presence of Fluorescent Master Mix (ProteinSimple, PS-ST01EZ-8) for 5 min at 95 °C before loading the plate. Primary antibodies used are detailed in Supplementary Table 4. Anti-mouse (ProteinSimple, DM-002) or anti-rabbit (ProteinSimple, DM-001) HRP-labeled secondary antibodies were used according to the species of the primary antibody. Assay separation time was 25 min of GFAP and GJA1, and 30 min for MBNL1 and MBNL2. Replex (ProteinSimple, RP-001) and total protein detection module for chemiluminescence-based total protein assay (ProteinSimple, DM-TP01) were used for the normalization of the signal obtained in each plate. Data were analyzed using Compass for Simple Western software v4.0.0. The resulting area under the curve (AUC) of each specific peak was divided by the total protein signal obtained in the same capillary and represented as the relative % to the average in WT samples.

**Primary astrocytes cell culture.** Primary dissociated cell cultures of cortical astrocytes were prepared from postnatal day 1 WT and DMSXL mouse littermates. After carefully removing the meninges in Leibovitz L- 15 Medium (Life Technologies; 11415049) supplemented with 30 μM Glucose (Sigma-Aldrich; 49139), the cortices were mechanically dissociated and cultured for 2 weeks in DMEM low glucose (Life Technologies, 31885-023), supplemented with 10% FBS and 0.05 mg/ml gentamycin (Life Technologies; 15710). All experiments were performed using primary astrocytes cultured for 2 weeks, unless stated otherwise. Independent cultures (biological replicates) were established from different animals. Corresponding WT control cultures were established from the same litters as DMSXL cultures.

**Primary neurons cell culture.** Primary dissociated cell cultures of cortical neurons were prepared from embryonic day 16.5 WT and DMSXL littermates. After carefully removing the meninges in Glucose-supplemented Leibovitz L-15 medium, the cortices were washed twice with Neuronal medium: Neurobasal- A (Life Technologies; 10888022), supplemented with 1X B27 supplement (Life Technologies; 17504044), 0.5 mM L-Glutamine (Life Technologies; 25030024), 1% antibiotic and antimycotic (Life Technologies; 15240-096). Cortices were then incubated 15 min in Neuronal medium containing Trypsin (Life Technologies; 25300096) and DNAseI (Sigma-Aldrich; 11284932001) and washed with Neuronal

medium supplemented with 5% FBS. After mechanical dissociation, cells were washed twice and counted before plating in suitable dishes coated with Poly-D-lysine (Sigma-Aldrich; P6407) and Laminin (Sigma- Aldrich; L2020) in Neuronal medium containing 5% FBS. Medium was changed 4 h after plating to Neuronal medium supplemented with 5 μM AraC (Sigma-Aldrich; C6645-25MG) and without FBS to avoid astrocytes proliferation. Half of the medium was changed each 3–4 days.

**MIO-M1 cell line.** The generation and maintenance of the MIO-M1 cell model has been described elsewhere[29]. Briefly, MIO-M1 cells were cultured in DMEM supplemented with 10% FBS, 100 U/mL penicillin, and 100 μg/mL streptomycin. Transgene expression was induced by 1 μg/mL doxycycline added to the culture medium for 3 days prior to analysis.

**Acute isolation of astrocytes from mouse brain.** Brains from 1-month-old mice were dissected. Forebrain, frontal cortex, hippocampus, and cerebellum tissue samples were enzymatically and mechanically dissociated using the adult brain dissociation kit (Miltenyi; 130-107-677), following the manufacturer's protocol. The astrocytes were isolated from the resulting single cell suspension using magnetic anti-ACSA-2 labeled microbeads (Miltenyi; 130-097-678) for positive selection, following the manufacturer's protocol. The cells in the ACSA-2 positive cell fraction (retained in the column) and ACSA-2 negative cell fraction (flow through) were counted and a cell pellet was collected by centrifugation at 300 g for 5 min. The sequences of the oligonucleotide primers used in the RT-PCR analysis of astrocyte enrichment are listed in Supplementary Table 5.

**Impedance-based real-time monitoring of cell adhesion and growth.** Cell population adhesion and growth was monitored using the xCELLigence RTCA MP system and software (Agilent), which measures the electrical impedance of the cells in real time, translated into a cell index proportional to the number, surface, or adhesion strength of the plated cells. Primary neurons (200.000 cells/well) were plated onto Poly-D-lysin and Laminin coated E-plates (Ozyme; 5232368001) while primary astrocytes (20,000 cells/well) were plated on uncoated E-plates. After a 30 min settling of the cells at RT, impedance was measured each 5 min during the first 8 h and each 15 min for the following 72 h, inside the incubator.

**FACS analysis of astrocytes cell cycle.** Primary astrocytes were arrested in G0 by completely removing the serum from the culture medium for 24 h. After cell cycle release in medium containing 5% of FBS, cells were incubated with 10 μM Bromodeoxyuridine (BrdU) (Euromedex; NU-122S) for 1 h, trypsinized, fixed O/N in 70% ethanol at −20 °C, and stained with antiBrdU-FITC (BD Horizon; 347583; dilution 1:500 per $10^6$ cells) and 3 μM propidium iodide (PI) (Sigma; P4170), before FACS detection of BrdU-FITC and PI in 10,000 cells per embryo. Three different biological replicates were analyzed per genotype. Cell cycle phases were considered as follows: G1-phase cells are BrdU negative with low PI signal (200 intensity units), S-phase cells are BrdU positive and show a range of low to high PI intensity (200 to 400 intensity units) and G2/M cells are BrdU negative with high PI intensity (400 intensity units). Cell cycle phases were analyzed with FlowJo software (BD Biosciences).

**FACS analysis of astrocytes death.** Cell death was analyzed in astrocytes at 3 days post plating after O/N incubation with DMSO (solvent control) or 0.5 μM Staurosporine (Euromedex; LS9300-A). Cells were trypsinized, counted and $10^6$ cells were stained with a mix of 5% Annexin V-FITC (BD Horizon; 556547) to detect apoptotic cells, PI to detect necrotic cells and 2.5% Cd11b-V450 (BD Horizon; 560456) to exclude possible microglia contamination. Staining was performed for 15 min, prior to FACS analysis of 10,000 cells per embryo. Six different biological replicates were analyzed per genotype. The percentage of AnnexinV-positive, PI-positive WT and DMSXL astrocytes (negative for Cd11b staining) was analyzed in DMSO and Staurosporine conditions, using FlowJo software (BD Biosciences).

**Videomicroscopy monitoring of cells dynamics.** Primary astrocyte adhesion and spreading was monitored using the IncuCyte Zoom video-microscope (Essen BioScience) after seeding 20,000 cells/well in 96-wells plates and taking phase-contrast pictures every 45 min. Images were acquired with a 20×/0.45NA objective and the IncuCyte base analysis software. Cell confluence was determined using the IncuCyte ZOOM basic analysis method. Cell surface was measured by manual tracing. Cell number was counted manually using Image J. Experiments were performed in triplicate. Astrocytes migration was assessed using the Cell Migration Assay kit and migration module of the IncuCyte followed by manual tracking of the astrocytes using Fiji Manual Tracking plugin and quantitative analysis of speed and mean square displacement, using DiPer macro on Microsoft Office Excel, as previously described[76].

**Immunofluorescence.** Cultured cells were fixed in 4% PFA for 15 min at room temperature, washed 3 × 10 min in 1X PBS and incubated in blocking and permeabilization solution for 1 h at room temperature (1× PBS; 0.2% Triton X-100;

10% normal goat serum, Sigma, G6767). Primary antibody was diluted in blocking and permeabilization and incubated over night at 4 °C. Primary antibodies and dilutions are indicated in Supplementary Table 4. Following 3 × 10 min washes in 1× PBS, cells were incubated with secondary antibody diluted in the blocking and permeabilization solution for 1 h at room temperature. Excess antibody was washed 3 × 10 min in 1× PBS, and incubated with 0.0002% DAPI (Sigma, 10236276001) for 15 min at room temperature. Stained cells were finally washed 3 × 10 min in 1× PBS and mounted with Vectashield® Mounting Medium (Eurobio Scientific, H-100) prior to observation. Images were acquired with a laser scanning confocal microscope (SP8, Leica), using a 20×/0.8NA or 60×/1.4NA objective and Leica Application Suite X software.

**Fluorescent in situ hybridization (FISH) and immunofluorescence.** Primary cells cultured for 2 weeks were washed 3 × 2 min in 1× PBS and then fixed for 15 min in 4% PFA (VWR; J61899.AP) at room temperature. Cryostat tissue sections (10 μm) were directly fixed in 4% PFA. Following 5 × 2 min washes in 1× PBS, cells and tissues were permeabilized in 2% ice-cold acetone in 1× PBS for 5 min at room temperature. Prehybridization was carried out in 30% formamide/2× SSC for 10 min at room temperature. Hybridization with 1 ng/mL 5′-Cy3-labeled (CAG)₅ PNA probe in hybridization buffer (30% formamide, 2× SSC, 0.02%BSA, 66 mg/ml yeast tRNA, 2 mM vanadyl complex) was carried out for 2 h at 37 °C in a dark humidified chamber. Next, cells/tissues were washed for 30 min in 30% formamide/2× SSC at 50 °C, followed by one last wash of an additional 30 min in 1× SSC at room temperature. Prior to visualization, cell nuclei were stained with 0.0002% DAPI (Sigma, 10236276001) for 15 min at room temperature. Stained samples were finally washed 3 × 10 min in 1× PBS and mounted with Vectashield® Mounting Medium (Eurobio Scientific, H-100) prior to observation. RNA foci were counted in 3D stacks using the Spot Detector plugin of the ICY bioimageanalysis open-source program (http://icy.bioimageanalysis.org). Immunofluorescence (IF) combined with fluorescent in situ hybridization (FISH) was performed as previously described[14]. When combined with immunofluorescence, cell, and tissue samples were directly incubated in primary antibody following the 1× SSC post-hybridization wash. Primary antibodies and dilutions are indicated in Supplementary Table 4. Washes and secondary antibody incubation were performed as described above. Images were acquired with a laser scanning confocal microscope (SP8, Leica), using 60×/1.4NA objective and Leica Application Suite X software.

**MBNL1 and MBNL2 nucleo-cytoplasmic distribution.** Following MBNL1 or MBNL2 immunofluorescence, the mean fluorescence intensity ratio of nuclear and cytoplasmic protein was quantified on confocal images, using the "Intensity Ratio Nuclei Cytoplasm Tool" plugin on Fiji Software (Intensity Ratio Nuclei Cytoplasm Tool, RRID:SCR_018573). In short, the tool first used a thresholding method in the nucleus channel for nucleus segmentation. The background intensity was then adjusted and the nuclear signal was removed to measure MBNL1 and MBNL2 signal intensity in the cytoplasm channel.

**Focal adhesion detection and quantification.** Focal adhesions formation was quantified as previously described[77]. In short, astrocytes were PFA-fixed 3 h post plating and stained with anti-vinculin antibodies, to count the number of focal adhesions. DAPI staining was used to determine the number of cells. Images were taken as Z stacks with a Leica SP8 Confocal Microscope, using 60×/1.4NA objective and Leica Application Suite X software (1024 × 1024; line average = 2; constant PMT parameters). Vinculin-rich clusters were detected after background subtraction, using rolling ball method, gamma adjustment of 1.4. Particles with a greater intensity than the auto threshold, and with an area between 0.1 and 40 μm2 considered to be focal adhesions. The 40 μm² size allowed to exclude perinuclear signal and staining artefacts.

**Imaging of neurite growth in primary neurons and neuron-astrocyte co-cultures.** Primary neurons were plated at 90,000 cells/cm² on 96-well plates and maintained for 7 DIV. Images acquired every 3 h for a total of 7 DIV, using a 20×/0.45NA objective and the IncuCyte base analysis software Semiautomated, label-free monitoring of neuritogenesis in primary neurons growing alone was performed using the Neurotrack module of the IncuCyte Zoom video-microscope on phase contrast images, as previously described[28]. Phase masks were used to contour the cell bodies and to trace neurites. The length of neurites was automatically measured and plotted overtime. Co-cultures of neurons and astrocytes were established as previously described[22]. Briefly, primary neurons were plated at 90,000 cells/cm² and infected 4 h after plating (MOI = 3) with NeuroLight Red Lentivirus (Essen BioScience, 4584), encoding the mKate2 fluorescent protein, expressed under the Synapsin 1 promoter, in serum-free Neuronal medium. The following day neurons were washed with Neuronal medium. Mouse primary astrocytes, cultured 2 weeks prior to neurons, were plated on top at a density of 150,000 cells/cm². Fluorescent images of co-cultures were acquired every 3 h for a total of 7 DIV, using a 20×/0.45NA objective and the IncuCyte base analysis software. mKate2 expression was detected 4 days after transfection. NeuroTrack analysis module of the IncuCyte Zoom video-microscope was used to quantify neurite length and arborization from 4 DIV onwards.

**Immunodetection and quantification of cortical synapses in mouse brain.** Immunodetection of synapses in mouse frontal cortex was performed as previously described[78]. Briefly, 40 μm mouse brain slices were blocked with PBS-Gelatine-Triton X-100 (1× PBS, 0.002% gelatine, 0.25% Triton X-100) for 1 h at room temperature, incubated 48 h at 4 °C with primary antibodies (Supplementary Table 4) in blocking solution, washed three times with PBS, incubated overnight with secondary antibodies in blocking solution and then washed three times with PBS, before mounting with Abberior mounting medium.

Images were acquired with a confocal microscope (Abberior Instruments GmbH). The microscope is based on a Scientifica microscope body (Slice Scope, Scientifica) equipped with an Olympus 100×/1.4 ULSAPO objective lens. Images were acquired with the Imspector Software (Abberior Instruments GmbH). Excitatory synapses were identified by co-localization clusters of pre-synaptic VGLUT1 (SLC17A7) and post-synaptic -HOMER1. Inhibitory synapses were identified by co-localization clusters of pre-synaptic GAD1/GA2 (GAD65/GAD67) and post-synaptic GPHN (gephyrin). Images of stained mouse brain slices were acquired with a confocal laser-scanning microscope (Abberior Instruments GmbH). The analysis of pre/post-synaptics proteins was performed using an in-house developed plugin on Fiji[79]. Briefly, pre and post dot channels were segmented using StarDist libraries[80] using an in-house trained model on dots. Then synapses were defined as center point between two pre and post dots having at least one co-localized pixel.

**RNA isolation, cDNA synthesis, and RT-PCR analysis.** RNA extraction was performed with the RNeasy Mini kit (QIAGEN; 74104) following the manufacturer's protocol, including a DNAse digestion step (RNase-Free DNase Set; QIAGEN; 79254) after the first wash with RW1 buffer. RNA concentration was assessed using the NanoDrop (Thermo Scientific) and RNA quality was verified by electrophoresis on an agarose gel. cDNA synthesis and semi-quantitative reverse-transcriptase PCR analysis of alternative splicing were performed as previously described[14,22]. Oligonucleotide primers used for RT-PCR analysis are indicated in Supplementary Table 6 (mouse samples) and Supplementary Table 7 (human samples).

**Quantitative RT-PCR.** Human *DMPK*, murine *Dmpk,* and *18S* internal control transcripts were quantified in a 7300 Real Time PCR System (Applied Biosystems) with a Power SybrGreen detection method (Thermo Scientific; 4367659) using oligonucleotide primers and conditions previously described[19]. Samples were quantified in triplicate and experiments were repeated at least twice. *18S rRNA* was used as internal control. *DMPK*, *Dmpk,* and *18S* transcript levels were interpolated from a linear range of serial diluted cDNA standards. The *DMPK/18S* and *Dmpk/18S* expression ratios were normalized using the same standard sample.

**RNA sequencing.** The RNA sequencing data used in this study were previously collected, published, and made available in public databases[36]. RNA samples were prepared from cortical WT and DMSXL primary astrocytes, grown for 14 days in vitro. Illumina-compatible precapture barcoded mRNA libraries were constructed, and a series of 24 barcoded libraries was pooled at equimolar concentrations. The capture process was performed according to the manufacturer's protocols for TruSeq Stranded mRNA (Illumina) and sequencing on an Illumina HiSeq2500. Global gene expression was analyzed as previously described[81] using the DESeq2 package. Significant expression changes were considered for further analysis if fold change between genotypes >1.4, and *p* value (corrected for multiple comparisons) <0.05. Alternative splicing analyses were performed using the open source FARLINE pipeline (http://kissplice.prabi.fr/pipeline_ks_farline/) as previously described[82,83]. Significant missplicing events were selected for further analysis if the percentage of splicing inclusion between genotypes >10 and *p* value (corrected for multiple comparisons) <0.05 (general linear mode). The percentage of splicing inclusion (PSI) was calculated as (inclusion of alternative exon)/ [(inclusion of alternative exon) + (exclusion of alternative exon)].

**Gene ontology analysis.** Gene ontology (GO) analyses were performed using Webgestalt tool[84] to identify significant functional enrichment in three GO categories: biological processes, cellular component, and molecular function. FDR < 0.01 and gene number ≥2 were defined as selection criteria for further analyses. Post-processing weight set cover methods were used to summarize and reduce the redundancy, through the identification of the most representative GO terms.

**Statistical analysis.** Statistical analyses were performed with Prism (GraphPad Software, Inc). Data are presented as mean ± standard error of means (±SEM). Individual datapoints are shown in graphs, except if it compromises clarity. Tukey box-and-whiskers plots are used to represent large numbers of replicates ($n > 10$) to improve clarity. Tukey whisker plots display the median and extend from the 25th percentile up to the 75th percentiles. The whiskers are drawn down to the 10th percentile, and up to the 90th percentile. Chi-square test was used to compare categorical variables between two groups. After performing a normality test on the numeric variables, we used two-tailed Student's *t* test for parametric data and Mann–Whitney *U* test for non-parametric data, when two groups were compared. When three or more groups were compared, we performed a one-way ANOVA or Repeated-measures ANOVA on parametric data, or Kruskal–Wallis test on non-

parametric data. We used two-way ANOVA to assess the statistical interaction between two independent variables. Post hoc pairwise comparisons were performed to account for multiple comparisons. $*p < 0.05$; $**p < 0.01$; $***p < 0.001$.

**Reporting summary**. Further information on research design is available in the Nature Research Reporting Summary linked to this article.

## Data availability

Source data are provided as a Source Data file. Uncropped and unprocessed images of all the gels and blots shown in the manuscript are provided as Supplementary Information. All the RNA sequencing data sets that were used as input for the study are available at GEO (Gene Expression Omnibus). GEO accession: GSE162093. Source data are provided with this paper.

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

## Acknowledgements

We are grateful to the personnel of CERFE (Center d'Exploration et de Recherche Fonctionelle Expérimentale, Genopole, Evry, France) and LEAT (Laboratoire d'Expérimentation Animale, Imagine Institute, Paris, France) for attentively caring for the mice. We thank the Cell Imaging Platform of the Structure Fédérative de Recherche Necker for the useful help in microscopy acquisitions and analyses, and the support provided by the PSMN (Pôle Scientifique de Modélisation Numérique) of the ENS de Lyon for computing resources. We would like to thank Ruben Artero for providing access to equipment and reagents. This study was supported by grants from Agence Nationale de Recherche (ANR, France, Project Grant DM_Neuroglia to G.G., N.R., D.A., and C.M.), AFM-Téléthon (France, Project Grant 19920 to M.G.-P.), Project European Research Council (Consolidator Grant #683154 to N.R.), European Union's Horizon 2020 research and innovation program (Marie Sklodowska-Curie Innovative Training Networks, grant #722053, EU-GliaPhD to N.R.), Région Ile de France (France), CONACyT (Mexico, Project Grant CB-2012-01/183697 to O.H.H.), INSERM (France), Université Paris Descartes (France), as well as PhD fellowships from Ministère Français de la Recherche et Technologie (France, to D.M.D., G.S., and L.-E.P.), AFM-Téléthon (France, to D.M.D. and G.S.). L.L. was awarded a PhD fellowship from Fondation pour la Recherche Médicale (France, PLP201910009933).

## Author contributions

Conceptualization, D.M.D., C.M., D.A., N.R., C.F.B., G.G., and M.G.-P.; methodology, D.M.D., L.L., G.S., C.M., C.N.A.-V., L.E.A-J, O.H.-H., D.A., G.G., and M.G.-P.; validation, L.L., A.G.-B., S.O.B., and J.T.-B.; formal analysis, D.M.D., L.L., A.G.-B., N.C., L.-E.P., H.P., C.F.B., and M.G.-P.; investigation, D.M.D., L.L., A.G.-B., N.C., S.O.B., G.S., L-E.P., P.M., A.H.-L., H.B., C.N.A.-V., L.E.A-J, J.T.-B., and M.G.-P.; resources, O.H.-H., C.M., D.A., N.R., C.F.B., G.G., and M.G.P.; writing – original draft, D.M.D., and M.G.-P.; writing – review & editing, D.M.D., L.L., A.G.-B., N.C., C.M., D.A., N.R., C.F.B., G.G., and M.G.-P.; visualization, D.M.D., L.L., A.G.-B., N.C., L.E.-P., O.H.-H., and M.G.-P.; supervision, D.A., N.R., C.F.B., G.G., and M.G.-P.; project administration, M.G.-P.; funding acquisition, C.M., D.A., N.R., C.F.B., G.G., and M.G.-P. All authors read and approved the final manuscript.

## Competing interests

The authors declare no competing interests.
