## [Peer Review File · Nature Communications]

Reviewers' Comments:

Reviewer #1:

Remarks to the Author:

Myotonic dystrophy RNA toxicity alters morphology, adhesion and migration of mouse and human astrocytes.
Dincă et al.

Myotonic dystrophy type 1 (DM1) is caused by expansion of the CTG trinucleotide in the 3' untranslated region of the DMPK gene. RNAs containing expanded CUG triplets accumulate in the nucleus of DM1 cells, forming RNA foci that perturb the localization and function of RNA-binding proteins. Using a transgenic mouse model of DM1, the authors investigate the effects of DM1 on CNS cells. Interestingly, they report the most significant effects on astrocytes, the major non-neuronal cell type in the brain. Astrocytes exhibit impaired ramification and polarization *in vivo* and defects in adhesion, spreading and migration *in vitro*. Transcriptome analysis revealed mis-splicing of transcripts regulating adhesion, cytoskeleton and morphogenesis. The authors report these changes were also seen in human DM1 cells. Hence, the authors conclude that DM1 impacts astrocyte biology, presumably compromising the cells ability to support and regulate synaptic function.

The idea that astrocytes play a major (if not the dominant) role in CNS disease is an interesting idea and, if true, would represent a paradigm shift in neuroscience. However, in my opinion, there are several issues which raise question marks over the suitability of the work for publication at this time.

Major issues:

The bulk of the work reported in this manuscript is performed using cultured astrocytes produced using the "classical method". Astrocytes cultured in this manner morphologically resemble fibroblasts with a flat morphology and show high rates of growth. This is in contrast to the star-shaped cells found *in vivo*, which show (comparatively) low rates of growth (Clavreul et al., Nat Commun, 2019). Furthermore, sequencing studies show large differences between cultured cells and acutely isolated astrocytes (Foo et al., Neuron, 2011). Finally, astrocytes *in vivo* do not migrate in response to injury (Tsai et al., Science, 2012), in direct contrast to the scratch assay data shown in the paper. In my mind, the question which needs to be asked is to what degree "classical" astrocyte cultures represent the *in vivo* situation (Foo et al., Neuron, 2011) and whether cultures of DM1 astrocytes are an accurate disease model. This question also holds for immortalized cells lines, such as the human glial cell line MIO-M1, used in this study.

It is obvious that the authors are trying to link the strong effects they see in culture with the very modest effects seen on astrocyte morphology and polarization *in vivo*. However, in my opinion, the link is weak. In particular, their argument is undermined by the fact that only one developmental time point is presented for *in vivo* analysis: six weeks old mice. Given the rapidity with which RNA foci develop (the author's own data from their tetracycline inducible system), I would expect to see effects during development (for example, see the time-course of cortical astrocyte development in Stogsdill et al., Nature, 2017). In fact, the authors themselves raise the issue of "Altered adhesion of astrocytes in DM1 affecting migration, orientation and integration into the complex microenvironment of the brain." Have the authors any *in vivo* information of this type which would support the thrust of their manuscript? Otherwise, how can later onset astrocyte degeneration, as the underlying cause of reduced ramification and loss of cell polarity, be excluded *in vivo*?

Along similar lines, the authors also present transcriptome analysis from human cells, which is a very nice addition, and which supports the transcriptome data from mouse. Unfortunately, there is no morphological information supplied to support an effect of CTG expansion in humans causing deficits in cell migration, adhesion and membrane trafficking.

While I am not opposed to work that includes cultured astrocytes, published manuscripts now tend to have substantial *in vivo* validation (e.g. Allen et al., Nature, 2012; Chung et al., Nature,

2013; Blanco-Suarez et al., Neuron, 2018). As it stands, I think this submission is extremely interesting but premature and lacks the weight of evidence needed to prove *in vivo* effect. Similarly, I think the claim that the results demonstrate "General vulnerability of astrocytes to CUG RNA toxicity" is also an over statement. In this respect, studying cortical and hippocampal astrocytes is not sufficient, as we now know astrocytes to be highly heterogenous between brain regions, Morel et al., J Neurosci, 2017; Chai et al., Neuron, 2017 – which may also influence the response to (toxic) expression of CUG repeats.

Finally, the authors make a well-reasoned case that changes in morphology due to DM1 would impact on synaptic function: "Impaired astrocyte arborization and polarity....may not only reduce the direct contacts between astrocyte processes and neuronal dendrites, but also impact the efficiency of synaptic transmission, contributing to altered plasticity." The authors have previously shown defects in Bergmann glia (impaired GLT1 function) impact on the activity of local Purkinje cells (Sicot et al., Cell Rep, 2017). Given the authors claims, and their pre-existing technical expertise, I would also expect data proving functional impairment in this manuscript.

Minor issues:

- Blotting for cell markers does not, in my opinion, provide information on absolute cell numbers. This needs to be performed using immunohistochemistry.
- Cytoplasmic GFP expression does not reliably label fine astrocyte processes, which may mean that the morphological studies are not as accurate as possible. Membrane-associating labels (e.g. Lck-GFP; Shigetomi et al., J Gen Physiol, 2013) may be more appropriate.
- It is not always clear why the authors jump from cortex to hippocampus for measurements. Likewise, some areas of the manuscript could benefit from better explanation; for example, I assume Figure 5 is a PCR.
- Is there a clear relationship between transgene expression/RNA foci number/ sequestration of splicing factors? Why does transgene expression appear to drop after time?

Reviewer #2:

Remarks to the Author:

The Nat. Comm. NCOMMS-20-51264 manuscript by Dinca and collaborators characterize DM1 CNS dysfunction, in primary cultures, transgenic mice and human brain samples. They have demonstrated aberrant adhesion, spreading and migration of astrocytes associated with more severe RNA spliceopathy than in neurons.

This study opens new perspectives on the involvement of astrocytes in the neurological symptoms of patients with DM1. However, further studies are needed to support the hypotheses put forward by the authors and to demonstrate the contribution of astrocytes and the mechanisms involved.

- 1) Based on the data in the paper it is difficult to figure out whether the change in astrocytes are due to a loss of maturation during development or an alteration occurring in fully mature astrocytes?
- 2) The authors studied the defect in the focal adhesion organization, however, other defects, for example in gap junction proteins, were not considered although they could also explain the altered morphology and polarity. It would be important to better characterize this astrocytic phenotype.
- 3) The source of astrocytes for the RNA sequencing analysis is unclear: cortical, and/or hippocampal? Did transcriptomic analysis reveal differences between the cortex and the hippocampus?
Do you expect to observe this splicing defect in all astrocytes of the brain? Additional data would be important to start to decipher the mechanism leading to this abnormal adhesion and spreading and further investigate the cellular specificity.
- 4) One hypothesis to explain the astrocytic defects is that inactivation of MBNL could reduce cell adhesion. Integrating new *in vitro* experiments on this subject would significantly increase the value of the paper.

Other comments:

In the first chapter, the authors, based on unchanged expression of some astrocytic markers, conclude that astrocyte cell density is unchanged. This could be the case but a more quantitative analysis should be performed to validate this conclusion and exclude confounding factors.

The analysis was carried out on one-month-old animals. What about later stages of the disease?

Is the phenotype in the cortex and hippocampus similar to the previous described cerebellar abnormalities?

Supplementary Figure 1: Which regions of the brain are the primary cultures derived from?

Focal adhesion measurement is central to the paper, a description in the materials and methods section would be essential.

Page 10, second chapter: what does the term "homogeneous primary cultures" mean?

Reviewer #3:

Remarks to the Author:

The authors have shown in a transgenic mouse model of DM1 and in human tissues that astrocytes are more affected than neurons.

This is an interesting, novel and original finding which could explain the cerebral involvement in DM1, particularly in behavioural aspects, such as "avoidant" -Meola et al, *Neuromuscular Disorders*, 2003, Winblad et al, *NMD*, 2005;

They have shown an alteration of cell biology and a pronounced spliceopathy in astrocytes from frontal and hippocampal regions in DMSX models. They found the same alterations in human frontal cortex regions. Did they investigate in human tissues other regions, such as hippocampus? If not they should explain why.

They should mention also some recent neuroimaging studies exploring lesion distribution and white matter damage in DM1 and in comparison to MS, where it is also emphasized the pronounced temporal pole alterations in DM1 (S. Leddy et al, *Neuroimage Clinical*, 29, 2021)

Dr Meola Giovanni

Reviewer #4:

Remarks to the Author:

The manuscript by Dinca et al. describes novel findings related to the CNS dysfunction in myotonic dystrophy type 1. The authors use a well-established transgenic mouse model of DM1 to demonstrate impaired morphogenesis of astrocytes, the key supporting cells of the CNS. They observe defects in astrocyte ramification, polarization, adhesion, spreading and migration. These findings are well supported by molecular analyses and also by RNA-seq in primary cell cultures, which revealed missplicing of transcripts involved in these processes. Importantly, the Authors verify these alternative splicing events in vivo in astrocyte-enriched cell fraction isolated from brains of DMSXL mice, but also in post-mortem brain tissue of DM1 patients. These are novel findings and although DM1-related pathogenic CUG foci have been reported in distinct cell types in the CNS including neurons and astrocytes, the mechanistic contribution of astrocytes to DM1 pathology has not been studied very extensively. Transcriptome and alternative splicing alterations were explored previously in DM1 tissues mainly in skeletal muscle and heart, and also very recently in DM1/DM2 frontal cortex; however, Dinca et al. revealed astrocyte-specific splicing defects which were otherwise undetectable in whole-brain tissue samples, emphasizing the important role of localized RNA toxicity in astrocytes.

Overall, this is an exciting and important topic and these novel data presented by the Authors will be a valuable addition to studies dealing with the pathological mechanisms of DM1 in the CNS. The experimental work is thorough and well documented. The manuscript is well-written and organized

and the data is presented with sufficient context and consideration of previous works. I only have a few points which the Authors could address prior to publication, to improve the clarity of the results and the message they want to convey.

1) Figure 1c – The authors could have used DM20 mice as an additional control, to exclude the potential effect of DMPK overexpression alone on astrocyte morphology and the observed hypotrophy of the cytoskeleton. This comment can be extended to other experiments, for example the ones shown in Figures 3-4. The authors use such control only in Figure 3d, however. Why?

2) Figure 5c – This data is not very convincing. The authors claim increased MBNL1&2 co-localization with CUG foci in DMSXL astrocytes, but majority of the signal is in the cytoplasm; there is surprisingly very little staining in the nuclei. Also, from the images shown it looks as if there is less MBNL1&2 signal in astrocytes compared to neurons. The authors might consider providing data for MBNL1-2 protein levels by western blotting and perform Mbnl1&2 alternative splicing analyses (i.e. alternative exons known to be affected in DM1 and determining cytoplasmic vs nuclear localization). If MBNLs sequestration is indeed the main mediator of splicing alterations observed in DMSXL astrocytes (as stated in the discussion, lines 341-345), these additional data could perhaps explain fewer splicing alterations observed in DMSXL neurons. Also, WT controls are missing.

3) Line 251-252 of the manuscript – concerning RNA seq experiments; which transcripts were significantly affected in DMSXL astrocytes in terms of expression level? How are they relevant to the observed phenotype? There is no further comment in the text, and I assume not every reader will want to dig through extensive supplementary tables to find it out.

4) Figure 6&7 – What mediates these splicing changes? The authors show unaltered CELF proteins (Fig. 5d) and claim that MBNLs inactivation by RNA foci (Fig. 5c) is the determinant event behind splicing changes. Again, the authors could consider analyzing alternative splicing of Mbnl1&2 pre-mRNAs as well as protein levels in DMSXL / wt astrocytes vs neurons.

5) Discussion - What is the Authors' interpretation of the phenotypes and mechanisms found in DM1 astrocytes: are they causative to the neurological manifestations observed in DM1, whereas defects in DM1 neurons are secondary effects? Stronger point could be made in the discussion.

Point-by-Point Response to Reviewers

We thank all the reviewers for their thorough and critical evaluation of our manuscript. We greatly appreciate their constructive comments and insightful suggestions, which have helped us improve our study. We have revised our manuscript with a series of additional experiments, and changes to the text to carefully address every point raised. Below is our point-by-point response to the Reviewers.

Reviewer #1 (Remarks to the Author):

Myotonic dystrophy RNA toxicity alters morphology, adhesion and migration of mouse and human astrocytes.
Dincă et al.

Myotonic dystrophy type 1 (DM1) is caused by expansion of the CTG trinucleotide in the 3' untranslated region of the DMPK gene. RNAs containing expanded CUG triplets accumulate in the nucleus of DM1 cells, forming RNA foci that perturb the localization and function of RNA-binding proteins. Using a transgenic mouse model of DM1, the authors investigate the effects of DM1 on CNS cells. Interestingly, they report the most significant effects on astrocytes, the major non-neuronal cell type in the brain. Astrocytes exhibit impaired ramification and polarization *in vivo* and defects in adhesion, spreading and migration *in vitro*. Transcriptome analysis revealed mis-splicing of transcripts regulating adhesion, cytoskeleton and morphogenesis. The authors report these changes were also seen in human DM1 cells. Hence, the authors conclude that DM1 impacts astrocyte biology, presumably compromising the cells ability to support and regulate synaptic function.

The idea that astrocytes play a major (if not the dominant) role in CNS disease is an interesting idea and, if true, would represent a paradigm shift in neuroscience. However, in my opinion, there are several issues which raise question marks over the suitability of the work for publication at this time.

We thank Reviewer #1 for acknowledging the interest, novelty and conceptual advance provided by our work.

Major issues:

The bulk of the work reported in this manuscript is performed using cultured astrocytes produced using the “classical method”. Astrocytes cultured in this manner morphologically resemble fibroblasts with a flat morphology and show high rates of growth. This is in contrast to the star-shaped cells found *in vivo*, which show (comparatively) low rates of growth (Clavreul et al., Nat Commun, 2019). Furthermore, sequencing studies show large differences between cultured cells and acutely isolated astrocytes (Foo et al., Neuron, 2011). Finally, astrocytes *in vivo* do not migrate in response to injury (Tsai et al., Science, 2012), in direct contrast to the scratch assay data shown in the paper.

In my mind, the question which needs to be asked is to what degree “classical” astrocyte cultures represent the *in vivo* situation (Foo et al., Neuron, 2011) and whether cultures of DM1 astrocytes are an accurate disease model. This question also holds for immortalized cells lines, such as the human glial cell line MIO-M1, used in this study.

The Reviewer’s concerns are justified. We agree on the critical importance that the molecular and cellular phenotypes reported in culture represent *in vivo* conditions. To provide evidence that the data collected in primary cell cultures provide important insight into the situation *in vivo*, we have now isolated mature astrocytes from adult mouse brains and studied them shortly after plating, hence avoiding possible confounding effects introduced by long culture periods. As expected, adult astrocytes grown for 3 DIV displayed more ramified and irregular cell shapes and lower growth rates, when compared to classical primary astrocytes derived from newborn mice. More importantly, data on adult astrocytes confirmed the reduced cell size and lower number of focal adhesions, previously reported in “classical” newborn DMSXL primary astrocyte cultures (Fig. 4e).

Results, page 11, lines 211-218:

“To investigate whether the adhesion defects in newborn primary DMSXL astrocytes likely reflect *in vivo* phenotypes, we plated mature cortical astrocytes acutely isolated from 1-month-old mice and investigated cell spreading and focal adhesion organization. The analysis confirmed a significant reduction in cell size and number of focal adhesions per cell in DMSXL astrocytes (Fig. 4e). Together with reduced ramification of astrocytes in DMSXL brains, these data provide further evidence of the impact of expanded CUG RNA on cell adhesion and morphology.”

Furthermore, we used the same pipeline described in our manuscript, to compare the RNA sequencing data of primary WT astrocytes (our own data), with WT astrocytes isolated from mouse cortex at postnatal day 7 (Zhang et al. 2015 J Neurosci. doi: 10.1523/JNEUROSCI.1860-14.2014). The analysis revealed a highly significant correlation between the percentage of splicing inclusion (PSI) of 13,433 alternative exons identified in both samples, indicating relevant similarities in splicing regulation (Fig. A, below). Exons were classed as alternatively spliced, when $0.05 < \text{PSI} < 0.95$ in at least one sample. In other words, exons showing $\text{PSI} < 0.05$ or $\text{PSI} > 0.95$ were considered to be constitutively excluded or included and excluded from the comparison. This comparison suggests that “classical” primary astrocyte cultures provide important insight into transcriptome regulation *in vivo*. In further support of this view, the splicing changes found in primary DMSXL astrocytes were confirmed in astrocytes derived from adult DMSXL forebrain (Fig. 7e), as well as in other brain regions (Supplementary Fig. 9).

Figure A (for reviewers only). Correlation between the PSI values of alternative exons in cultured primary astrocytes derived from newborn mice, and astrocytes isolated from mouse cortex at postnatal day 7.

It is obvious that the authors are trying to link the strong effects they see in culture with the very modest effects seen on astrocyte morphology and polarization *in vivo*. However, in my opinion, the link is weak. In particular, their argument is undermined by the fact that only one developmental time point is presented for *in vivo* analysis: six weeks old mice. Given the rapidity with which RNA foci develop (the author’s own data from their tetracycline inducible system), I would expect to see effects during development (for example, see the time-course of cortical astrocyte development in Stogsdill et al., Nature, 2017). In fact, the authors themselves raise the issue of “Altered adhesion of astrocytes in DM1 affecting migration, orientation and integration into the complex microenvironment of the brain.” Have the authors any *in vivo* information of this type which would support the thrust of their manuscript? Otherwise, how can later onset astrocyte degeneration, as the underlying cause of reduced ramification and loss of cell polarity, be excluded *in vivo*?

We agree with the Reviewer’s comment, and we acknowledge that the inclusion of additional time points would strengthen the manuscript. Accordingly, we have made the following changes, which we hope fully address the Reviewer’s comment.

- We have studied astrocyte ramification *in vivo* at earlier (postnatal day 10) and later (4 months) time points (Supplementary Fig. 1a-c).
- We have quantified total GFAP protein expression in mouse frontal cortex at earlier and later time points (Supplementary Fig. S1d).
- We have performed additional RT-PCR splicing analysis of candidate transcripts at postnatal day 10 and at 4 months of age (Supplementary Fig. 8e, f).

Together the new results reveal that the morphological phenotypes of DMSXL astrocytes are absent shortly after birth, appearing postnatally between day 10 and 30, and persisting (and possibly aggravating) until 4 months of age (Supplementary Fig. 1c). Importantly, astrocyte phenotypes are accompanied by splicing dysregulation of cytoskeleton- and adhesion-related transcripts, which is

undetected before postnatal day 10 but persists until 4 months of age at least (**Supplementary Fig. 8e, f**).

Interestingly, the cytoskeleton- and adhesion-related transcripts studied undergo a postnatal fetal-to-adult splicing transition in WT mice between postnatal P10 and P30 (**Supplementary Fig. S8b, e**). Full splicing switch is impaired in DMSXL mice, resulting in the abnormal expression of fetal splicing isoforms in adult brains. Specifically, we have recently reported that DMSXL astrocytes continue to express splicing isoforms that are typical of immature astroglia (Gonzalez-Barriga et al. 2022 Front Cell Neurosci. doi: 10.3389/fncel.2021.662035). These data support the view that astrocyte phenotypes are likely the consequence of a defective developmental splicing program, which under physiological conditions orchestrates postnatal splicing transitions. This point is now discussed in the final section of the manuscript.

Discussion, pages 21, lines 449-455:

“Defective DMSXL astrocyte branching in vivo emerges between postnatal day 10 and 1 month, a time window corresponding to the developmental switch from embryonic and adult splicing isoforms. We recently reported that DMSXL astrocytes show increased expression of splicing isoforms typical of immature astroglia⁴¹. These results corroborate the view that DM1 perturbs primarily the splicing of developmentally regulated transcripts⁴², and suggest that astrocyte phenotypes result mainly from the postnatal impairment to express adult splicing isoforms.”

We could not investigate the polarization of astrocytes before 1 month, because astrocyte re-orientation takes place between postnatal week 3 and 4 (Nixdorf-Bergweiler et al. 1994 Glia. doi: 10.1002/glia.440120304). The magnitude of the polarity index changes in DMSXL astrocytes in *striatum radiatum* (relative change of ~20%), is comparable to that reported during mouse postnatal development, between P10 and P30 (relative change of ~25%) (Ghézal et al. 2018 Development. doi: 10.1242/dev.15527). Similarly, the magnitude of the arborization defects of DMSXL astrocytes *in vivo* is comparable to that reported in different mouse models of astrocyte dysfunction (e.g. Wahis et al. 2021 Nature Neuroscience. doi: 10.1038/s41593-021-00800-0; Pillet et al. 2020 Glia. doi: 10.1002/glia.23801; Codeluppi et al. 2021 Int J Neuropsychopharmacol. doi: 10.1093/ijnp/pyab052; Virmani et al. 2021 Eur J Neurosci. doi: 10.1242/jcs.258430), or following pharmacological intervention (Sethi et al. 2021 J Cell Sci. doi: 10.1242/jcs.258430). We have added brief comments in the final discussion, to place our observations in the perspective of previous results.

Discussion, page 24, lines 530-535:

“We propose that the impaired astrocyte arborization and polarity in DMSXL mice (comparable to that found in other mouse models of astrocyte dysfunction^{24,65}), affects the direct contacts between astrocyte processes and neuronal dendrites in vivo.”

Along similar lines, the authors also present transcriptome analysis from human cells, which is a very nice addition, and which supports the transcriptome data from mouse. Unfortunately, there is no morphological information supplied to support an effect of CTG expansion in humans causing deficits in cell migration, adhesion and membrane trafficking.

We have analyzed additional parameters of the MIO-M1 cell line, to further validate the mouse findings in a human glial cell model.

Results, page 10, lines 187-192:

*“Like DMSXL astrocytes, for the same number of cells attached (**Supplementary Fig. 3b**), dox-induced MIO-M1 cells displayed reduced confluence 45 minutes after plating relative to non-induced controls (**Supplementary Fig. 3c**). We monitored cell spreading and morphology by live cell videomicroscopy (**Supplementary Fig. 3d**) and found that MIO-M1 cells expressing toxic CUG transcripts showed decreased cell spreading over a period of 12 hours (**Supplementary Fig. 3e**).”*

The validation of astrocyte phenotypes in human tissue samples would strengthen the present manuscript. However, DM1 brain samples are rarely collected and are scarce in tissue banks. The human tissue samples used in this study were snap frozen for RNA and protein collection, and they are not suitable for histological analysis. However, there is an ongoing joint effort of the myotonic dystrophy community to expand the autopsy retrieval and build a large repository of tissues suitable for molecular and histological analysis, and that will include rare congenital, childhood and juvenile

cases, presenting relevant neurological manifestations. Unfortunately, at the moment these samples are not available.

While I am not opposed to work that includes cultured astrocytes, published manuscripts now tend to have substantial *in vivo* validation (e.g. Allen et al., *Nature*, 2012; Chung et al., *Nature*, 2013; Blanco-Suarez et al., *Neuron*, 2018). As it stands, I think this submission is extremely interesting but premature and lacks the weight of evidence needed to prove *in vivo* effect. Similarly, I think the claim that the results demonstrate “General vulnerability of astrocytes to CUG RNA toxicity” is also an over statement. In this respect, studying cortical and hippocampal astrocytes is not sufficient, as we now know astrocytes to be highly heterogeneous between brain regions, Morel et al., *J Neurosci*, 2017; Chai et al., *Neuron*, 2017 – which may also influence the response to (toxic) expression of CUG repeats.

We now provide additional validation of astrocyte phenotypes *in vivo*, through the quantification of astrocyte cell density, GFAP protein expression and Sholl analysis in mouse brains at additional ages (**Supplementary Fig. 1a-d**). The abnormal cell spreading and focal adhesion assembly in astrocytes acutely isolated from juvenile mouse brains (**Fig. 4e**) further corroborates the findings in primary DMSXL astrocytes derived from newborn pups. Finally, we confirmed astrocyte spliceopathy *in vivo*, through the analysis of astrocytes acutely isolated from multiple regions of adult mouse brains (**Fig. 7e; Supplementary Fig. 9**). The new data have not only elucidated the timing of astrocyte abnormalities, but they have also offered some mechanistic insight into the molecular defects behind reduced astrocyte arborization.

We agree with the Reviewer’s comment on the astrocyte heterogeneity between brain regions. We have initially focused our studies on the frontal cortex, given the suggested role of the frontal lobe in the prevalent executive dysfunction of DM1. Hippocampus was also studied because it is a key brain region involved in the regulation of attention, visuospatial memory and emotion, which are frequently impaired in DM1 (Meola et al. 2007, *Muscle and Nerve*, doi: 10.1002/mus.20800; Okkersen et al. 2017, *Cortex*, doi: 10.1016/j.cortex.2017.08.008). We have now clarified the focus on frontal cortex and hippocampus in the introduction of our revised manuscript.

Introduction, page 3, lines 52-54:

“While the involvement of frontal lobe is suggested by the prevalent executive dysfunction, hippocampus neuropathology may contribute to deficits in visuospatial memory and learning^{6,9}.”

Importantly, we have extended the splicing analysis to astrocytes acutely isolated from adult frontal cortex, hippocampus and cerebellum (**Supplementary Fig. 9**), demonstrating that astroglia are affected by toxic CUG RNA repeats across the mouse brain. However, subtle but relevant regional differences can be detected between specific astrocyte subpopulations. This is the topic of ongoing projects in our laboratory, which we believe to be out of the scope of this manuscript. Here, we aim at convincingly demonstrating that astroglia is molecularly and phenotypically impacted by toxic CUG RNA.

Results, page 17, lines 374-379:

“While Dmd, Itga6, Mpdz and Numa1 exhibited widespread missplicing, other transcripts showed region-specific defects, suggesting regional susceptibility of astrocyte subpopulations to toxic CUG RNA. Sorbs1 exon6 was only significantly affected in frontal cortex astrocytes, Fermt2 abnormalities were detected in both frontal cortex and hippocampus astrocytes, and Capzb was only significantly misspliced in DMSXL cerebellum.”

Despite these new and convincing data, we agree with the Reviewer that the expression “general vulnerability of astrocytes to CUG RNA toxicity” may be seen as an overstatement. As a result, we have toned down our final conclusions.

Discussion, page 20, lines 426-428:

“Here, we uncovered the wider vulnerability of astrocytes to CUG RNA toxicity, beyond the involvement of Bergmann glia.”

Finally, the authors make a well-reasoned case that changes in morphology due to DM1 would impact on synaptic function: “Impaired astrocyte arborization and polarity....may not only reduce the direct contacts between astrocyte processes and neuronal dendrites, but also impact the efficiency of synaptic transmission, contributing to altered plasticity.” The authors have previously shown defects in Bergmann glia (impaired

GLT1 function) impact on the activity of local Purkinje cells (Sicot et al., Cell Rep, 2017). Given the authors claims, and their pre-existing technical expertise, I would also expect data proving functional impairment in this manuscript.

To demonstrate the influence of DMSXL astrocytes on neurons, we have monitored neurite outgrowth in co-culture systems of mixed genotypes. We found a negative impact of DMSXL astrocytes on neurite development of both DMSXL and WT neurons. However, synaptic density remained unaltered in adult DMSXL frontal cortex. These new data are included in two new figures (**Fig. 5, Supplementary Fig. 5**).

Results, pages 12-13, lines 248-272:

“DMSXL astrocytes enhance neuritogenesis defects of primary neurons

*Astrocytes can promote neurite growth and synapse formation³⁴, therefore we tested if DMSXL astrocytes affected neuritogenesis. We first monitored primary neuron neuritogenesis by time-lapse video-microscopy. Bright field tracking of neurite growth over longer periods of time (7 days in vitro, DIV) revealed a mild reduction in the growth rate of DMSXL neurites, which translated into significantly shorter neurites at later points, from 4.5 DIV onwards (**Fig. 5a**). The defect in neuritogenesis was confirmed by the analysis of transfected primary neurons, expressing mKate2 fluorescent protein under the Synapsin 1 promoter (**Fig. 5b**).*

*To investigate the influence of DMSXL astrocytes on neuritogenesis, we monitored neurite outgrowth of DMSXL and WT neurons co-cultured with astrocytes of either genotype. Fluorescent time-lapse videomicroscopy revealed a significant reduction of neurite outgrowth in DMSXL/DMSXL co-cultures, relative to WT/WT controls (**Fig. 5c**). More importantly, we found a negative impact of DMSXL astrocytes on the neurite development of both DMSXL and WT neurons, which resulted in significant lower neurite growth rates. Interestingly, DMSXL and WT neurons exhibited similar neurite growth rates in the presence of DMSXL astrocytes, corroborating the dominant effect of DMSXL astrocytes on neurite projection development.*

*Since astrocytes tightly interact and functionally regulate synapses³⁴, we investigated synaptic density in DMSXL mouse brains, as a first indication of potential disrupted synapse formation or maintenance. Immunolabeling of synapses in mouse frontal cortex revealed that the number of excitatory and inhibitory synapses remained unchanged in DMSXL, indicating similar total synaptic contacts (**Fig. 5d and Supplementary Fig. 5a, b**).*

To conclude, the interplay of DMSXL astrocytes with neurons affects neuronal development and maturation in culture”

Although our results do not point to overt changes in synaptic density, we have also gathered functional evidence of abnormal neurotransmission *in vivo*, demonstrated by significant defects in neurotransmitter homeostasis and synaptic plasticity that we published very recently. Notably, DMSXL mice display defective glutamate uptake *in vivo*, impaired short-term and long-term synaptic plasticity, in association with increased glutamate and GABA tonic currents (Parrot et al. 2021 ACS Chem Neurosci. doi: 10.1021/acchemneuro.1c00634; Potier et al. 2022 Int J Mol Sci. doi: 10.1242/jcs.258430). It is conceivable that defective astrocyte ramification contributes to detrimental changes in the synaptic ultrastructure, impaired neurotransmitter clearance, elevated ambient levels of neurotransmitters, higher tonic currents and abnormal synaptic transmission. Our original hypothesis is now briefly discussed in light of our results, recently published. The link between altered astrocyte morphology and synaptic dysfunction requires however further investigation, that is currently undergoing in our laboratory.

Discussion, page 24, lines 523-535:

“Alternatively, since astrocyte-neuron communication is intimately dependent on the specialized morphology of both cell types, aberrant astrocyte ramifications are predicted to be detrimental to neurons. In this context, we showed that DMSXL astrocytes delay neurite growth in culture, but the overall synaptic density is not altered in vivo. [...] We propose that the impaired astrocyte arborization and polarity in DMSXL mice (comparable to that found in other mouse models of astrocyte dysfunction^{24,64}), affects the direct contacts between astrocyte processes and neuronal dendrites in vivo. Defects in the structural and functional interplay between astrocytes and neurons may contribute to the aberrant tonic currents and synaptic plasticity of DMSXL mice⁶⁵, exacerbating neuronal and synaptic dysfunction.

Minor issues:

- Blotting for cell markers does not, in my opinion, provide information on absolute cell numbers. This needs to be performed using immunohistochemistry.

We agree with the Reviewer's concern. We have replaced the immunoblotting of astrocyte protein markers with the immunofluorescent detection of routinely used markers. The double detection of proteins highly enriched in adult mouse astrocytes (nuclear SOX9 and cytoplasmic S100B.) allowed us to quantify astrocyte density in mouse cortex and confirm normal levels in DMSXL animals (**Fig. 1a**). This section of the manuscript has been carefully reviewed.

Results, page 6, lines 93-98:

"We first estimated astrocyte cell density in the mouse brain by immunodetection of two astrocyte-specific markers (SOX9, SRY-Box Transcription Factor 9; and S100B, S100 calcium-binding protein B)²³. The percentage of co-labeled cells (SOX9+/S100B+) in the frontal cortex was not significantly different between DMSXL and WT mice at 1 month of age, suggesting unaltered astrocyte density (Fig. 1a)."

- *Cytoplasmic GFP expression does not reliably label fine astrocyte processes, which may mean that the morphological studies are not as accurate as possible. Membrane-associating labels (e.g. Lck-GFP; Shigetomi et al., J Gen Physiol, 2013) may be more appropriate.*

The use of soluble GFP to visualize whole cell shape of astrocytes *in vivo* has been validated by a number of significant and recent studies (Lanjakornsipiran et al. 2018 Nat Commun. doi: 10.1038/s41467-018-03940-3; Cheung et al. 2022 Nat Commun. doi: 10.1038/s41467-022-28331-7; Ung et al. 2021 Nat Commun. doi: 10.1038/s41467-021-25444-3; Rafaeli et al. 2021 Glia. doi: 10.1002/glia.24044; Codeluppi et al. 2021. Int J Neuropsychopharmacol. doi: 10.1093/ijnp/pyab052). We however agree that membrane-targeted Lck-GFP would reveal an increased number of fine astrocytic processes, and we thank the Reviewer for this insightful suggestion. However, Lck-GFP labeling reveals a complex spongiform structure of astrocytes (see figures below), which makes it very challenging to reliably identify individual fine astroglial processes. This labeling is thus not compatible with the classical Sholl analysis that we performed here. Instead, Lck-GFP is suitable for the quantification of the total volume occupied by individual astrocytes. Of course, this approach would provide an additional element in the morphological analysis of DMSXL astrocytes *in vivo*, but is unlikely to provide further quantitative assessment of individual fine astroglial processes. Given these difficulties, and as our current study focus primarily on the demonstration of the deleterious influence of CUG RNA toxic repeats on astroglia, we view this experiment beyond the scope of this manuscript. Although we believe this will be an interesting future experiment, we consider that the Sholl analysis of cytoplasmic GFP-labelled processes provide convincing evidence of the impact of toxic CUG RNA on astrocytes.

Figure B (for reviewers only). Cell imaging of individual cells in mouse prefrontal cortex, showing the complex spongiform morphology of a Lck-GFP-expressing astrocyte. From Testen et al. 2019 Neuroscience. doi: 10.1016/j.neuroscience.2018.12.044.

Figure C (for reviewers only). Cell imaging of a Lck-GFP-labeled astrocyte (green) in the mouse brain. GFAP counterstaining is also shown (red). The complex spongiform structure of the Lck-GFP-labeled astrocytes makes it very difficult to study the ramification of individual cytoplasmic processes.

- It is not always clear why the authors jump from cortex to hippocampus for measurements. Likewise, some areas of the manuscript could benefit from better explanation; for example, I assume Figure 5 is a PCR.

We have clarified these points. As discussed above, we have focused on frontal cortex and hippocampus given the relevance of both brain regions to the neuropathology of DM1. The selection of these two brain regions has been justified in the manuscript. Astrocyte polarity was only studied in the hippocampus, however, since this is the region where the orientation of GFAP-expressing processes has been well reported to change postnatally (Nixdorf-Berweiler et al. 1994 *Glia*. doi: 10.1002/glia.440120304).

Results, page 7, lines 120-124:

“Starting at the third week of postnatal development, i.e. at the time of hippocampal synaptogenesis, the CA1 astrocytes of the stratum radiatum change the orientation of GFAP-rich stem processes to a fusiform orientation, perpendicular to the pyramidal cell layer²⁶, in a process that depends on the dynamic changes of the cytoskeleton and cell morphology.”

The Reviewer is correct: Fig. 5a (now **Fig. 6a**) represents quantitative RT-PCR analysis of gene expression. We apologize for the ambiguity that we have clarified in the figure legend.

Fig. 6, page 58, lines 1170-1172:

“Quantitative RT-PCR expression analysis of human DMPK transgene and mouse endogenous Dmpk relative to 18S rRNA internal control in primary DMSXL astrocytes and neurons at 6, 12 and 30 days in vitro.”

- Is there a clear relationship between transgene expression/RNA foci number/ sequestration of splicing factors? Why does transgene expression appear to drop after time?

The relationship between transgene expression, RNA foci and sequestration/inactivation of splicing factors is an important question in the field, that has been alluded to by the comparative analysis of different transgenic mouse models of DM1: higher levels of toxic transcripts in muscle are associated with more abundant RNA foci, pronounced RNA missplicing and severe muscle pathology (Mahadevan et al. 2006. *Nat Genet*. doi: 10.1038/ng1857; Gudde et al. 2016 *Hum Mol Genet*. doi: 10.1093/HMG/DDW042). However, the situation appears to be more complex, and pathology may not depend exclusively on the levels of toxic CUG transcript levels. In an attempt to gain insight into the factors contributing to the more severe spliceopathy of DMSXL astrocytes, we have recently quantified the levels of human *DMPK* transcripts, relative to mouse *Mbnl1* and *Mbnl2*. Interestingly, DMSXL astrocytes displayed the highest *DMPK/Mbnl1* and *DMPK/Mbnl2* ratios relative to other brain cell types (Gonzalez-Barriga et al. 2022 *Front Cell Neurosci*. doi: 10.3389/fncel.2021.662035). Transgene expression levels may not be the sole determinant of cell dysfunction, as suggested by others (Otero et al. 2021 *Cell Rep*. doi: 10.1016/j.celrep.2020.108634). We have considered these important points in the final discussion, and we thank the Reviewer for bringing up this point.

Discussion, pages 20-21, lines 439-444:

“Previous comparisons between DM1 mouse models revealed that higher expression of toxic CUG transcripts is associated with greater foci content and more severe muscle spliceopathy and pathology^{39,40}. In addition to higher DMPK expression, DMSXL astrocytes display the highest DMPK/Mbnl1 and

DMPK/Mbnl2 ratios (relative to other brain cell types), which provide conditions conducive to MBNL protein sequestration as previously suggested^{37,41}.”

The Reviewer raises another relevant question, concerning the differentiation- and development-dependent regulation of *DMPK* gene expression. The decrease in *DMPK* transcripts reported in primary astrocytes mimics that of endogenous *Dmpk* gene (**Fig. 6a**), as well as the decrease of *DMPK* expression from fetal to adult human astrocytes *in vivo* (Zhang et al. 2014. J Neurosci. doi: 10.1523/JNEUROSCI.1860-14.2014). Similarly, *DMPK* expression in human myogenic cells peaks during early differentiation of myoblasts into myotubes, dropping shortly afterwards (Gudde et al. 2016, Hum Mol Genet. doi: 10.1093/HMG/DDW042). *In vivo*, *DMPK* expression also decreases postnatally in DMSXL skeletal muscle (Michel et al. 2015, PLoS ONE. doi: 10.1371/journal.pone.0137620) and in human skeletal muscle (Furling et al. 2003, Am J Pathol. doi: 10.1016/s0002-9440(10)63894-1). Together these findings suggest a differentiation- and development-dependent regulation of gene expression, whose mechanisms are not fully elucidated. A comment on the regulation of *DMPK* gene expression has been integrated in the final discussion.

Discussion, page 20, lines 433-436:

“Transgene expression in astrocytes follows the profile of endogenous Dmpk mouse gene, decreasing during cell differentiation like the human DMPK gene in brain¹⁵ and skeletal muscle³⁸, and demonstrating that the DMSXL transgene contains important elements for the regulation of its own expression.”

Reviewer #2 (Remarks to the Author):

The Nat. Comm. NCOMMS-20-51264 manuscript by Dinca and collaborators characterize DM1 CNS dysfunction, in primary cultures, transgenic mice and human brain samples. They have demonstrated aberrant adhesion, spreading and migration of astrocytes associated with more severe RNA spliceopathy than in neurons.

This study opens new perspectives on the involvement of astrocytes in the neurological symptoms of patients with DM1. However, further studies are needed to support the hypotheses put forward by the authors and to demonstrate the contribution of astrocytes and the mechanisms involved.

We thank Reviewer #2 for acknowledging the new insight provided by our work.

1) Based on the data in the paper it is difficult to figure out whether the change in astrocytes are due to a loss of maturation during development or an alteration occurring in fully mature astrocytes?

The Reviewer raises an important question on the onset and progression of astrocyte phenotypes. Accordingly, we have made the following changes, which we hope

- a) We have studied astrocyte ramification *in vivo* at earlier (postnatal day 10) and later (4 months) time points (**Supplementary Fig. 1a-c**).
- b) We have quantified total GFAP protein expression in mouse frontal cortex at earlier and later time points (**Supplementary Fig. S1d**).
- c) We have performed additional RT-PCR splicing analysis of candidate transcripts at postnatal day 10 and at 4 months of age (**Supplementary Fig. 8e, f**).

We believe the analysis of additional time points provided in Supplementary Figure 1, together with the new analysis of missplicing at postnatal day 10 and 4 months provides important insight. Our data show that astrocyte phenotypes emerge postnatally, between postnatal 10 and 1 month of age, a period that corresponds to the transition between fetal and adult splicing isoforms (**Supplementary Fig. 8e**). Interestingly, other MBNL target transcripts undergo a developmental switch in the skeletal muscle of wild-type mice (Lin et al. 2006 Hum Mol Genet. doi: 10.1093/hmg/ddl132). We have recently demonstrated that primary DMSXL astrocytes differentiated in culture for 14 DIV express higher levels of RNA splicing isoforms typical of astroglia precursors isolated at P1 (Gonzalez-Barriga et al. 2022 Front Cell Neurosci. doi: 10.3389/fncel.2021.662035). We conclude that the phenotypic changes detected are the result of a defective postnatal developmental splicing switch, which normally occurs postnatally during the first month postnatally. Based on the new experimental data collected, this view is now fully discussed in our revised manuscript.

Results, page 6, lines 105-111:

*“To explore the progression of astrocyte hypotrophy, we investigated additional mouse ages. While DMSXL astrocyte ramification was unaltered at 10 days of age (**Supplementary Fig. 1a**), it was significantly reduced at 4 months (**Supplementary Fig. 1b**). Interestingly, the reduction in astrocyte processes was more severe at 4 months in processes distanced 10-15 μm from the nucleus (**Supplementary Fig. 1c**). GFAP downregulation appears to precede astrocyte shrinkage, because it was already detected at 10 days, persisting until the age of 4 months (**Supplementary Fig. 1d**).”*

Results, page 16, lines 347-352:

*“To gain insight into the onset and progression of splicing dysregulation throughout brain development and ageing, we studied some selected exons at postnatal day 10 and at 4 months. While the absence of defective astrocyte ramification at P10 (**Supplementary Fig. 1a**), was associated with the nearly complete absence of splicing abnormalities (only *Itga6* was significantly misspliced at this early age) (**Supplementary Fig. 8e**), we found persistent spliceopathy at 4 months of age (**Supplementary Fig. 8f**).”*

Discussion, page 21, lines 449-455:

*“Defective DMSXL astrocyte branching *in vivo* emerges between postnatal day 10 and 1 month, a time window corresponding to the developmental switch from embryonic and adult splicing isoforms. We recently reported that DMSXL astrocytes show increased expression of splicing isoforms typical of immature astroglia⁴¹. These results corroborate the view that DM1 perturbs primarily the splicing of developmentally regulated transcripts⁴², and suggest that astrocyte phenotypes result mainly from the postnatal impairment to express adult splicing isoforms.”*

2) The authors studied the defect in the focal adhesion organization, however, other defects, for example in gap junction proteins, were not considered although they could also explain the altered morphology and polarity. It would be important to better characterize this astrocytic phenotype.

We thank the Reviewer for this suggestion. To better characterize the phenotypes of DMSXL astrocytes, we have performed immunofluorescent detection of gap junction proteins in culture, and quantified the protein levels of their main component (GJA1/Connexin-43) (**Supplementary Fig. 4**). The results did not reveal obvious abnormalities in gap junction organization, nor did we detect altered GJA1 levels in DMSXL astrocytes. The absence of gap junction expression defects is further supported by the gene ontology analysis, which did not hint at dysregulated gap junctions (**Fig. 7**).

Results, page 12, lines 240-246:

*“Finally, we investigated whether gap junctions were affected by the expression of toxic CUG RNA. Gap junctions are indeed specialized structures of direct intercellular communication that crosstalk with the cytoskeleton and focal adhesions, hence affecting cell morphology, adhesion, polarity and migration³². Immunofluorescent detection of GJA1 (or connexin 43), the major connexin in cultured astrocytes³³, did not reveal overt changes in cell-to-cell contacts between primary DMSXL astrocytes (**Supplementary Fig. 4a**), and the total GJA1 protein levels were not altered (**Supplementary Fig. 4b**).”*

Discussion, page 24, lines 520-522:

“Astrocyte gap junctions also control neuronal transmission and behavior⁶². However, we did not find evidence of defective gap junction assembly or abnormal expression of GJA1 in DMSXL astrocytes.”

3) The source of astrocytes for the RNA sequencing analysis is unclear: cortical, and/or hippocampal? Did transcriptomic analysis reveal differences between the cortex and the hippocampus? Do you expect to observe this splicing defect in all astrocytes of the brain? Additional data would be important to start to decipher the mechanism leading to this abnormal adhesion and spreading and further investigate the cellular specificity.

The cells used for RNA sequencing were collected from mouse frontal cortex, given the suggested implication of this brain region in DM1 brain pathology (Meola et al. 2007, Muscle and Nerve, doi: 10.1002/mus.20800; Okkersen et al. 2017, Cortex, doi: 10.1016/j.cortex.2017.08.008). This point has been clarified and we apologize for the careless description.

Results, page 15, lines 317-319:

*“Transcripts from a total of 16,878 genes were detected in primary astrocytes derived from newborn mouse cortex (**Supplementary Table 1**)”*

Results, pages 15, lines 324-326:

*“The same thresholds revealed expression abnormalities in two genes, and splicing defects in 12 transcripts among the 17,089 transcripts identified in primary cortical DMSXL neurons (**Supplementary Table 1**).”*

Methods, page 35, lines 778-779:

“RNA samples were prepared from cortical WT and DMSXL primary astrocytes, grown for 14 days in vitro.”

The heterogeneity between astrocyte subtypes in the brain is an emergent research topic. The Reviewer makes an excellent suggestion, proposing that different astrocyte subpopulations are affected by toxic CUG RNA toxicity to different extents. To gain insight into the influence of toxic CUG RNA on the transcriptome of regional astrocyte populations, we have investigated the splicing profiles of astrocytes acutely isolated from adult frontal cortex, hippocampus and cerebellum (**Supplementary Fig. S9**). Our results demonstrate that overall astroglia respond to toxic CUG RNA repeats across the mouse brain. We found, however, some minor but interesting differences in the extent of the splicing defects of some specific transcripts between individual brain areas, which hint at regional differences in the susceptibility of astroglia to CUG RNA toxicity. These results are discussed in the revised manuscript.

Results, page 17, lines 369-379:

“However, astrocytes are a diverse population of cells displaying brain area-specific properties and functions, driven by intrinsic molecular programs³⁴. We investigated whether CUG RNA toxicity

impacted the transcriptomic regulation of astrocytes in a region-specific manner, through the analysis of alternative splicing in mouse astrocytes acutely isolated from frontal cortex, hippocampus and cerebellum (Supplementary Fig. 9). While Dmd, Itga6, Mpdz and Numa1 exhibited widespread missplicing, other transcripts showed region-specific defects, suggesting regional susceptibility of astrocyte subpopulations to toxic CUG RNA. Sorbs1 exon6 was only significantly affected in frontal cortex astrocytes, Fermt2 abnormalities were detected in both frontal cortex and hippocampus astrocytes, and Capzb was only significantly misspliced in DMSXL cerebellum.

4) One hypothesis to explain the astrocytic defects is that inactivation of MBNL could reduce cell adhesion. Integrating new in vitro experiments on this subject would significantly increase the value of the paper.

We thank the Reviewer for this excellent suggestion, which provided valuable insight into the mechanisms underlying the spliceopathy and the phenotypes of DMSXL phenotypes. To investigate this question we have included an entire new section in our manuscript. We have shown that the knocking down of *Mbnl1* and *Mbnl2* in primary WT astrocytes reduces culture confluence and the number of focal adhesions per cell, in association with the missplicing of the cytoskeleton- and adhesion-related transcripts (Fig. 8). Hence, based on our new data we suggest that altered MBNL protein activity is a key pathogenic event in DM1 astrocyte pathology.

Results, pages 18, lines 384-400:

“MBNL1 and MBNL2 proteins mediate the cell phenotypes and spliceopathy of DMSXL astrocytes

*Given the pronounced reduction in the steady-state levels of MBNL1 and MBNL2 proteins, together with their co-localization with nuclear RNA foci in primary DMSXL astrocytes, we asked if the inactivation of MBNL proteins was sufficient to perturb astrocyte cell growth and adhesion. To answer this question, we used siRNA strategies to knockdown MBNL1 and MBNL2 in primary WT astrocytes (Fig. 8a), as previously described²², and found significantly lower cell confluence (Fig. 8b) and reduced number of focal adhesions per cell (Fig. 8c), when compared to scrambled siRNA controls. To gain insight into the role of MBNL proteins in the splicing regulation of transcripts affected in primary DMSXL astrocytes, we studied the splicing profiles in MBNL double knockdown astrocytes. Overall, reduced MBNL1 and MBNL2 protein levels recreated the splicing abnormalities of primary DMSXL astrocytes, except for *Clasp1* exon 28, which showed a modest change following siRNA treatment (Fig. 8d).*

In conclusion, MBNL proteins regulate the splicing of cell adhesion- and cytoskeleton-related transcripts in primary astrocytes, and their inactivation impacts cell growth and adhesion in culture.”

Other comments:

In the first chapter, the authors, based on unchanged expression of some astrocytic markers, conclude that astrocyte cell density is unchanged. This could be the case, but a more quantitative analysis should be performed to validate this conclusion and exclude confounding factors.

As explained above, we have fully reviewed the analysis of astrocyte density. To this end, we have performed immunofluorescent detection of astrocyte markers. The double detection of proteins highly enriched in adult mouse astrocytes (nuclear SOX9 and cytoplasmic S100B.) allowed us to quantify astrocyte density in mouse cortex and confirm normal cell levels in DMSXL animals (Fig. 1a). This section of the manuscript has been carefully reviewed.

Results, page 6, lines 93-98:

“We first estimated astrocyte cell density in the mouse brain by immunodetection of two astrocyte-specific markers (SOX9, SRY-Box Transcription Factor 9; and S100B, S100 calcium-binding protein B)²³. The percentage of co-labeled cells (SOX9+/S100B+) in the frontal cortex was not significantly different between DMSXL and WT mice at 1 month of age, suggesting unaltered astrocyte density (Fig. 1a).”

The analysis was carried out on one-month-old animals. What about later stages of the disease?

To gain insight into the progression of the astrocyte phenotypes with age, we have performed additional morphological analysis at P10 and 4 months (**Supplementary Fig. 1a-c**) which we complemented with the quantification of the levels of GFAP protein at both ages in the frontal cortex of DMSXL mice. (**Supplementary Fig. 1d**). In addition, we have also included RT-PCR analysis of candidate alternative exons at P10 and 4 months, to demonstrate the postnatal onset of DMSXL spliceopathy and the persisting splicing dysregulation in older animals (**Supplementary Fig. 8e, f**).

Results, page 6, lines 105-111:

*“To explore the progression of astrocyte hypotrophy, we investigated additional mouse ages. While DMSXL astrocyte ramification was unaltered at 10 days of age (**Supplementary Fig. 1a**), it was significantly reduced at 4 months (**Supplementary Fig. 1b**). Interestingly, the reduction in astrocyte processes was more severe at 4 months in processes distanced 10-15 μm from the nucleus (**Supplementary Fig. 1c**). GFAP downregulation appears to precede astrocyte shrinkage, because it was already detected at 10 days, persisting until the age of 4 months (**Supplementary Fig. 1d**).”*

Results, page 16, lines 347-352:

*“To gain insight into the onset and progression of splicing dysregulation throughout brain development and ageing, we studied some selected exons at postnatal day 10 and at 4 months. While the absence of defective astrocyte ramification at P10 (**Supplementary Fig. 1a**), was associated with the nearly complete absence of splicing abnormalities (only *Itga6* was significantly misspliced at this early age) (**Supplementary Fig. 8e**), we found persistent spliceopathy at 4 months of age (**Supplementary Fig. 8f**).”*

Is the phenotype in the cortex and hippocampus similar to the previous described cerebellar abnormalities?

We have previously found pronounced RNA foci accumulation and missplicing in the Bergmann glia of DMSXL cerebellum, relative to neighboring Purkinje cells, but we did not investigate astrocyte morphology in the cerebellum. In the same study, we found hyperexcitability of Purkinje cells in DMSXL cerebellum, mediated by the downregulation of the astrocyte-specific GLT1 glutamate transporter (Sicot et al. 2017 Cell Rep. doi: 10.1016/j.celrep.2017.06.006). More recently, we have reported higher glutamate tonic currents in DMSXL hippocampus, suggestive of higher glutamate ambient levels and neuronal hyperexcitability, in association with GLT1 downregulation (Potier et al. 2022 Int J Mol Sci. doi: 10.1242/jcs.258430). Similarly, we have shown low GLT1 expression and reduced glutamate uptake in DMSXL frontal cortex (Parrot et al. 2021 ACS Chem Neurosci. doi: 10.1021/acchemneuro.1c00634).

In this regard, the electrophysiological phenotypes of DMSXL frontal cortex and hippocampus relate to the abnormalities described in the cerebellum, and seem partially mediated by altered neurotransmitter homeostasis, which relies in part on proper astrocyte function. Some differences may nonetheless exist between brain regions. While in the cerebellum neuronal hyperexcitability is likely dictated by GLT1 downregulation in Bergmann astrocytes that are closely apposed to Purkinje cells (Sicot et al. 2017. doi: 10.1016/j.celrep.2017.06.006), we can speculate that in frontal cortex and hippocampus the effect is exacerbated by reduced ramification and synaptic coverage by neighboring astrocytes.

We hope to have clarified this point with a few considerations that we have included in the final discussion.

Discussion, page 24, lines 530-535.

“We propose that the impaired astrocyte arborization and polarity in DMSXL mice (comparable to that found in other mouse models of astrocyte dysfunction^{24,64}), affects the direct contacts between astrocyte processes and neuronal dendrites in vivo. Defects in the structural and functional interplay between astrocytes and neurons may contribute to the aberrant tonic currents and synaptic plasticity of DMSXL mice⁶⁵, exacerbating neuronal and synaptic dysfunction.”

Supplementary Figure 1: Which regions of the brain are the primary cultures derived from?

Astrocyte and neuron primary cultures were derived from mouse brain cortex. We apologize for the omission. We have clarified the origin of primary cells in the materials and methods and in the legend of Supplementary Fig. 2 (former Supplementary Fig. 1).

Methods, page 29, lines 641-642:

“Primary dissociated cell cultures of cortical astrocytes were prepared from postnatal day 1 WT and DMSXL mouse littermates.”

Methods, page 30, lines 650-651:

“Primary dissociated cell cultures of cortical neurons were prepared from embryonic day 16.5 WT and DMSXL littermates.”

Focal adhesion measurement is central to the paper, a description in the materials and methods section would be essential.

We have described our detailed protocol of focal adhesion detection and quantification in the methods section.

Methods, pages 32-33, lines 709-717:

Focal adhesion detection and quantification

“Focal adhesions formation was quantified as previously described⁷². In short, astrocytes were PFA-fixed 3 h post plating and stained with vinculin, to count the number of focal adhesions. DAPI staining was used to determine the number of cells. Images were taken as Z stacks with a Leica SP8 Confocal Microscope (1024 x 1024; line average = 2; constant PMT parameters). Vinculin-rich clusters were detected after background subtraction, using rolling ball method, gamma adjustment of 1.4. Particles with a greater intensity than the auto threshold, and with an area between 0.1 and 40 μm^2 considered to be focal adhesions. The 40 μm^2 size allowed to exclude perinuclear signal and staining artefacts.”

Page 10, second chapter: what does the term "homogeneous primary cultures" mean?

We understand the confusion and we accept that the term “homogeneous” was inaccurately used to describe primary cell cultures. Although highly enriched for GFAP-expressing astrocytes or MAP2-expressing neurons, these cultures may show low degrees of cross-contamination (**Supplementary Fig. 2**). Furthermore, we did not check for the presence of different astrocyte and neuron subtypes in cell culture. We thank the Reviewer for bringing up this point. We have replaced the term “homogenous” for a more conservative expression, the first time we describe the cell cultures used in our study.

Results, page 8, lines 137-138:

*“Following the characterization of astrocyte phenotypes in vivo, we used primary cell cultures highly enriched for individual cell type (**Supplementary Fig. 2**), [...]”*

Reviewer #3 (Remarks to the Author):

The authors have shown in a transgenic mouse model of DM1 and in human tissues that astrocytes are more affected than neurons. This is an interesting, novel and original finding which could explain the cerebral involvement in DM1, particularly in behavioural aspects, such as "avoidant -Meola et al, *Neuromuscular Disorders*, 2003, Winblad et al, *NMD*, Barris;

We thank Reviewer #3 for recognizing the interest and novelty of our findings, and their implication in the understanding of brain pathology in DM1.

They have shown an alteration of cell biology and a pronounced spliceopathy in astrocytes from frontal and hippocampal regions in DMSX models. They found the same alterations in human frontal cortex regions. Did they investigate in human tissues other regions, such hippocampus? If not they should explain why.

We limited our first analyses to the frontal cortex, given the relevance of this brain region to the cognitive functions primarily affected in DM1 (Meola et al. 2007, *Muscle and Nerve*, doi: 10.1002/mus.20800). Following the Reviewer's suggestion, we have extended analysis of alternative splicing to hippocampus, and found a pronounced spliceopathy in this brain region (Fig. 9), corroborating its role in the brain pathology of DM1.

Introduction, page 3, lines 52-54:

"While the involvement of frontal lobe is suggested by the prevalent executive dysfunction, hippocampus neuropathology may contribute to deficits in visuospatial memory and learning^{6,9}."

Results, pages 18-19, lines 404-420:

"To determine if the insight gained from DMSXL astrocytes is relevant for human disease, we examined post-mortem human frontal cortex and hippocampus. [...] Overall, most exons studied were significantly misspliced in adult DM1 brain samples, compared to non-DM controls: seven out of ten exons were affected in frontal cortex (ITGB4 exon 35 and MPDZ exon 28 showed a clear trend but did not reach statistical significance) (Fig. 9d), whereas all exons studied were misspliced in the hippocampus of DM1 patients (Fig. 9e).

Like in DMSXL mice, the brains of DM1 patients exhibited marked RNA foci accumulation in astrocytes and missplicing of transcripts relevant for cytoskeleton, cell adhesion and morphology in two relevant brain regions: the frontal cortex and hippocampus."

They should mention also some recent neuroimaging studies exploring lesion distribution and white matter damage in DM1 and in comparison to MS, where it is also emphasized the pronounced temporal pole alterations in DM1 (S. Leddy et al, *Neuroimage Clinical*, 29,2021)

In the study recently published by Leddy and colleagues, the authors used imaging techniques to compare the myelination in the brains of DM1 and multiple sclerosis patients, as well as healthy individuals. The authors found convincing evidence of reduced myelin density within the white matter lesions of DM1 brains. These results are extremely interesting, they further support glial cell pathology in DM1, and we discuss them briefly in the final discussion. It is however difficult to directly correlate the reduced myelin density found in white matter lesions with the astrocyte phenotypes reported in our study. Myelin defects may instead reflect defects in oligodendroglia, for which we have also gathered considerable evidence in our mouse model, which we will submit for publication soon. The work of Leddy and colleagues will be more extensively discussed in the context of our future publication.

Discussion, page 25, lines 543-546:

"Recent studies have also found evidence of demyelination in DM1 white matter lesions, particularly in the anterior temporal lobe⁶⁹. The role of myelinating oligodendrocytes in DM1 brain pathology in the brain warrants further investigation."

Reviewer #4 (Remarks to the Author):

The manuscript by Dinca et al. describes novel findings related to the CNS dysfunction in myotonic dystrophy type 1. The authors use a well-established transgenic mouse model of DM1 to demonstrate impaired morphogenesis of astrocytes, the key supporting cells of the CNS. They observe defects in astrocyte ramification, polarization, adhesion, spreading and migration. These findings are well supported by molecular analyses and also by RNA-seq in primary cell cultures, which revealed missplicing of transcripts involved in these processes. Importantly, the Authors verify these alternative splicing events *in vivo* in astrocyte-enriched cell fraction isolated from brains of DMSXL mice, but also in post-mortem brain tissue of DM1 patients. These are novel findings and although DM1-related pathogenic CUG foci have been reported in distinct cell types in the CNS including neurons and astrocytes, the mechanistic contribution of astrocytes to DM1 pathology has not been studied very extensively. Transcriptome and alternative splicing alterations were explored previously in DM1 tissues mainly in skeletal muscle and heart, and also very recently in DM1/DM2 frontal cortex; however, Dinca et al. revealed astrocyte-specific splicing defects which were otherwise undetectable in whole-brain tissue samples, emphasizing the important role of localized RNA toxicity in astrocytes.

Overall, this is an exciting and important topic and these novel data presented by the Authors will be a valuable addition to studies dealing with the pathological mechanisms of DM1 in the CNS. The experimental work is thorough and well documented. The manuscript is well-written and organized and the data is presented with sufficient context and consideration of previous works.

We thank Reviewer #4 for the compliments on the importance, novelty and quality of our experimental and written work.,

I only have a few points which the Authors could address prior to publication, to improve the clarity of the results and the message they want to convey.

1) Figure 1c (Scholl analysis). The authors could have used DM20 mice as an additional control, to exclude the potential effect of *DMPK* overexpression alone on astrocyte morphology and the observed hypotrophy of the cytoskeleton. This comment can be extended to other experiments, for example the ones shown in Figures 3-4. The authors use such control only in Figure 3d, however. Why?

The investigation of additional control mouse lines has certainly strengthened our manuscript, and we thank the Reviewer for this valuable suggestion. We have performed additional experiments in control DM20 mice (already mentioned in our original submission), which express high levels of a shorter *DMPK* human gene with 20 CTG repeats, integrated in a different locus. Importantly, we have recently obtained DM130 mice by spontaneous repeat contraction in the DMSXL mouse line. DM130 mice carry the same human transgene, integrated in the same site, but with a shorter repeat tract of ~130 CTG, when compared with DMSXL mice (> 1000 CTG). The normal astrocyte ramification of DM130 astrocytes *in vivo* (**Supplementary Fig. 1a**) reveals that the low repeat number of these mice is not sufficient to affect astrocyte morphology. Similarly, DM130 astrocytes grow normally in culture (**Fig. 3e**). Molecularly, both DM130 and DM20 mice express normal levels of GFAP protein in the frontal cortex (**Supplementary Fig. 1b, c**) and they do not display splicing defects (**Supplementary Fig. 8d**).

Results, page 7, lines 114-119:

"In contrast, control mice carrying shorter 130 CTG repeats display normal astrocyte ramification (Supplementary Fig. 1e) and GFAP levels in frontal cortex (Supplementary Fig. 1f). Similarly, transgenic DM20 mice, which express higher levels of shorter DMPK transcripts with 20 CTG repeats²⁵ did not show altered GFAP levels (Supplementary Fig. 1g), indicating that astrocyte changes could not be accounted for by DMPK overexpression alone."

Results, page 9, lines 172-176:

"In contrast, control DM20 and DM130 astrocyte cultures expressing shorter DMPK transcripts with a lower number CTG repeats²⁵ exhibited normal growth profiles over time (Fig. 3d, e), indicating that the reduced confluence of DMSXL astrocytes could not be accounted for by DMPK overexpression or the transgene integration site."

Results, page 16, lines 343-346:

"In contrast to DMSXL, control DM20 and DM130 mice did not show splicing defects in transcripts that were markedly affected in DMSXL mouse brain tissue (Supplementary Fig. 8d)."

Methods, page 26, lines 555-560:

“DMSXL and DM130 transgenic mice (>99% C57BL/6 background) carry a 45-kb fragment of human genomic DNA from a DM1 patient¹⁸. The DMSXL mice used in this study carry more than 1500 CTG repeats within the DMPK transgene. DM130 mice result from a spontaneous trinucleotide repeat contraction, and carry a short ~130-CTG tract in the same integration site. Control DM20 mice carry a similar transgene with a non-expanded 20 CTG repeat sequence, derived from a healthy individual²⁵.”

2) Figure 5c (CUG FISH/MBNL IF). This data is not very convincing. The authors claim increased MBNL1&2 co-localization with CUG foci in DMSXL astrocytes, but majority of the signal is in the cytoplasm; there is surprisingly very little staining in the nuclei. Also, from the images shown it looks as if there is less MBNL1&2 signal in astrocytes compared to neurons. The authors might consider providing data for MBNL1-2 protein levels by western blotting and perform Mbnl1&2 alternative splicing analyses (i.e. alternative exons known to be affected in DM1 and determining cytoplasmic vs nuclear localization). If MBNLs sequestration is indeed the main mediator of splicing alterations observed in DMSXL astrocytes (as stated in the discussion, lines 341-345), these additional data could perhaps explain fewer splicing alterations observed in DMSXL neurons. Also, WT controls are missing.

We agree with the Reviewer that the majority of MBNL1 and MBNL2 signal is found in the cytoplasm of primary cells, both neurons and astrocytes (**Fig. 6c**, former Fig. 5c), making the co-localization with RNA foci difficult to detect. Nonetheless, while some MBNL1 and MBNL2 appear to co-localize with CUG RNA foci in the nuclei of DMSXL astrocytes (yellow spots in merge panels), protein co-localization with neuronal RNA foci is less obvious, particularly for MBNL1 (**Fig. 6c**). To gain further insight into the regulation of MBNL proteins in primary cell cultures we performed some additional experiments. The quantification of protein steady-state levels by western blot revealed an intriguing reduction of MBNL1 and MBNL2 only in DMSXL astrocytes (**Fig. 6d**; **Supplementary Fig. 6a, b**). The reduction is not explained by lower transcript levels of *Mbnl1* and *Mbnl2* (**Fig. 6e**) and it is not specific to cell culture, because it is also found in the frontal cortex and hippocampus of DMSXL mice (**Supplementary Fig. 7**).

Results, page 14, lines 289-296:

*“We also found an intriguing reduction in the steady-state levels of MBNL1 and MBNL2 proteins in DMSXL astrocytes by western blot (**Fig. 6d**), which was confirmed with two independent primary antibodies (**Supplementary Fig. 6**) and equally detected in DMSXL frontal cortex and hippocampus (**Supplementary Fig. 7**). In contrast, MBNL protein levels in primary neurons were undistinguishable between genotypes and were significantly lower when compared to DMSXL astrocytes (**Supplementary Fig. 6a, b**). No obvious changes in CELF1 and CELF2 protein levels were found in DMSXL cells (**Fig. 6d**).”*

As suggested by the Reviewer, we studied the splicing of alternative exons of *Mbnl1* and *Mbnl2* reported to affect intracellular protein distribution. The three exons studied revealed pronounced missplicing in DMSXL astrocytes, but remained unaltered in DMSXL neurons (**Fig. 6f**). Finally, we investigated whether *Mbnl1* and *Mbnl2* missplicing in DMSXL astrocytes affected protein distribution between the nucleus and the cytoplasm. We found a mild increase in the nuclear accumulation of MBNL2 in DMSXL astrocytes (**Supplementary Fig. 6d**), in line with the reported nuclear accumulation of *Mbnl1* isoforms that include the analogous exon 7 (Lin et al. 2006 Hum Mol Genet. doi: 10.1093/hmg/ddl132).

Results, pages 14-15, lines 297-308:

*“MBNL protein downregulation could not be attributed to lower transcript levels, since both *Mbnl1* and *Mbnl2* transcripts were surprisingly higher in DMSXL astrocytes (**Fig. 6e**), suggesting complex mechanisms of gene expression. We studied the splicing of regulatory alternative exons of *Mbnl1* and *Mbnl2*³⁵ and found noticeably different splicing profiles between WT astrocytes and neurons. Furthermore, the alternative exons studied were abnormally spliced only in primary DMSXL astrocytes, remaining unaffected in DMSXL neurons, relative to WT control cells (**Fig. 6f**). Since these alternative exons regulate the nuclear localization of MBNL proteins, we studied the impact of missplicing on the intracellular distribution of MBNL1 and MBNL2 between nucleus and cytoplasm in DMSXL astrocytes. We found a significant increase in the nuclear localization of MBNL2 in DMSXL astrocytes relative to WT controls. MBNL1 distribution remained, however, unchanged between genotypes (**Supplementary Fig. 6d**).”*

Methods, page 34, lines 756-759:

“MBNL1 and MBNL2 nucleo-cytoplasmic ratio was quantified on confocal images using nuclear DAPI or MBNL signal to create a mask of the nucleus and cytoplasm respectively, and to measure the MBNL1 and MBNL2 intensity inside the mask.”

Quantitative changes in the expression of MBNL proteins have been previously reported in DM1 muscle cells (Andre et al. 2019 PLoS ONE. doi: 10.1371/journal.pone.0217317), and they are possibly mediated by an imbalanced collection of splice variants in the nucleus and cytoplasm. These new data and hypothesis are now discussed in our revised manuscript.

Discussion, page 21, lines 456-466:

“MBNL protein inactivation in astrocytes is a determinant event behind the coordinated dysregulation of gene sets associated with cytoskeleton and cell adhesion. The lower activity of MBNL results from the combined co-localization with RNA foci and the surprising reduction in total protein levels. The reduction in total MBNL protein levels was previously reported in DM1 muscle cells⁴³, and it is possibly mediated by an imbalanced collection of Mbnl1 and Mbnl2 splicing variants in the nucleus and cytoplasm. DMSXL astrocytes show aberrant Mbnl1 and Mbnl2 splicing of multiple alternative exons, which may also contribute to the functional depletion of MBNL proteins in the nucleoplasm and the missplicing of target transcripts. The regulation of Mbnl1 and Mbnl2 gene expression is complex and multifaceted³⁵, and future studies will clarify the mechanisms behind the intriguing downregulation of MBNL proteins in multiple cell types.”

3) Line 251-252 of the manuscript – concerning RNA seq experiments; which transcripts were significantly affected in DMSXL astrocytes in terms of expression level? How are they relevant to the observed phenotype? There is no further comment in the text, and I assume not every reader will want to dig through extensive supplementary tables to find it out.

We have listed the five transcripts whose expression is significantly altered in DMSXL astrocytes in a new supplementary table, where we have also included a brief description of the function of the encoded proteins (**Supplementary Table 2**). Although, previous studies have not revealed gene functions directly related to the phenotypes of DMSXL astrocytes, the involvement of the dysregulated genes in astrocyte pathology requires further investigation.

Results, page 15, lines 319-320:

*“Stringent selection criteria revealed that the most severe expression changes affected only five transcripts in primary DMSXL astrocytes (**Supplementary Table 2**).”*

Gene	Expression (RPM)		Log ₂ fold change	P _{adjusted}	Protein function
	WT	DMSXL			
Fbxl7 ^a	208.5±16.7	22.8±5.5	-3.2	3.6E-34	Component of a E3 ubiquitin protein complex ¹
Rnf130	2217.0±64.9	1586.0±17.9	1.7	6.4E-06	E3 ubiquitin protein ligase ²
Col15a1	35.2±1.5	110.7±15.4	-0.48	8.5E-09	Structural protein of the extracellular matrix ³
Vsir (4632428N05Rik)	950.0±28.9	712.6±17.2	-0.41	1.2E-03	Immunoregulatory receptor ⁴
Mov10	748.2±28.8	550.2±13.6	-0.44	5.1E-03	RNA helicase. miRNA processing ⁵

^aTransgene integration site. *Fbxl7* gene expression is disrupted by the human transgene.

4) Figure 6&7 – What mediates these splicing changes? The authors show unaltered CELF proteins (Fig. 5d) and claim that MBNLs inactivation by RNA foci (Fig. 5c) is the determinant event behind splicing changes. Again, the authors could consider analyzing alternative splicing of *Mbnl1&2* pre-mRNAs as well as protein levels in DMSXL / wt astrocytes vs neurons.

Please see response to Reviewer #2 (Point 4) and Reviewer #4 (Point 2). We have collected critical new data that clearly demonstrate that the splicing events studied are mediated by the lower activity of MBNL1 and MBNL2 proteins (**Figure 8**).

Results, page 18, lines 384-400:

“MBNL1 and MBNL2 proteins mediate the cell phenotypes and spliceopathy of DMSXL astrocytes

Given the pronounced reduction in the steady-state levels of MBNL1 and MBNL2 proteins, together with their co-localization with nuclear RNA foci in primary DMSXL astrocytes, we asked if the

inactivation of MBNL proteins was sufficient to perturb astrocyte cell growth and adhesion. To answer this question, we used siRNA strategies to knockdown MBNL1 and MBNL2 in primary WT astrocytes (Fig. 8a), as previously described²², and found significantly lower cell confluence (Fig. 8b) and reduced number of focal adhesions per cell (Fig. 8c), when compared to scrambled siRNA controls. To gain insight into the role of MBNL proteins in the splicing regulation of transcripts affected in primary DMSXL astrocytes, we studied the splicing profiles in MBNL double knockdown astrocytes. Overall, reduced MBNL1 and MBNL2 protein levels recreated the splicing abnormalities of primary DMSXL astrocytes, except for Clasp1 exon 28, which showed a modest change following siRNA treatment (Fig. 8d).

In conclusion, MBNL proteins regulate the splicing of cell adhesion- and cytoskeleton-related transcripts in primary astrocytes, and their inactivation impacts cell growth and adhesion in culture.”

5) Discussion - What is the Authors' interpretation of the phenotypes and mechanisms found in DM1 astrocytes: are they causative to the neurological manifestations observed in DM1, whereas defects in DM1 neurons are secondary effects? Stronger point could be made in the discussion.

We now provide convincing evidence that DMSXL astrocytes impact neuritogenesis in co-culture systems (Figure 5). These findings argue in favor of some DM1 neuronal abnormalities being secondary to astrocyte changes, as discussed in the final section of the manuscript. We previously reported neuron-specific abnormalities in chief synaptic vesicles, together with defective neurosecretion in neuronal cell models of DM1 (Hernandez-Hernandez et al. 2013 Brain. doi: 10.1093/brain/aws367). Hence, we hypothesize that the neurological manifestations of DM1 are mediated by a combination of cell-autonomous neuronal dysfunction, which is exacerbated by the defective interplay with astroglia. The relative contribution of neuronal and glial defects towards DM1 brain pathology must be addressed in future mouse models. We have developed this point in the final discussion.

Discussion, page 24, lines 523-535:

Alternatively, since astrocyte-neuron communication is intimately dependent on the specialized morphology of both cell types, aberrant astrocyte ramifications are predicted to be detrimental to neurons. In this context, we showed that DMSXL astrocytes delay neurite growth in culture, but the overall synaptic density is not altered in vivo [...]. We propose that the impaired astrocyte arborization and polarity in DMSXL mice (comparable to that found in other mouse models of astrocyte dysfunction^{24,64}), affects the direct contacts between astrocyte processes and neuronal dendrites in vivo. Defects in the structural and functional interplay between astrocytes and neurons may contribute to the aberrant tonic currents and synaptic plasticity of DMSXL mice⁶⁵, exacerbating neuronal and synaptic dysfunction.”

Reviewers' Comments:

Reviewer #1:

Remarks to the Author:

Myotonic dystrophy RNA toxicity alters morphology, adhesion and migration of mouse and human astrocytes.

Dincă et al.

Myotonic dystrophy type 1 (DM1) is caused by expansion of the CTG trinucleotide in the 3' untranslated region of the *DMPK* gene. RNAs containing expanded CUG triplets accumulate in the nucleus of DM1 cells, forming RNA foci that perturb the localization and function of RNA-binding proteins. Using several transgenic mouse models of DM1, the authors investigate the effects on CNS cells. As expected, major effects are limited to the one mouse model carrying massive expansion of the CTG trinucleotide repeat. Interestingly, Dincă and colleagues report the major CNS effect caused by this expansion is on astrocytes, the major non-neuronal cell type in the brain. Astrocytes show impaired ramification and polarization *in vivo* which appears during post-natal development, as well as defects in adhesion, spreading and migration *in vitro*. Transcriptome analysis revealed mis-splicing of transcripts (across multiple brain regions) regulating adhesion, cytoskeleton and morphogenesis, consistent with impaired maturation from a juvenile to adult state. The authors also report these effects were seen in human DM1 cells. These effects are hypothesized to be linked to accumulation of toxic RNA foci in the nucleus, which sequester key splicing factors (knock down of which phenocopies the DM1 phenotype). Mutant astrocytes in culture also fail to promote neuronal neurite outgrowth, although synaptic density *in vivo* is unaffected. Hence, the authors conclude that DM1 impacts astrocyte biology and by extension presumably compromises their ability to support and regulate synaptic function. The idea that astrocytes play a major (if not the dominant) role in CNS disease is an interesting idea and is increasingly seen as a 'hot topic' (for example, see the recent paper on astrocytes and ASD by Allen et al., *Mol Psychiatry*, 2022). Taking account of the contribution of astrocytes to disease initiation and/or progression is essential if we are ever to find treatments for severe (currently untreatable) CNS diseases.

This is a resubmitted manuscript. During the revision period, the authors have added a large amount of data which substantially strengthens their arguments and satisfactorily answers my critiques from the first round of review.

Most of my comments are now aimed at trying to help the authors improve the manuscript prior to publication.

Major comments:

(i) I find it intriguing that the authors report effects on astrocyte morphology/orientation and function (promotion of neuronal neurite outgrowth), which they speculate 'affects the direct contacts between astrocyte processes and neuronal dendrites *in vivo*' (Page 24). How do they reconcile this to the fact that both the density of excitatory and inhibitory synapses appears unchanged? This apparent contradiction deserves comment in the manuscript.

(ii) Astrocyte heterogeneity is not solely driven by intrinsic molecular programs. There is now increasing evidence that local neuronal activity plays a role in astrocyte development (reviewed in Farmer and Murai, *Front Cell Neurosci*, 2017; Pestana et al., *Brain Sci*, 2020).

(iii) When assessing the RNA foci in mutant mouse astrocytes and neurons there appears a disconnect between the numbers shown in the graph and the images (Fig. 6b). Representative images (including more cells) would considerably strengthen the authors' point. This also holds for the work looking in human tissue (Fig. 9; Supplementary Fig. 10).

Minor comments:

(i) There seems to be a disconnect between the data shown in graphs and their description in the main text. For example, I think five genes show significant differences in splicing in frontal cortex not four (Fig. 7c), while mis-splicing in astrocytes occurs in ten out of the eleven genes assayed (Fig. 7e) not nine. *ITGB4* exon 35 does appear mis-spliced (Fig. 9d) etc etc. Please check

the text and correct, if necessary.

(ii) In Figures with Western Blot data, what exactly is being shown in the 'Total Protein' panel? Is this the region of the blot subsequently used for antibody staining? Likewise, are 'Frontal Cortex' and 'Hippocampal' samples whole tissue lysates? Can this be explicitly stated for purposes of clarity.

(iii) The authors claim to show 'that the DM1 repeat expansion impacts critical structural and functional features of astrocytes, such as cell morphology, adhesion and orientation'. This could be a bit strong considering that the majority of such work is, in fact, performed in culture - a situation which the authors acknowledge in their rebuttal can cause issues. I think this claim could/should be toned down. Likewise, the authors refer to 'human glial cells in culture' when in fact they use an immortalized cell line. Furthermore, I could not find details of how this line is maintained or toxic RNA production induced by deoxycycline. In my opinion, the 'Methods' are generally poor (in comparison to the rest of the manuscript) and need work before publication (see also below).

(iv) Given the (large) numbers of astrocytes in white matter, I would suggest the *relative* role of myelinating oligodendrocytes in DM1 brain pathology warrants further investigation (Page 25).

(v) Fig. 8: why are non-transfected controls present in some panels and missing in others?

(vi) The figure legend for Supplementary Fig. 5 appears incomplete.

(vii) In general, I found the 'Methods' section to be lacking and not up to the standards of the rest of the manuscript. I think the authors need to go through and spend more time on this section. Below are examples. The list is not exhaustive:

- 5' and 3' ends of primers are usually indicated. What method was used for quantitation in qPCR: ddCt?

- How was the astrocyte nucleus found for the *in vivo* morphological analysis?

- Were focal adhesions (Fig. 4) really detected and quantified by staining with vinculin or an anti-vinculin antibody? I think 'VCL' in Supplementary Table 4 needs to be defined. In general, details of the IHC protocols used are lacking: descriptions of permeabilization conditions, acquisition parameters (including microscope and objective used (magnification and numerical aperture), microscope control software) and post-acquisition analysis packages etc.

- Are images in Fig. 5a and Fig. 5b taken from the same cultures? If not, how was neurite outgrowth assessed in Fig5a? Phase contrast microscopy? Use of fiduciary markers to indicate neurite outgrowth (and possibly higher magnification images) would be useful for interpretation.

- What are independent cultures?

- How was the increase in nuclear localization of MBNL2 determined in Supplementary Fig. 6d? Information is lacking in the text and figure legend.

- There are multiple types of Triton with distinct chemical properties: I assume the authors used Triton X-100 for cell permeabilization - but this should be stated.

- The section on 'Acute isolation of astrocytes from mouse brain' seems added into the text at random and, in my opinion, would best be placed earlier, alongside the description of 'Primary Cultures'

Very minor comment: please amend 'S100B' to 'S100β'.

Reviewer #2:

Remarks to the Author:

The authors have responded appropriately to the questions raised by the reviewers and the quality of the manuscript is significantly improved.

Reviewer #3:

Remarks to the Author:

The authors in this original and innovative work have given an enormous contribution in the field to understand the cerebral involvement in myotonic dystrophy type 1.

The methods are now sound and all my points raised have been addressed.

Dr Giovanni Meola

Reviewer #4:

Remarks to the Author:

Thank you for the opportunity to review this revised manuscript.

My main concerns from the first review were addressed satisfactorily. Additional control mouse lines strengthened the data by ruling out the potential effect of DMPK overexpression alone on astrocyte morphology. New data included analyses of the expression and splicing of MBNL 1/2, evidently showing that the phenotypes and splicing alterations in DMSXL astrocytes are mediated by lower activity of these proteins. This is nicely supported by siRNA-mediated knock-down of MBNL1/2 in primary astrocytes, which recapitulated spliceopathy of DMSXL astrocytes. However, I am intrigued by the results showing discrepancy in Mbnl1/2 RNA and protein abundance (i.e. reduced MBNL1/2 protein levels accompanied by increased Mbnl1/2 RNA) in DMSXL vs WT astrocytes. Could these results be linked to reduced translation or protein stability in DMSXL astrocytes? In the case of MBNL1, alternative splicing of the first coding exon has been shown to significantly affect protein stability and activity; I wonder if the Authors also looked into that. Perhaps the Authors could try to interpret or speculate a bit more on these interesting results in the discussion.

Overall, this is a comprehensive study that puts forward novel hypotheses and provides substantial experimental evidence to support it. The in vivo relevance is also clear. I enjoyed reading it and I have no doubt it will be appreciated by Nature Communications readers.

“Myotonic dystrophy RNA toxicity alters morphology, adhesion and migration of mouse and human astrocytes”

Point-by-Point Response to Reviewers

Reviewer #1 (Remarks to the Author):

**Myotonic dystrophy RNA toxicity alters morphology, adhesion and migration of mouse and human astrocytes.
Dincă et al.**

Myotonic dystrophy type 1 (DM1) is caused by expansion of the CTG trinucleotide in the 3' untranslated region of the *DMPK* gene. RNAs containing expanded CUG triplets accumulate in the nucleus of DM1 cells, forming RNA foci that perturb the localization and function of RNA-binding proteins. Using several transgenic mouse models of DM1, the authors investigate the effects on CNS cells. As expected, major effects are limited to the one mouse model carrying massive expansion of the CTG trinucleotide repeat. Interestingly, Dincă and colleagues report the major CNS effect caused by this expansion is on astrocytes, the major non-neuronal cell type in the brain. Astrocytes show impaired ramification and polarization *in vivo* which appears during post-natal development, as well as defects in adhesion, spreading and migration *in vitro*. Transcriptome analysis revealed mis-splicing of transcripts (across multiple brain regions) regulating adhesion, cytoskeleton and morphogenesis, consistent with impaired maturation from a juvenile to adult state. The authors also report these effects were seen in human DM1 cells. These effects are hypothesized to be linked to accumulation of toxic RNA foci in the nucleus, which sequester key splicing factors (knock down of which phenocopies the DM1 phenotype). Mutant astrocytes in culture also fail to promote neuronal neurite outgrowth, although synaptic density *in vivo* is unaffected. Hence, the authors conclude that DM1 impacts astrocyte biology and by extension presumably compromises their ability to support and regulate synaptic function. The idea that astrocytes play a major (if not the dominant) role in CNS disease is an interesting idea and is increasingly seen as a 'hot topic' (for example, see the recent paper on astrocytes and ASD by Allen et al., *Mol Psychiatry*, 2022). Taking account of the contribution of astrocytes to disease initiation and/or progression is essential if we are ever to find treatments for severe (currently untreatable) CNS diseases.

This is a resubmitted manuscript. During the revision period, the authors have added a large amount of data which substantially strengthens their arguments and satisfactorily answers my critiques from the first round of review.

Most of my comments are now aimed at trying to help the authors improve the manuscript prior to publication.

Major comments:

(i) I find it intriguing that the authors report effects on astrocyte morphology/orientation and function (promotion of neuronal neurite outgrowth), which they speculate 'affects the direct contacts between astrocyte processes and neuronal dendrites *in vivo*' (Page 24). How do they reconcile this to the fact that both the density of excitatory and inhibitory synapses appears unchanged? This apparent contradiction deserves comment in the manuscript.

We agree that our observations may seem surprising at first glance. It is known that astrocytes regulate synaptogenesis, and as such, one could expect that the astrocyte abnormalities found in DMSXL mouse brains resulted in a reduced number of synapses. Despite this, we did not detect noticeable alterations in adult DMSXL frontal cortex, when synaptic density was estimated by co-localization of pre- and post-synaptic proteins.

Given the multiple aspects governed of by astrocytes, it is conceivable that the reduced astrocyte ramification and impaired reorientation of DMSXL astrocytes may impact primarily synaptic maturation and function, in the absence of obvious changes in the number of synapses. In support of this idea, we have gathered evidence of altered synaptic function in DMSXL animals (doi:

10.3390/ijms23020592), as discussed in the manuscript. Of course, it will be important to complement these studies with the ultrastructural investigation of the tripartite synapse, including the distance between astrocyte processes and the synapses, as well as the analysis of dendritic morphology and arborization *in vivo*. Dual recordings of neuronal and astrocyte activity will further evaluate functional neuroglial interactions in the mouse brain.

At present, and based on our current results we can only speculate on these aspects. In order to address the Reviewer's concern and to avoid an apparent discrepancy in our results, we have toned down our conclusions, and included some additional considerations in the final discussion.

Page 24-25, lines 535-541:

"We propose that impaired astrocyte arborization and polarity in DMSXL mice (comparable to that found in other mouse models of astrocyte dysfunction^{24,68}), may affect the ultrastructure and maturation of the synapse in vivo, in terms of astroglial coverage, synapse-to-astrocyte distance and dendritic arborization. In this context, it will important to investigate defects in the ultrastructural and functional interplay between astrocytes and neurons, which may contribute to the aberrant tonic currents and synaptic plasticity of DMSXL mice⁶⁹."

(ii) Astrocyte heterogeneity is not solely driven by intrinsic molecular programs. There is now increasing evidence that local neuronal activity plays a role in astrocyte development (reviewed in Farmer and Murai, *Front Cell Neurosci*, 2017; Pestana et al., *Brain Sci*, 2020).

We appreciate the Reviewer's feedback. The original sentence has been changed to present the idea that astrocyte heterogeneity is also influenced by external factors.

Page 17, lines 371-373:

"However, astrocytes are a diverse population of cells displaying brain area-specific properties and functions³⁴, driven by intrinsic molecular programs and context cues provided by neighboring cells^{37,38}."

(iii) When assessing the RNA foci in mutant mouse astrocytes and neurons there appears a disconnect between the numbers shown in the graph and the images (Fig. 6b). Representative images (including more cells) would considerably strengthen the authors' point. This also holds for the work looking in human tissue (Fig. 9; Supplementary Fig. 10).

To provide a more accurate illustration of the average number of foci per individual nucleus, we have included additional images, showing a higher number of mouse primary cells in Supplementary Fig. 2, and human brain cells in Supplementary Fig. 10. We have opted to show a higher number of individual cells imaged at higher magnification, because nuclear RNA foci accumulation is difficult to be appreciated in lower magnification images. We thank the Reviewer for this suggestion, which we believe has provided strong visual support to our quantitative analysis.

Supplementary Fig. 2, legend:

"(c) FISH detection of RNA foci in primary DMSXL mouse brain cells, illustrating a higher number of nuclear aggregates in GFAP-expressing astrocytes than in MAP2-expressing neurons. Scale bar, 10 μm."

Supplementary Fig. 10, legend:

"(b) RNA FISH combined with protein immunofluorescence, illustrating greater foci accumulation in GFAP-expressing-astrocytes, relative to RBFOX3/NeuN-expressing neurons."

Minor comments:

(i) There seems to be a disconnect between the data shown in graphs and their description in the main text. For example, I think five genes show significant differences in splicing in frontal cortex not four (Fig. 7c), while mis-splicing in astrocytes occurs in ten out of the eleven genes assayed (Fig. 7e) not nine. *ITGB4* exon 35 does appear mis-spliced (Fig. 9d) etc etc. Please check the text and correct, if necessary.

We thank the Reviewer for spotting these discrepancies between the main text and the statistical comparisons shown in the figure. We apologise for these mistakes. We have carefully reviewed the statistical analyses and main text, and we have corrected all remaining inconsistencies.

(ii) In Figures with Western Blot data, what exactly is being shown in the 'Total Protein' panel? Is this the region of the blot subsequently used for antibody staining? Likewise, are 'Frontal Cortex' and 'Hippocampal' samples whole tissue lysates? Can this be explicitly stated for purposes of clarity.

The "Total Protein" panels in the figures show the most intense and representative bands produced by stain free gels (full membrane blots are shown in the supplementary file that accompanies the manuscript, which contains the original and uncropped gel and membrane images). Stain-free methods allow for the quantitation of total proteins on the same blot as the proteins of interest, without the need for stripping and reprobing the membrane. Total protein normalization provides accurate protein quantitation and western blotting results (Guther et al. 2013 doi: 10.1016/j.ab.2012.10.010).

Frontal Cortex and Hippocampal samples correspond to tissue lysates prepared from these brain regions dissected from mouse brains, or collected from *post-mortem* human brain material.

We have clarified both points in the Methods section and in figure legends, as detailed below.

Methods, page 28, lines 606-607:

"Proteins from primary cells, whole mouse frontal cortex and hippocampus, and human frontal cortex collected at post-mortem were extracted using RIPA buffer (...)"

Methods, page 28, lines 613-616:

"After electrophoresis, gels were activated for 2 minutes under UV light, proteins were transferred onto Nitrocellulose membranes using Trans-Blot® Transfer System (Bio-Rad) and total protein on the membrane was imaged using the ChemiDoc Imaging System (Bio-Rad) as used for protein normalization ⁷⁵."

Multiple western blot figure legends:

"Representative stain-free protein bands are shown to illustrate total protein loading."

Fig. 1 legend, page 53, lines 1148-1149:

"Western blot quantification of GFAP in whole frontal cortex and hippocampus tissue lysates from DMSXL mice at 1 month."

Fig. 9 legend, page 63, lines 1320-1321:

"Western blot quantification of GFAP in tissue lysates prepared from post-mortem human frontal cortex"

Supplementary Fig. 1 legend:

"(d) Quantification of GFAP expression in whole frontal cortex tissue lysates of control and DMSXL mice at 10 days and 4 months of age. (...) (f) Quantification of GFAP expression in whole frontal cortex tissue lysates of control DM130 mice at 1 month, (...) (g) Quantification of GFAP expression in whole frontal cortex tissue lysates of control DM20 mice, at 1 month."

Supplementary Fig. 7 legend:

(a) Quantification of MBNL1 and MBNL2 protein levels in whole mouse frontal cortex tissue lysates at 1 month capillary basis electrophoresis, using antibodies against recombinant full length human proteins. (...) (b) Quantification of MBNL1 and MBNL2 protein levels in whole mouse hippocampus tissue lysates at 1 month by capillary basis electrophoresis, using antibodies raised against recombinant full length human proteins."

(iii) The authors claim to show 'that the DM1 repeat expansion impacts critical structural and functional features of astrocytes, such as cell morphology, adhesion and orientation'. This could be a

bit strong considering that the majority of such work is, in fact, performed in culture - a situation which the authors acknowledge in their rebuttal can cause issues. I think this claim could/should be toned down. Likewise, the authors refer to 'human glial cells in culture' when in fact they use an immortalized cell line. Furthermore, I could not find details of how this line is maintained or toxic RNA production induced by doxycycline. In my opinion, the 'Methods' are generally poor (in comparison to the rest of the manuscript) and need work before publication (see also below).

To avoid overstatements and misinterpretation of our data, we have reformulated the final discussion to specify the nature of the model system used in our study.

Results, page 9, lines 183-184:

"We used an immortalized glial cell line to validate the impact of toxic RNA on the adhesion and spreading of a human cell model of DM1."

Results, page 9, lines 195-196:

"In summary, DM1 repeat expansion affects the adhesion, spreading and morphogenesis of mouse astrocytes and immortalized human glia cells."

Discussion, page 20, lines 429-431:

"We showed that the DM1 repeat expansion impacts critical structural and functional features of astrocytes, such as cell morphology and orientation in the mouse brain, as well as adhesion in cell culture models of the disease."

Concerning the MIO-M1 cells, the protocols used for the establishment and maintenance of this cell line, as well as the conditions for transgene induction have published in the meantime by our collaborators (doi: 10.3390/biom11020159). We have reviewed the Methods section: we now cite the original reference and provide a brief description of the cell culture protocol.

Methods, pages 30-31, lines 674-678:

"MIO-M1 cell line

The generation and maintenance of the MIO-M1 cell model has been described elsewhere ²⁹. Briefly, MIO-M1 cells were cultured in DMEM supplemented with 10% FBS, 100 U/mL penicillin and 100 µg/mL streptomycin. Transgene expression was induced by 1 µg/mL doxycycline added to the culture medium for 3 days prior to analysis."

(iv) Given the (large) numbers of astrocytes in white matter, I would suggest the *relative* role of myelinating oligodendrocytes in DM1 brain pathology warrants further investigation (Page 25).

We thank the Reviewer for this recommendation. The text has been modified, as suggested.

Discussion, page 25, lines 551-552:

"The relative role of myelinating oligodendrocytes in DM1 brain pathology in the brain warrants further investigation."

(v) Fig. 8: why are non-transfected controls present in some panels and missing in others?

We previously omitted the confluence measurements of non-transfected cells from the IncuCyte growth profiles for clarity reasons, since the profiles overlap extensively with those obtained with scrambled-treated cells. To be consistent throughout the we now display the results of non-transfected cells in all the panels. The figure and legend have been modified accordingly.

Figure 8 legend, page 62, lines 1301-1303:

"Primary MBNL1/MBNL2 knockdown astrocytes are compared with non-transfected and scrambled-transfected controls."

(vi) The figure legend for Supplementary Fig. 5 appears incomplete.

We thank the reviewer for spotting the mistake and we apologize for the truncated legend. The missing text has been added.

Supplementary Fig. 5, legend:

“High magnification insets show co-localization analysis (white arrowheads) performed on binary images (masks). Scale bar, 5 μm.”

(vii) In general, I found the ‘Methods’ section to be lacking and not up to the standards of the rest of the manuscript. I think the authors need to go through and spend more time on this section. Below are examples. The list is not exhaustive:

- 5’ and 3’ ends of primers are usually indicated. What method was used for quantitation in qPCR: ddCt?

5’ and 3’ end of oligonucleotide primer sequences are now indicated in Supplementary Tables 5, 6 and 7.

The qPCR quantification of *DMPK* and *Dmpk* transcripts was performed as previously described by our group (doi: 10.1371/journal.pgen.1003043). We used relative quantification normalized against a reference gene (*18S rRNA*). The experimental details are now described in the Methods section.

Methods, page 36, lines 845-848:

“Samples were quantified in triplicate and experiments were repeated at least twice. 18S rRNA was used as internal control. DMPK, Dmpk and 18S mRNA levels were interpolated from a linear range of serial diluted cDNA standards. The DMPK/18S and Dmpk/18S expression ratios were normalized using the same standard sample.”

- How was the astrocyte nucleus found for the *in vivo* morphological analysis?

The localization of astrocyte nucleus was determined using DAPI staining of genomic DNA. This point has been clarified in the Methods section.

Methods, page 27, lines 600-603:

“The ImageJ plugin “Sholl analysis”⁷³ was used to measure the number of intersections between GFP stained processes and concentric circles spaced by 5 μm and centered on astrocyte nucleus, visualized by DAPI staining.”

- Were focal adhesions (Fig. 4) really detected and quantified by staining with vinculin or an anti-vinculin antibody? I think ‘VCL’ in Supplementary Table 4 needs to be defined. In general, details of the IHC protocols used are lacking: descriptions of permeabilization conditions, acquisition parameters (including microscope and objective used (magnification and numerical aperture), microscope control software) and post-acquisition analysis packages etc.

We used anti-vinculin antibodies to stain vinculin-rich clusters. We have clarified this point in the Methods section and defined VCL in Supplementary Table 4.

Methods, page 33, lines 783-785:

“In short, astrocytes were PFA-fixed 3 h post plating and stained with anti-vinculin antibodies, to count the number of focal adhesions.”

We have detailed the immunofluorescence protocols and included the acquisition parameters, objectives used and post-acquisition software analysis packages.

Methods, page 33, lines 736-771:

“Immunofluorescence

Cultured cells were fixed in 4% PFA for 15 min at room temperature, washed 3 x 10 min in 1X PBS and incubated in blocking and permeabilization solution for 1 h at room temperature (1X PBS; 0.2% Triton X-100; 10% normal goat serum, Sigma, G6767). Primary antibody was diluted in blocking and permeabilization and incubated over night at 4°C. Primary antibodies and dilutions are indicated in **Supplementary Table 4**. Following 3 washes in 1X PBS, cells were incubated with secondary antibody diluted in the blocking and permeabilization solution for 1 h at room temperature. Excess antibody was washed 3 x 10 min in 1X PBS, and incubated with 0.0002% DAPI (Sigma, 10236276001) for 15 min at room temperature. Stained cells were finally washed 3 x 10 min in 1X PBS and mounted with Vectashield® Mounting Medium (Eurobio Scientific, H-100) prior to observation. Images were acquired with a laser scanning confocal microscope (SP8, Leica), using a 20X/0.8NA or 60X/1.4NA objective and Leica Application Suite X software.

Fluorescent in situ hybridization (FISH) and immunofluorescence

Primary cells cultured for 2 weeks were washed 3 x 2 min in 1X PBS and then fixed for 15 min in 4% PFA (VWR; J61899.AP) at room temperature. Cryostat tissue sections (10 µm) were directly fixed in 4% PFA. Following 5 x 2 min washes in 1X PBS, cells and tissues were permeabilized in 2% ice-cold acetone in 1X PBS for 5 min at room temperature. Prehybridization was carried out in 30% formamide/2X SSC for 10 min at room temperature. Hybridization with 1 ng/mL 5'-Cy3-labelled (CAG)₅ PNA probe in hybridization buffer (30% formamide, 2X SSC, 0.02%BSA, 66 mg/ml yeast tRNA, 2 mM vanadyl complex) was carried out for 2 h at 37°C in a dark humidified chamber. Next, cells/tissues were washed for 30 min in 30% formamide/2X SSC at 50°C, followed by one last wash of an additional 30 min in 1X SSC at room temperature. Prior to visualization, cell nuclei were stained with 0.0002% DAPI (Sigma, 10236276001) for 15 min at room temperature. Stained samples were finally washed 3 x 10 min in 1X PBS and mounted with Vectashield® Mounting Medium (Eurobio Scientific, H-100) prior to observation. RNA foci were counted in 3D stacks using the Spot Detector plugin of the ICY bioimageanalysis open-source program (<http://icy.bioimageanalysis.org>). Immunofluorescence (IF) combined with fluorescent in situ hybridization (FISH) was performed as previously described¹⁴. When combined with immunofluorescence, cell and tissue samples were directly incubated in primary antibody following the 1X SSC post-hybridization wash. Primary antibodies and dilutions are indicated in **Supplementary Table 4**. Washes and secondary antibody incubation were performed as described above. Images were acquired with a laser scanning confocal microscope (SP8, Leica), using 60X/1.4NA objective and Leica Application Suite X software.”

- Are images in Fig. 5a and Fig. 5b taken from the same cultures? If not, how was neurite outgrowth assessed in Fig5a? Phase contrast microscopy? Use of fiduciary markers to indicate neurite outgrowth (and possibly higher magnification images) would be useful for interpretation.

Fig. 5a and Fig 5b were not taken from the same culture: while Fig. 5a shows primary neurons cultured alone, Fig. 5b depicts fluorescent neurites of neurons co-cultured with unlabelled astrocytes. We apologize for the confusion. We have now clarified the nature of the different cultures and labelling methods used, and we have further detailed the experimental protocol to quantify neurite outgrowth. Since these methods have been extensively described in a previous technical publication of our laboratory (doi: 10.1007/978-1-4939-9784-8_14), we provide a brief summary in the Methods section.

Figure 5, page 58, lines 1224-1228:

“(a) Representative brightfield images of unlabeled DMSXL and WT neurons at 4 and 7 DIV, showing neurite arborization in culture. Fire pseudocolor was applied on phase contrast images for clearer contrast. Scale bar, 50 µm. Semi-automated, label-free quantification of neurite growth rate and neurite length over 7 DIV.”

Methods, pages 35-36, lines 793-809:

“Imaging of neurite growth in primary neurons and neuron-astrocyte co-cultures

Primary neurons were plated at 90,000 cells/cm² on 96-well plates and maintained for 7 DIV. Images acquired every 3 h for a total of 7 DIV, using a 20X/0.45 objective and the IncuCyte base analysis software Semiautomated, label-free monitoring of neuritogenesis in primary neurons growing alone was performed using the NeuroTrack module of the IncuCyte Zoom video-microscope on phase contrast images, as previously described²⁸. Phase masks were used to contour the cell bodies and to trace neurites. The length of neurites was automatically measured and plotted overtime. Co-cultures of neurons and astrocytes were established as previously described²². Briefly, primary neurons were plated at 90,000 cells/cm² and infected 4 h after plating (MOI=3) with NeuroLight Red Lentivirus (Essen BioScience, 4584), encoding the mKate2 fluorescent protein, expressed under the Synapsin 1 promoter, in serum-free Neuronal medium. The following day neurons were washed with Neuronal medium. Mouse primary astrocytes, cultured 2 weeks prior to neurons, were plated on top at a density of 150,000 cells/cm². Fluorescent images of co-cultures were acquired every 3 h for a total of 7 DIV, using a 20X/0.45 objective and the IncuCyte base analysis software. mKate2 expression was detected 4 days after transfection. NeuroTrack analysis module of the IncuCyte Zoom video-microscope was used to quantify neurite length and arborization from 4 DIV onwards.”

- What are independent cultures?

We used the term independent cultures to refer to primary cultures derived from different animals (true biological replicates, as opposed to replicates resulting from the splitting of the same original culture). This point has been explained in the figure legend, where the term appears for the first time, and more generally in the Methods section.

Figure 2, page 54, lines 1167-1168:

“Data are mean ± SEM, n = 3 - 6 independent cultures per group, each one established from a different animal”

Methods, pages 29-30, lines 654-656:

“Independent cultures (biological replicates) were established from different animals. Corresponding WT control cultures were established from the same litters as DMSXL cultures.”

- How was the increase in nuclear localization of MBNL2 determined in Supplementary Fig. 6d? Information is lacking in the text and figure legend.

The nucleus to cytoplasmic ratio of MBNL1 and MBNL2 protein was quantified by the immunofluorescence analysis of confocal images. We have added the information to the figure legend and main text. In addition, the protocol used is now described in the Methods section.

Results, page 14, lines 307-308:

“We found a significant increase in the nuclear to cytoplasmic ratio of MBNL2 immunofluorescence in DMSXL astrocytes relative to WT controls.”

Supplementary Fig. 6 legend:

“(d) Immunofluorescence analysis of nucleus to cytoplasmic ratio of MBNL1 and MBNL2 proteins in primary astrocytes.”

Methods, page 35, lines 773-780:

“MBNL1 and MBNL2 nucleocytoplasmic distribution

Following MBNL1 or MBNL2 immunofluorescence, the mean fluorescence intensity ratio of nuclear and cytoplasmic protein was quantified on confocal images, using the “Intensity Ratio Nuclei Cytoplasm Tool” macro on Fiji Software (Intensity Ratio Nuclei Cytoplasm Tool, RRID:SCR_018573). In short, the tool first used a thresholding method in the nucleus channel for nucleus segmentation. The background intensity was then adjusted and the nuclear signal was removed to measure MBNL1 and MBNL2 signal intensity in the cytoplasm/green channel.”

- There are multiple types of Triton with distinct chemical properties: I assume the authors used Triton X-100 for cell permeabilization – but this should be stated.

The Reviewer's assumption is correct: we have used Triton X-100. This point has now been specified in the Methods section.

Methods, page 36, lines 736-739:

“Cultured cells were fixed in 4% PFA for 15 min at room temperature, washed 3 x 10 min in 1X PBS and incubated in blocking and permeabilization solution for 1 h at room temperature (1X PBS; 0.2% Triton X-100; 10% normal goat serum, Sigma, G6767).”

Methods, page 36, lines 813-814:

“Briefly, 40- μ m mouse brain slices were blocked with PBS□Gelatine□Triton X-100 (1X PBS, 0.002% gelatine, 0.25% Triton X-100) for 1 h at room temperature, (...)”

- The section on 'Acute isolation of astrocytes from mouse brain' seems added into the text at random and, in my opinion, would best be placed earlier, alongside the description of 'Primary Cultures'

We thank the Reviewer for the suggestion. This section has been moved, together with the supplementary table associated.

Very minor comment: please amend 'S100B' to 'S100 β '.

We thank the reviewer for bringing up this point. S100B is the Human Gene Nomenclature Committee approved name for this gene and protein, and we believe it would be incorrect for us to continue to promote the unofficial name. We have however referred the traditional S100 β alias the first time we mention the protein in the main text.

Results, page 6, lines 93-96:

“We first estimated astrocyte cell density in the mouse brain by immunodetection of two astrocyte-specific markers: SOX9, SRY-Box Transcription Factor 9; and S100B, S100 calcium-binding protein B (traditionally known as S100 β)²³.”

Reviewer #2 (Remarks to the Author):

The authors have responded appropriately to the questions raised by the reviewers and the quality of the manuscript is significantly improved.

We thank Reviewer #2 for considering that the manuscript quality has been significantly improved and that the previous concerns have now been addressed.

Reviewer #3 (Remarks to the Author):

The authors in this original and innovative work have given an enormous contribution in the field to understand the cerebral involvement in myotonic dystrophy type 1.

The methods are now sound and all my points raised have been addressed.

Dr Giovanni Meola

We are grateful to Reviewer #3 for their very positive evaluation of our work, and for considering that the points previously raised have been properly addressed.

Reviewer #4 (Remarks to the Author):

Thank you for the opportunity to review this revised manuscript.

My main concerns from the first review were addressed satisfactorily. Additional control mouse lines strengthened the data by ruling out the potential effect of DMPK overexpression alone on astrocyte morphology. New data included analyses of the expression and splicing of MBNL 1/2, evidently showing that the phenotypes and splicing alterations in DMSXL astrocytes are mediated by lower activity of these proteins. This is nicely supported by siRNA-mediated knock-down of MBNL1/2 in primary astrocytes, which recapitulated spliceopathy of DMSXL astrocytes. However, I am intrigued by the results showing discrepancy in *Mbnl1/2* RNA and protein abundance (i.e. reduced MBNL1/2 protein levels accompanied by increased *Mbnl1/2* RNA) in DMSXL vs WT astrocytes. Could these results be linked to reduced translation or protein stability in DMSXL astrocytes? In the case of MBNL1, alternative splicing of the first coding exon has been shown to significantly affect protein stability and activity; I wonder if the Authors also looked into that. Perhaps the Authors could try to interpret or speculate a bit more on these interesting results in the discussion.

Reviewer #4 highlights a very intriguing finding: the unexpected protein downregulation of MBNL1 and MBNL2 in DMSXL astrocytes, which cannot be explained by lower transcript levels. We know today that the expression of MBNL paralogs is governed by complex mechanisms, which involve shifts in transcription initiation sites, alternative splicing and translation initiation, as well as changes in miRNA and circRNA species (doi.org/10.1080/15476286.2017.1384119). This multifaceted mechanism provides key autoregulatory feedback loops that maintain proper splicing activity of MBNL proteins. Although we have not explored the mechanisms of MBNL downregulation, it is tempting to speculate that DMSXL astrocytes exhibit altered *Mbnl1* and *Mbnl2* transcription initiation and/or exon 1 skipping, which shift translation initiation and produce unstable protein isoforms, and which might be accompanied by increased transcript levels, as demonstrated in DM1 skeletal muscle and cell culture by the Sobczak's lab ([doi: 10.1093/nar/gkw1158](https://doi.org/10.1093/nar/gkw1158)). It is our plan to investigate these hypotheses, other potential contributors for MBNL protein downregulation in the future, including changes in miRNA translational repressors of MBNL1 and MBNL2 ([10.1038/S41467-018-04892-4](https://doi.org/10.1038/S41467-018-04892-4)). At present, and to address the Reviewer's concern, we have reformulated our final discussion to include a few additional considerations on the points discussed above and to cover some of the work previously published.

Discussion, pages 21-22, lines 458-471:

*"MBNL protein inactivation in astrocytes is a determinant event behind the coordinated dysregulation of gene sets associated with cytoskeleton and cell adhesion. The lower activity of MBNL in DMSXL astrocytes results from the combined co-localization with RNA foci and the surprising reduction in total protein. MBNL protein downregulation was previously reported in proliferating DM1 myoblasts in the presence of unaltered transcript levels⁴⁵, through mechanisms that have not been fully elucidated. The regulation of *Mbnl1* and *Mbnl2* gene expression is complex and multifaceted, involving alternative transcription and translation initiation sites, splicing events, miRNA and circRNA species³⁵. It is tempting to speculate that changes in *Mbnl1/Mbnl2* transcription initiation and/or exon 1 skipping may shift translation initiation to produce unstable protein isoforms, as previously reported in DM1 skeletal muscle⁴⁶. Alternatively, miRNA-mediated translation suppression in DMSXL astrocytes may result in MBNL1 and MBNL2 protein downregulation⁴⁷. In both cases, lower protein levels would not be accompanied by a corresponding reduction in *Mbnl1* and *Mbnl2* transcripts."*

Overall, this is a comprehensive study that puts forward novel hypotheses and provides substantial experimental evidence to support it. The in vivo relevance is also clear. I enjoyed reading it and I have no doubt it will be appreciated by Nature Communications readers.

We thank Reviewer #4 for highlighting the novelty of our findings and for their kind remarks.

Reviewers' Comments:

Reviewer #1:

Remarks to the Author:

I have taken a look at the changes made to the manuscript by Dr. Gomes-Pereira and colleagues and am happy with the changes.

I think the manuscript is now suitable for publication.

Reviewer #4:

Remarks to the Author:

In this revised submission the authors have satisfactorily addressed my recent comments and responded to all concerns from other reviewers. The manuscript improved significantly since first submission. This is a high quality work and I believe it is acceptable for publication.